# Sustainable and recyclable super engineering thermoplastic from biorenewable monomer

Seul-A Park [1,5], Hyeonyeol Jeon [1,5], Hyungjun Kim [2,5], Sung-Ho Shin [1], Seunghwan Choy[3], Dong Soo Hwang [3], Jun Mo Koo [1], Jonggeon Jegal [1], Sung Yeon Hwang [1,4], Jeyoung Park [1,4] & Dongyeop X. Oh [1,4]

Environmental and health concerns force the search for sustainable super engineering plastics (SEPs) that utilise bio-derived cyclic monomers, e.g. isosorbide instead of restricted petrochemicals. However, previously reported bio-derived thermosets or thermoplastics rarely offer thermal/mechanical properties, scalability, or recycling that match those of petrochemical SEPs. Here we use a phase transfer catalyst to synthesise an isosorbide-based polymer with a high molecular weight >100 kg mol$^{-1}$, which is reproducible at a 1-kg-scale production. It is transparent and solvent/melt-processible for recycling, with a glass transition temperature of 212 °C, a tensile strength of 78 MPa, and a thermal expansion coefficient of 23.8 ppm K$^{-1}$. Such a performance combination has not been reported before for bio-based thermoplastics, petrochemical SEPs, or thermosets. Interestingly, quantum chemical simulations show the alicyclic bicyclic ring structure of isosorbide imposes stronger geometric restraint to polymer chain than the aromatic group of bisphenol-A.

[1] Research Center for Bio-based Chemistry, Korea Research Institute of Chemical Technology (KRICT), Ulsan 44429, Republic of Korea. [2] Department of Chemistry, Incheon National University, Incheon 22012, Republic of Korea. [3] Devision of Integrative Bioscience and Biotechnology, Pohang University of Science and Technology (POSTECH), Pohang 37673, Republic of Korea. [4] Advanced Materials and Chemical Engineering, University of Science and Technology (UST), Daejeon 34113, Republic of Korea. [5] These authors contributed equally: Seul-A Park, Hyeonyeol Jeon, Hyungjun Kim. Correspondence and requests for materials should be addressed to S.Y.H. (email: crew75@krict.re.kr) or to J.P. (email: jypark@krict.re.kr) or to D.X.O. (email: dongyeop@krict.re.kr)

Since plastics have become indispensable in our life, their consumption has exponentially increased[1]. The colossal demand for plastics has led to a large amount of wastes. For example, abandoned electronics notably create printed circuit board (PCB) waste. The typical content of metals, plastics, and ceramics in PCBs is ~40, 30, and 30 wt%, respectively[2,3]. Among plastics, thermosets, e.g. epoxy, and pseudo-thermoplastics, e.g. polyimide, generally have higher thermal stability; thus, they are more preferred to thermoplastics as materials for PCB[4,5]. After curing, thermosets and pseudo-thermoplastics do not melt and dissolve; the separation of metal from PCB requires harsh chemical degradation or pyrolysis of plastics, and the recycling of plastics is difficult[4,5]. If electronic parts are made of thermally durable thermoplastics, both plastics and metals from PCB can be effectively recycled by melting or dissolution[6]. Likewise, substituting thermosets with thermoplastics for many other applications increases the recycling rate of plastic wastes.

According to superiority of thermal and mechanical performances, thermoplastics are generally classified in the following order: commodity plastics <engineering plastics (EPs) < super engineering plastics (SEPs). There is no appropriate quantitative standard for the precise classification because most physical properties of thermoplastics exist across all the above-mentioned three classes[7,8]. In polymer science, glass transition temperature ($T_g$) is a general indicator to represent thermomechanical characteristics of polymers. In the same order, the three classes of thermoplastics typically have the $T_g$ ranges of <100, 100–150, and >150 °C[1,7–10]. SEPs, also known as high-performance or specialty thermoplastics, are gradually replacing thermosets and pseudo-thermoplastics as thermally and mechanically robust materials for aircrafts, automobiles, electronics, dental devices and in household/children's products because of their recyclability[2,11]. Poly (arylene ether)s (PAEs) are a major group of SEPs, and they include polysulphone (PSU), polyether ether ketone, and polyphenylsulfone[12,13].

In recent, the many environmental concerns associated with plastic's constituents have led to the search for sustainable high-performance thermoplastics that are entirely or partially derived from bio-derived feedstocks, instead of petrochemicals, and match those that they replace in terms of thermomechanical properties[1]. Aromatic petrochemicals such as bisphenol-A (BPA), biphenols, styrenes, and terephthalates are key monomers in determining the thermal and mechanical properties of EPs and SEPs; however, many of them are toxic and pollute the environment. Among the EPs and SEPs, PSU and polycarbonate (PC) are widely used as transparent and heat/stress-resistant parts of electronic and biomedical devices such as circuit boards, battery seals, heat shields, power circuits, and dental instruments. There is great public health concern about BPA in PSU and PC, because it causes developmental and reproductive problems in humans[11–16].

The growing environmental and health concerns have prompted efforts to substitute toxic petro-based aromatic monomers for plastics[17–20] by bio-derived cyclic compounds, such as isosorbide (1,4:3,6-dianhydro-D-glucitol, ISB)[21–24], 2,5-furandicarboxylic acid[25–29], sugar[30], terpene[31–35], lignin derivatives[36–41], and others[42,43]. ISB, a bicyclic sugar derivative, is an attractive alternative of BPA[23,44–46]. The ISB moiety enhances the mechanical, thermal, and optical properties of the host polymer due to its unique molecular structure[47–56]. Moreover, the safety of ISB has been demonstrated by its use in pharmaceuticals and cosmetics. The commercial application of ISB production technology has been developing over the past few years[53,54]. A French agricultural company recently has achieved the world's highest annual high-purity ISB production of 20,000 tons.

Bio-based high $T_g$ thermoplastics are defined as polymers that (i) are entirely or partially derived from bio-derived feedstocks, (ii) have $T_g$ of >150 °C, and (iii) are melt processable[1]. However, bio-based high-performance thermoplastics, i.e. with a high $T_g$ of >150 °C, have been relatively less reported than bio-based thermosets/ pseudo-thermoplastics and have the following limitations. Thus, it has limited the expansion of renewable thermoplastics in industry and has created an opportunity in academia[1]. The condensation polymers for thermoplastics from bio-derived cyclic compounds have a relatively low molecular weight of <50 kg mol$^{-1}$, even though they could achieve a $T_g$ as high as SEPs due to their rigid cyclic structure (Supplementary Table 1)[55–58]. The high melt viscosity of the bio-derived cyclic-compound-based polymers causes diffusion limitations, which actually hinder the chain growth[45]. As a result of their low molecular weight, most of these bio-based polymers with a high $T_g$ of >150 °C have poor or unknown mechanical properties, let alone practical applications. To the best of our knowledge, there are few studies investigating their mechanical properties as well as melt processability[29,39].

Here, we report the production of an ISB-incorporated PAE with a molecular weight over 100 kg mol$^{-1}$, which has not been reported before for bio-based high $T_g$ polymers from the current literature on thermoplastic research (Fig. 1a, b). It achieves a high $T_g$ of 212 °C, a tensile strength of 78 MPa, and a remarkable coefficient of thermal expansion (CTE) are 23.8 and 81.2 ppm K$^{-1}$ at 30–80 and 80–200 °C, respectively. These values surpass those of most commercial EPs, SEPs, thermosets, and pseudo-thermoplastics (Fig. 1c, d and Supplementary Tables 1–3). This polymer can be recycled through melting and dissolution.

## Results

**Preparation of sustainable super engineering thermoplastics.** A typical synthesis route of aromatic PAEs is based on nucleophilic aromatic substitution ($S_NAr$). Briefly, an aromatic diol, e.g. BPA, reacts with an aromatic di-halide, e.g. 4,4′-difluorodiphenyl sulfone (DFPS) in a polar aprotic solvent containing potassium carbonate ($K_2CO_3$). BPA forms a complex consisting of $K^+$ and nucleophile [phenoxide]$^-$, which displaces the halogen of DFPS[59]. Water and potassium halide are generated as byproducts. Water is typically removed by toluene-mediated azeotropic distillation, because water reduces the nucleophilicity of anions and induces the hydrolysis of halide monomers.

There are major difficulties in obtaining ISB-based PAEs with high molecular weights. In contrast to the aromatic diol, the aliphatic diol of ISB does not form alkoxide readily in the presence of $K_2CO_3$. The alkoxide of ISB is less stable than the phenoxide. In addition, ISB is highly hygroscopic, which makes the removal of water challenging. However, in this study a high molecular weight ISB-based sulfone-type PAE, coded as SUPERBIO, was successfully synthesised with the aid of a phase-transfer catalyst instead of toluene distillation, otherwise it only gave an oligomer (Fig. 1a). Here, ISB and DFPS were polymerised at 155 °C in dimethylsulphoxide (DMSO) in the presence of a crown-ether, 18-crown-6 (5 mol% to ISB) under a $N_2$ flow. The chemical structure and molecular weight of products were analysed using nuclear magnetic resonance (NMR) and gel permeation chromatography (GPC, Supplementary Figs 1, 2). SUPERBIO achieved a weight-average molecular weight ($M_w$) of 114 kg mol$^{-1}$ and an inherent viscosity ($\eta_{inh}$) of 0.83 dL g$^{-1}$. Further, the molecular weight could be reproduced at 1-kg-scale, which is higher than those of other bio-based high $T_g$ thermoplastics by a factor of $10^2$–$10^4$ (Fig. 1b and Supplementary Table 1).

SUPERBIO achieved a huge jump in molecular weight for bio-based high $T_g$ condensation polymers. Note that the molecular

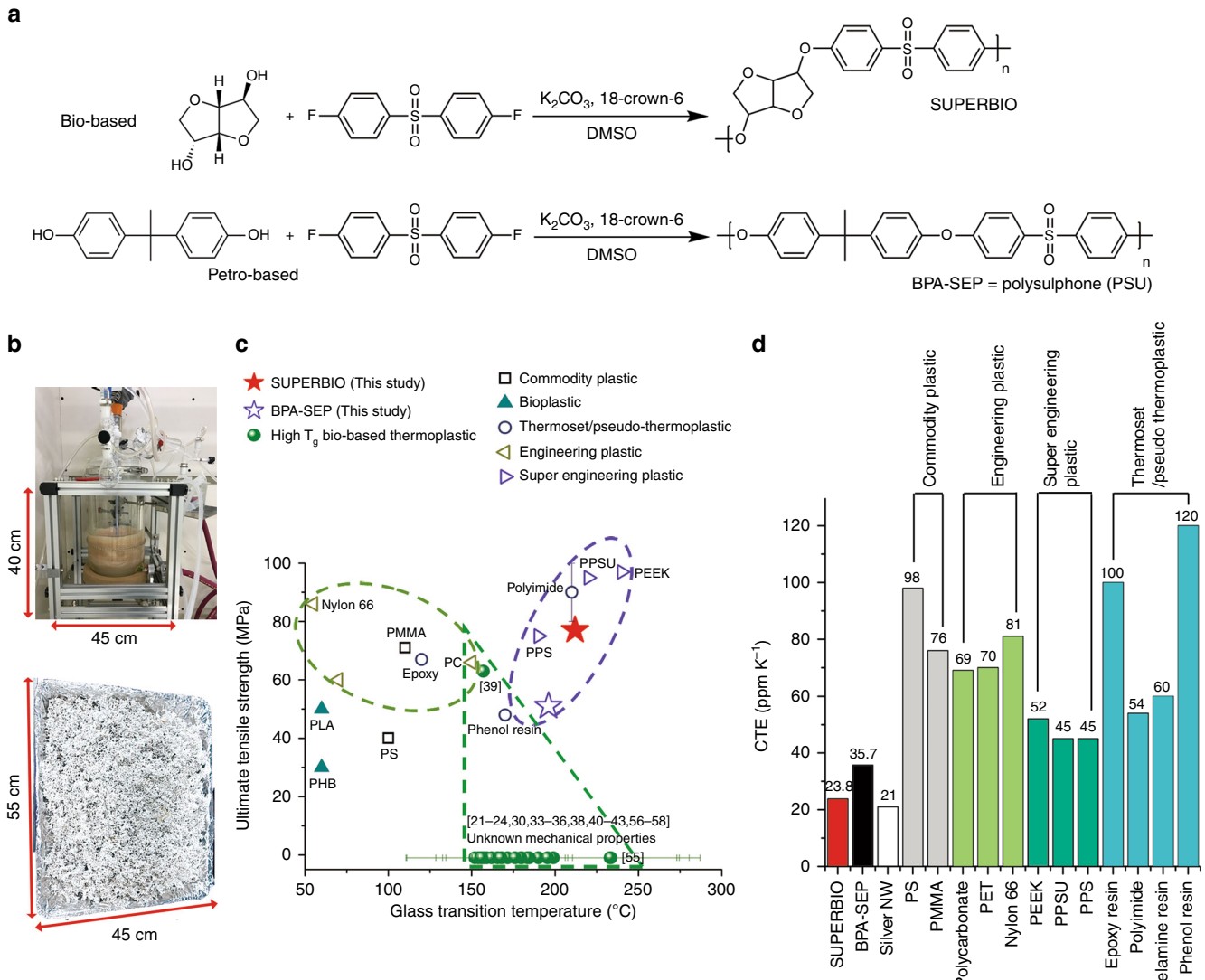

**Fig. 1** Preparation and thermal/mechanical properties of the bio-based super engineering plastic. **a** Synthetic scheme of (top) ISB- and (bottom) BPA-based poly(arylene ether)s, which are designated as SUPERBIO and BPA-SEP, respectively. **b** Photograph of the polymerisation reactor at 1-kg-scale, and the SUPERBIO product. **c** Ashby plot of ultimate tensile strength versus glass transition temperature. **d** Coefficient of thermal expansion of petrochemical plastics/thermosets/ pseudo-thermoplastics, bio-based high $T_g$ thermoplastics, SUPERBIO, and BPA-SEP at 30–80 °C

weight data may not be directly comparable, since the literature data were determined by diverse methods (NMR, mass spectroscopy, etc.). Nevertheless, considering the high inherent viscosity ($\eta_{inh}$) and great mechanical properties of SUPERBIO compared to other reported bio-based high $T_g$ thermoplastics, a much higher molecular weight of SUPERBIO can be presumed. Also, the $\eta_{inh}$ and GPC data of commercial PSU supports this claim (Supplementary Table 1). As a control, a PSU with $M_w = 151$ kg mol$^{-1}$ and $\eta_{inh} = 1.61$ dL g$^{-1}$ was synthesised with BPA and DFPS in the presence of the crown-ether, and coded as BPA-SEP. Instead of DFPS, a sulphur-free co-monomer is applicable to this polymer system. An ISB-based ketone-type PAE with a similar $M_w$ (93.6 kg mol$^{-1}$) called SUPERBIO-K was synthesized with a monomer combination of ISB and 4,4′-difluorobenzophenone by a method identical to DFPS synthesis (see Method Section & Supplementary Fig. 3).

To understand the role of the crown-ether, two other ISB-based PAEs were synthesised. One was prepared with toluene instead of crown-ether, and the other used neither crown-ether nor toluene (See Methods section). Their $M_w$ values were 72 and

12 kg mol$^{-1}$, respectively. It is undeniable that water critically reduced the $S_NAr$ reaction efficiency. The effect of the crown-ether can be explained by well-recognised theories[60,61]. As a phase-transfer catalyst, the crown-ether increases the solubility of ISB and K$_2$CO$_3$ in DMSO, promotes the alkoxide formation of ISB, and makes the [alkoxide]$^-$ naked by keeping the K$^+$ at a distance. The result increases the substitution efficiency on the halide group.

**Solvent/melt processing and mechanical characterisation.** The prepared SUPERBIO was simply solvent-casted into a ~70 μm-thick free-standing film, with a transparency of >97% in the visible light range (Fig. 2a and Supplementary Fig. 4). The SUPERBIO film was resilient enough to withstand rough handling. To demonstrate this, the film was folded into an origami ship and unfolded. It did not tear or show fatigue-induced whitening afterwards, possibly because the sufficient $M_w$ minimises molecular slipping (Supplementary Movie 1).

The SUPERBIO film ($M_w = 114$ kg mol$^{-1}$) exhibited superior tensile, tear, and impact strengths compared to BPA-SEP

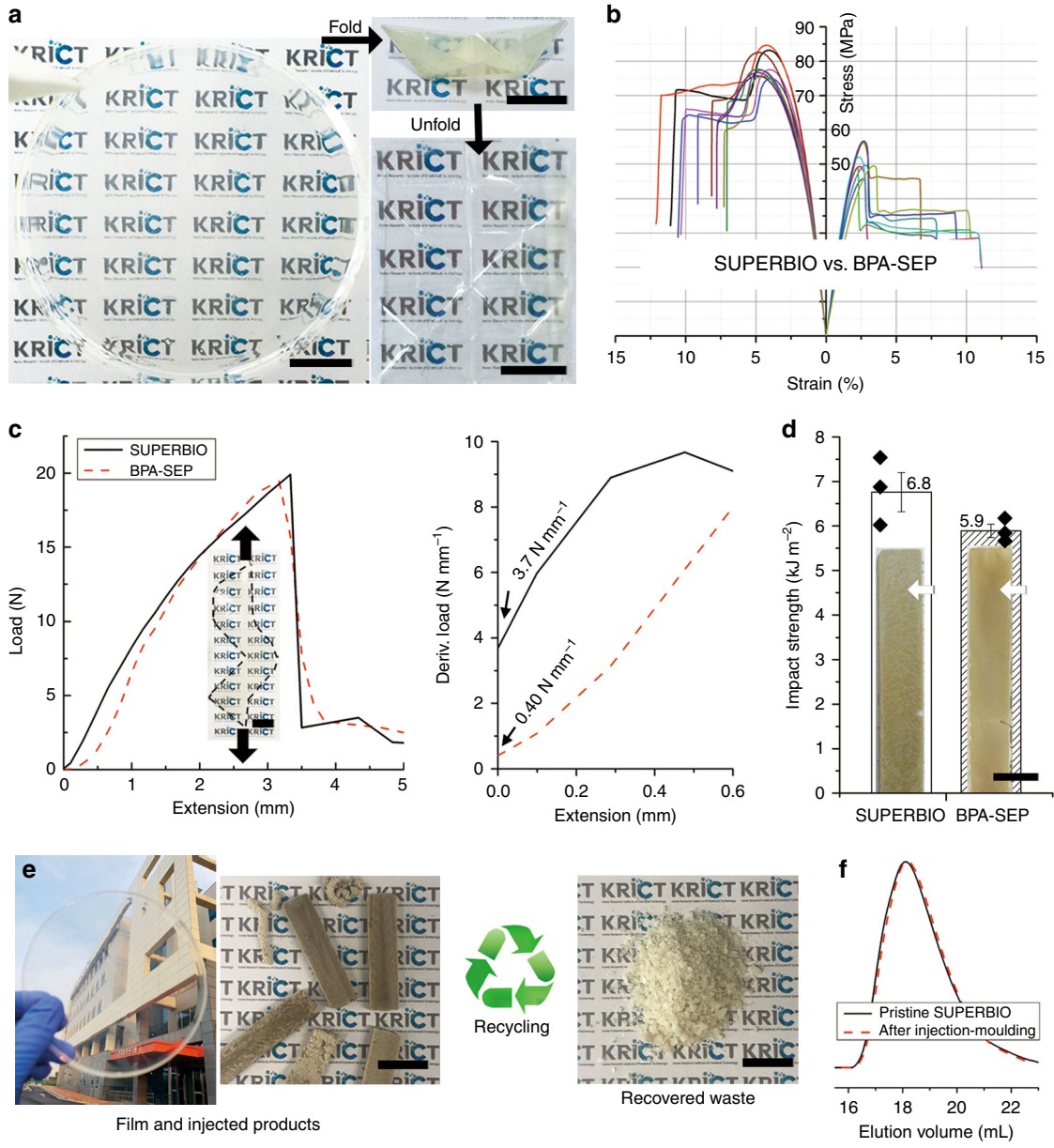

**Fig. 2** Damage-tolerant bio-based super engineering plastic. **a** Photographs of the solution-casted pristine, origami-folded (top), and unfolded (bottom) films of SUPERBIO (scale bar: 1 cm, Supplementary Movie 1). **b** Tensile stress-strain curves of SUPERBIO ($n = 10$) and BPA-SEP ($n = 8$) films. **c** (Left) Original and (right) differential tear load-distance curves of SUPERBIO and BPA-SEP films. Inset is photograph of the specimen for tear test (KS M ISO 34-1:2014, scale bar: 1 cm). **d** Impact strength of the injection-moulded SUPERBIO and BPA-SEP. Inset is photograph of the rectangular bar-shaped specimens for the impact test (scale bar: 1 cm). Each impact strength value represents the mean and standard error of triplicate samples. **e** Recycling of SUPERBIO products (scale bar: 1 cm). **f** DMF-GPC profiles before/after thermal processing. $M_w$s of pristine and injection-moulded SUPERBIO were 92.2, and 89.7 kg mol$^{-1}$, respectively

(Fig. 2b–d and Supplementary Fig. 5). SUPERBIO's tensile Young's modulus (3.7 GPa), ultimate tensile strength (UTS, 78 MPa), tensile toughness (5.6 MJ m$^{-3}$), tensile elongation (7.9 %), and tear strength (160 kN m$^{-1}$) were 1.2, 1.5, 1.8, 0.9, and 1.2 times those of BPA-SEP, respectively. It is worth to note that the initial differential tear stress value of SUPERBIO is 9.3-fold higher than that of BPA-SEP, as shown in Fig. 2c. Therefore, SUPERBIO has better resistance against tear initiation and propagation at cracks or notches. This argument is validated by the tear resistance test under applied load weights (Supplementary Fig. 6 and Supplementary Movie 2), in which SUPERBIO could bear a load more than 2-fold higher than BPA-SEP. SUPERBIO-K achieved a Young's modulus of 3.8 GPa, a UTS of 76 MPa, a

tensile toughness of 8.7 MJ m$^{-3}$, and an elongation at break of 13% (Supplementary Fig. 3). The tensile performances of SUPERBIO-K are as high as those of SUPERBIO.

We have investigated the effects of $M_w$ on the tensile properties of SUPERBIO (Supplementary Fig. 7). Along with the sample having its actual $M_w$ of 114 kg mol$^{-1}$, SUPERBIO samples with three different $M_w$ values of 30, 63, and 85 kg mol$^{-1}$ were synthesized by controlling the reaction time, and the tensile properties of the four different samples were compared. The Young's modulus and UTS gradually increased with $M_w$ to the aforementioned values achieved by the SUPERBIO sample with $M_w = 114$ kg mol$^{-1}$ because of increasing chain entanglements. The tensile toughness and elongation at break were the highest at

the $M_w$ of 85 kg mol$^{-1}$. The increasing strength with $M_w$ negatively impacts the ductility of SUPERBIO.

The melt processing is a representative recycling method and more cost effective and greener than solvent processing. However, there has only been a few studies on the melt processing of bio-based high $T_g$ thermoplastics, probably because of the small synthesis scale or inadequate molecular weight/viscosity. The SUPERBIO or BPA-SEP films (4 g) was chopped and each melted with polyethylene glycol (PEG) of 0.4 g as a plasticiser at 270 °C for 8 min, and then injection-moulded into a rectangular bar (see Methods for details) (Fig. 2d). In contrast to petrochemical plastics, many biopolymers brown at melt processes[27]. The SUPERBIO bar became brown relatively as less as the BPA-SEP one without an antioxidant. SUPERBIO achieved a 1.2-fold higher impact strength (6.8 kJ m$^{-2}$) than BPA-SEP. Moreover, unchanged molecular weights after injection-moulding confirmed the thermal stability of SUPERBIO at the melt state as well as recyclability (Fig. 2e, f). To evaluate the thermal stability of SUPERBIO in detail during melt processing, we have monitored the $M_w$ change of SUPERBIO during five programmed cycles of heat treatments (Supplementary Fig. 8). Each cycle consists of heating (30 to 270 °C) and cooling (270 to 30 °C) with a ramp rate of 10 °C min$^{-1}$ under a nitrogen atmosphere. The $M_w$ hardly changed until the second heat treatment. The $M_w$ of SUPERBIO decreased to only 9% after the fifth heat treatment. This suggests that SUPERBIO can be recycled through a series of melting and moulding[62].

SUPERBIO exhibits greater thermal dimensional stability due to the rigid aliphatic fused bicyclic ring of the ISB moiety, as revealed by our quantum chemical simulation (to be discussed

later). SUPERBIO presented a $T_g$ value of 212 °C, 16 °C higher than BPA-SEP (Supplementary Fig. 9). Notably, the CTE values of SUPERBIO at 30–80 and 80–200 °C are 23.8 and 81.2 ppm K$^{-1}$, being 1.5 and 10-fold lower than those of BPA-SEP (35.4 and 826 ppm K$^{-1}$), respectively, as shown in Fig. 3a. SUPERBIO's CTE value at 30–80 °C is as low as that of silver nanowires (AgNWs)[63], and lower than those of commercial SEPs and thermosets/pseudo-thermoplastics including polyimides and melamine resins by a factor of >2 (Fig. 1d and Supplementary Table 3).

To evaluate the thermal degradation stability, the samples' 5 and 10 wt% loss temperatures ($T_{d5}$ and $T_{d10}$) were measured using a thermogravimetric analyser (TGA). SUPERBIO had $T_{d5}$ = 411 °C and $T_{d10}$ = 422 °C, which are high or mid-high among the bio-based high $T_g$ thermoplastics (Supplementary Table 1) and other bio-based commodity plastics ($T_{d5}$ <316 °C)[45]. SUPERBIO only lost less than 1 wt% until 360 °C, a temperature that is higher than the typical melt processing temperature of 250–300 °C for SEPs (as was used to prepare the specimens for the impact strength test). However, SUPERBIO has poorer thermal degradation stability than BPA-SEP, which has $T_{d5}$ = 497 °C and $T_{d10}$ = 502 °C (Supplementary Fig. 10), because the thermal degradation stability is more strongly associated with bond dissociation energy (BDE) than molecular weight. The aliphatic bonds of ISB have lower BDE values compared to the conjugated bonds of BPA[45].

**Quantum chemical simulation.** The higher thermal and mechanical properties of SUPERBIO over BPA-SEP are quite surprising, because the aromatic BPA has been considered to be

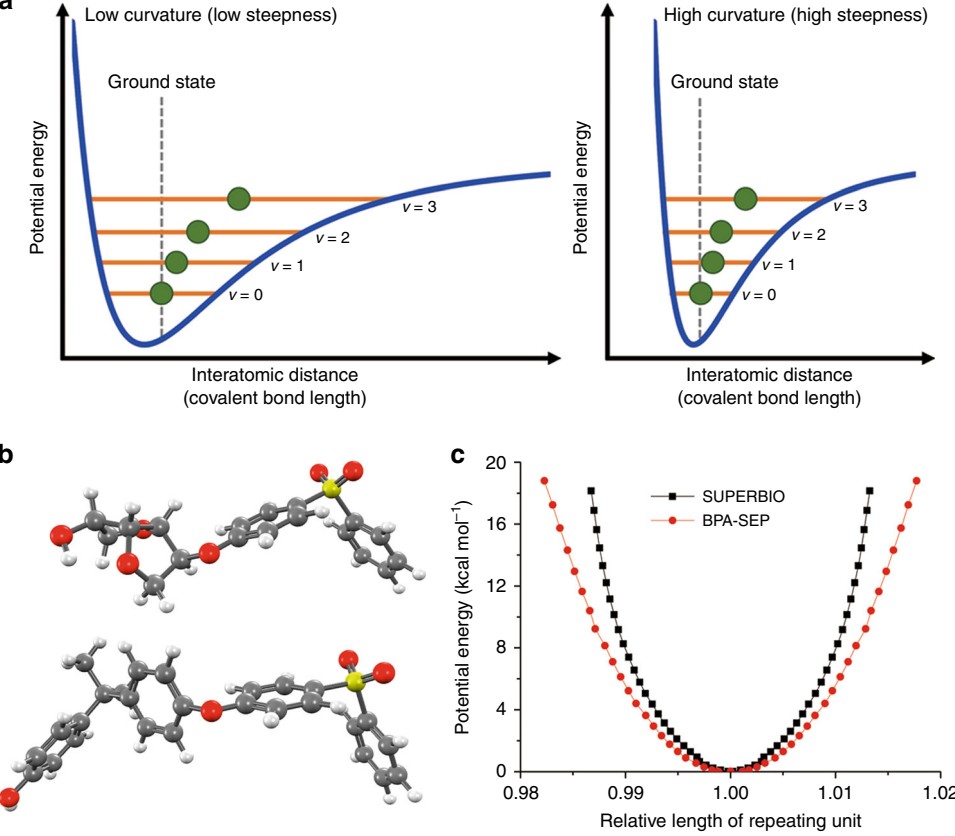

**Fig. 3** Quantum chemical simulation. **a** Schematic illustration of different vibration energy levels in a potential energy well (x-axis: interatomic distance). **b** Optimised ground state geometries of (top) SUPERBIO and (bottom) BPA-SEP repeating units. C, black; H, white; O, red; and S, yellow. **c** Potential energy curve along with relative length of repeating unit

more suited for such purposes than the aliphatic ISB. In the glassy state ($T < T_g$), a polymer behaves like a typical elastic solid, the thermal and mechanical expansion is the sum result of different energy-dependent oscillatory bonds: strong covalent and weak van der Waals bonds[64]. By a quantum chemical simulation, we have studied the effects of ISB as well as BPA on the thermal/mechanical properties of a single molecule of SUPERBIO, i.e. we have explored their contributions on the geometric restraint of the covalent linkages with exclusion of the physical interactions (i.e. inter-polymeric interactions). This approach marks a starting point to understand isosorbide's thermomechanical property at a fundamental level. Supplementary Discussion includes the detailed description of quantum chemical simulations.

A correlation between the vibrational energy gap and the relative covalent bond length was derived by calculating an anharmonic potential energy curve (PEC), $V(x) = \frac{1}{2}kx^2 - \lambda x^3$ where $k$ is the spring constant of chemical bond and $\lambda$ is the anharmonicity constant (Fig. 3a). The one-dimensional average position (green dot position) is expressed as $x^{(1)} = \frac{3\lambda\hbar}{\sqrt{mk}}\left(\nu + \frac{1}{2}\right)$ where $\hbar$ is the reduced Planck constant, $m$ is the mass, and $\nu$ is the vibrational energy level. For example, on increasing the temperature, as the vibrational energy is excited from ground state toward $\nu = 4$, the green dot deviates from the original position, i.e. bond length increases. This indicates that the increasing system energy gives rise to elongation of covalent bonds. The shape of PEC is dependent on the geometric restraint of covalent linkages. At the given $\nu$, i.e. temperature, the higher curvature (or steeper slope) of PEC results in the lower elongation of chemical bonds.

In the theoretical model, the repeating unit for each polymer is chosen with an assumption that the relative length change of each polymer chain is not significantly different from that of the repeating unit. This assumption is reasonable because both are fully amorphous polymers with only short-range order. After the geometry of each systematically elongated repeating unit was optimised to consider the relaxation effect from angle changes according to the density functional theory, well-known as DFT, by using the B3LYP/6-31 G* basis set (Fig. 3b), a PEC along with the relative bond length was calculated for a given vibrational level (Fig. 3c).

The vibrational levels are supported within the PEC like in the case of a Morse potential. The steepness of the PEC is related to the energy required to stretch the bonds of each repeating unit. The data indicate that, compared to BPA-SEP, the energy required to attain the same degree of geometric alternation for SUPERBIO is 1.41–1.57-fold higher. Interestingly, the steepness for SUPERBIO keeps increasing as the repeating unit is lengthened, while that of BPA-SEP remains relatively unchanged. The simulation outcome suggests that, when the structure is thermally extended, the unique fused bicyclic ring structure of ISB imposes stronger geometric restraint in a single molecule than the planar benzene group of BPA.

This theory is also useful for elucidating the mechanical property. A single molecular $k$ in a rigid and glassy polymer chain can be decided by the stretching and distortion of the covalent bonds, which can be derived from second-order derivative of the PEC[65]. The elongation of the repeating unit in SUPERBIO has a $k$ value 1.57 fold higher than that of BPA-SEP. However, the $k$ data cannot totally reflect the bulk mechanical properties. The Young's modulus and UTS are affected to a high degree by the molecular slipping and noncovalent failures, as well as macroscopically defective morphologies. Nevertheless, it is manifest that the structure restraint of SUPERBIO-single molecule by ISB playing an important role in the high mechanical properties.

**Fabrication of a transparent and flexible electric device.** To make the best use of SUPERBIO in consideration of its advantages noted above especially of the low CTE (Fig. 4a), its potential applications in advanced electronics were investigated. Initially, SUPERBIO and BPA-SEP films were spin-coated with AgNWs, forming two types of transparent electrodes. The SUPERBIO electrode was considered to sustain latent thermal and mechanical stresses inside the electronics as well as polyimide does. This electrode was highly transparent and bendable, with a high visible light transmittance of >90%, and a sheet resistance change of less than 20% at a bending radius of 0.6 mm (Fig. 4b, c and Supplementary Fig. 11).

The electrodes were gradually heated to three temperature stages of 250, 300, and 350 °C, and each temperature stage was kept for 1 h under a nitrogen atmosphere (Supplementary Fig. 12). The sheet resistance of neither electrodes increased, instead it remained at 22–24 Ω sq$^{-1}$ until 250 °C. At 300 °C, the sheet resistance of BPA-SEP jumped to >1 kΩ sq$^{-1}$ within 15 min, while that of SUPERBIO only increased moderately to ~110 Ω sq$^{-1}$ (Fig. 4d and Supplementary Movie 3). As a result, the light-emitting diode (LED) on the BPA-SEP electrode burned out at 300 °C, while that on the SUPERBIO electrode stayed on within the experimental time of 1 h. As shown in Supplementary Figs 12b, 13, the morphology of AgNWs on both electrode surfaces was examined using atomic force microscopy (AFM) and field-emission scanning electron microscopy (FE-SEM). The non-heated SUPERBIO and BPA-SEP electrodes both presented highly percolating networks of AgNWs. After the heat treatment of 300 °C, the AgNW network of the SUPERBIO electrode was relatively well conserved, while that on the BPA-SEP film was disconnected. It is obvious that the low CTE of SUPERBIO led to a lower thermal dimensional stress on the AgNWs than that of BPA-SEP (Supplementary Figs 13, 14).

An organic light-emitting diode (OLED) device was fabricated using SUPERBIO film as a transparent and heat resistant substrate. The AgNW embedding strategy was adopted to make an OLED substrate with a smoother surface, which helps prevent electrical shorts between neighbouring electrical components (Fig. 4f and Supplementary Figs 15–17). After the routine fabrication processes of a green OLED device, it successfully emitted green light even when it was strongly bent (Supplementary Fig. 18). The SUPERBIO film endured 250 °C thermal evaporation processes during the OLED device fabrication. For recycling the electrode, the SUPERBIO electrode (1 g) was dissolved in DMAc (9 g). The solution was then filtered by a Nylon syringe-filter with a pore size of 0.45 μm to separate AgNWs and successfully solvent-casted into a transparent free-standing film (Fig. 4g–i)[6].

**Biocompatibility tests for biomedical applications.** The increasing demand for orthodontic devices with better aesthetics has prompted the development of transparent plastic brackets and wires to replace metals in braces[11]. PSUs and glass fibre-reinforced PCs as bracket materials provide good colour stability, low biofilm fouling, and long-term mechanical durability for several years. However, it has been reported that the BPA in PC and PSU might be released, causing enamel defects after long-term exposure[16]. Here, SUPERBIO is suggested as a orthodontic material as well as diverse transparent bio-devices because it is likely to have better long-term mechanical and dimensional stability than BPA-SEP, according to the time-temperature superposition theory.

The bracket materials must provide hydration resistance because of the moist physiological environment. SUPERBIO and BPA-SEP were incubated in deionized (DI) water at 25 or 90

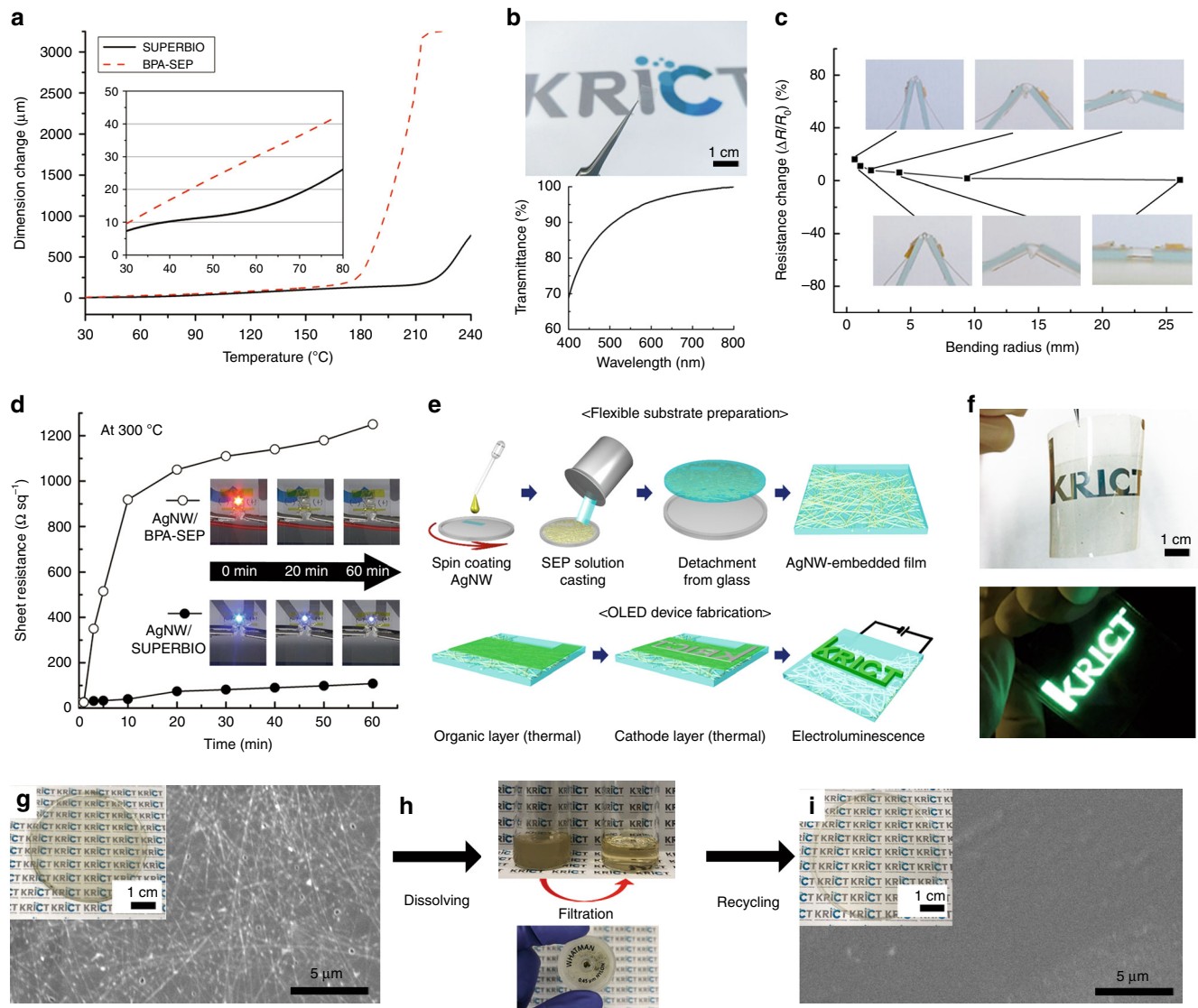

**Fig. 4** Thermal properties of bio-based SEP for flexible optoelectronics. **a** Linear thermal expansion curves of SUPERBIO and BPA-SEP films (inset: an enlarged part of the graph). **b** (Top) Photograph and (bottom) light transmission spectra of AgNW-coated SUPERBIO film. **c** Sheet resistance of AgNW-coated SUPERBIO film at different bending radii. **d** Comparison of sheet resistance changes of AgNW-coated SUPERBIO and BPA-SEP after thermal annealing at 300 °C for 1 h. Heat resistant operation of LED on the AgNW-coated SUPERBIO and BPA-SEP films at 300 °C, shown as time-lapse photograph over 1 h (Supplementary Movie 3). **e** Schematic illustration of flexible OLED fabrication in which SUPERBIO endured 250 °C thermal processes. **f** Photographs of (top) the fabricated OLED and (bottom) its electroluminescence operation with the applied voltage of 8 V under bending. The recycling of the SUPERBIO electrode. **g** Photograph (inset) and FE-SEM image of AgNW-coated SUPERBIO film. **h** Electrode dissolved in DMAc was filtered using a Nylon syringe-filter. Then, the filtered solution was casted into a transparent free-standing film. **i** Photograph (inset) and FE-SEM image of the resultant recycled SUPERBIO film

°C for 24 h; the specimen weight and $M_w$ of SUPERBIO and BPA-SEP were then measured. The experimental conditions did not affect the specimen weight and $M_w$ of both types of samples (Supplementary Fig. 19).

To test the physiological adaptation of SUPERBIO, in vitro toxicity tests of the L-929 cell line was carried out for SUPERBIO and BPA-SEP, based on ISO 10993-5 (Fig. 5a). In a typical method, the cells were cultivated in (1) 20% (v/v) extract-containing, (2) pristine, and (3) 5% DMSO-containing complete growth medium, as an experimental group, negative, and positive controls, respectively. SUPERBIO has negligible cytotoxicity to L-929, i.e. more than 80% of viability of the negative control. BPA-SEP also showed insignificant cytotoxicity to L-929, probably because the level of unreacted BPA was below the sub-toxic concentration. In addition, a protein adsorption of SUPERBIO

was as low as that of BPA-SEP (Supplementary Figs 20, 21). This property is beneficial in the orthodontic brackets to prevent the formation of biofilms.

The in vivo biocompatibility test of SUPERBIO was conducted by a contract clinical research organization [Daegu Gyeongbuk medical innovation foundation (DGMIF)] using a rat subcutaneous model, following the ISO 10993-6 Annex A standard (Fig. 5b). The ethical issue was approved by institutional animal care and use committee (IACUC) (Korea), and the approval code is DGMIF-18012301-00. In the experimental group, 10 mm-diameter films of SUPERBIO, and BPA-SEP were implanted into the subcutaneous connective tissue of each rat ($n = 5$), and high density polyethylene (HDPE) film was used as a negative control. The rats were sacrificed after 12 weeks. The histopathologic analyses of the subcutaneous tissues were conducted after routine

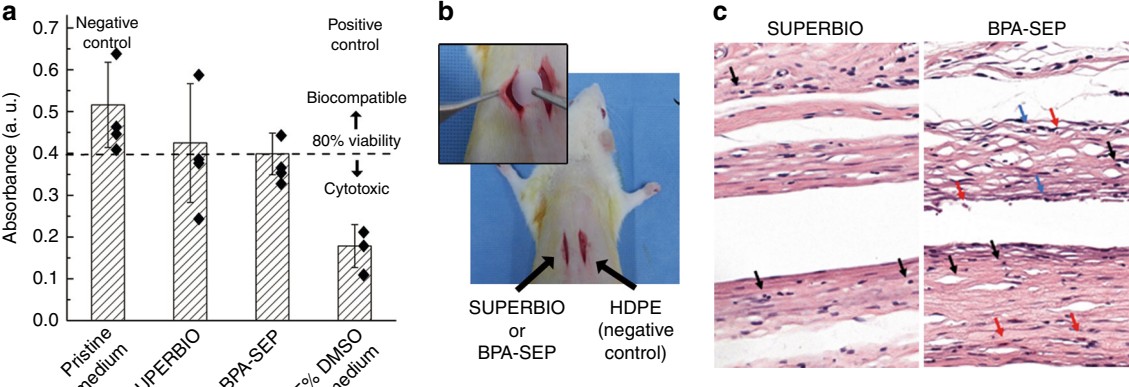

**Fig. 5** In vitro and in vivo biocompatibility tests. **a** In vitro cytotoxicity tests using pristine media (negative control) and those containing 5% DMSO (positive control) or polymer film extracts, following ISO 10993-5. Each value represents the mean and standard deviation of quintuplicate samples. **b** In vivo experiment procedure: rat subcutaneous connective tissues ($n = 5$) with HDPE (negative control), SUPERBIO, and BPA-SEP films. **c** Representative histopathologic tissue images after 12 weeks of healing. Arrows for inflammatory cells: red (polymorphonuclear cell), black (lymphocyte), and blue (macrophage)

fixing and dyeing processes. The histopathological tissue images (Fig. 5c) show that SUPERBIO had less inflammatory cells than BPA-SEP. The inflammatory responses were scored semi-quantitatively by a pathologist, according to the ISO 10993-6 guidelines: non-irritant < slight < moderate < severe, in the order of inflammatory reaction intensity. The SUPERBIO film scored the lower inflammatory intensity of slight, compared to BPA-SEP that scored moderate. In the limited experimental scope, SUPERBIO shows favourable biocompatibility without chronic and severe inflammation and with low biological interaction upon tissues (Fig. 5 and Supplementary Fig. 22). Certainly, the interpretation of the in vivo experiment cannot be extended to the commercial PSU because chemical companies utilise high-level technology to remove residual monomers.

## Discussion

In conclusion, we prepared a sustainable SEP using ISB, a bio-derived heterocyclic monomer. The high molecular weight of this SEP was achieved with the aid of 18-crown-6 to activate $S_NAr$ polymerisation. The superior mechanical strength and remarkable thermal dimensional stability, along with great transparency, processability, production scalability, and biocompatibility, realize this material as an ideal candidate for applications in extreme environments, where many bio-based polymers cannot compete. It endured thermal processing for the OLED fabrication, and its good biocompatibility was revealed. Our quantum chemical simulation provided a reasonable explanation for the higher robustness and lower thermal expansion of SUPERBIO compared to BPA-SEP. The distinctive repeating unit of ISB induces 1.41–1.57 folds higher geometric restrain when the structure is pulled, as compared to BPA. This sustainable SEP opens up applications where the use of plastics is limited by health and environmental concerns. As a future research scope and in order to avoid the environmental effects of the petrochemical part of SUPERBIO, an SEP completely derived from biological resources can be developed.

## Methods

**Materials**. Isosorbide (ISB) was kindly supplied by Roquette Frères (Lestrem, France) and used after recrystallization in acetone. Bisphenol-A (BPA, 99%), bis(4-fluorophenyl) sulfone (DFPS, 99%), and 4,4′-difluorobenzophenone (99%) were purchased from TCI (Tokyo, Japan) and recrystallized in methanol. Potassium carbonate ($K_2CO_3$, 99%, Sigma–Aldrich, St. Louis, MO, US) was ground into a fine power and dried with phosphorus pentoxide under a vacuum. Dimethyl sulfoxide (DMSO, 99.7%), $N,N′$-dimethylacetamide (DMAc, 99.8%), toluene (99.5%), acetic

acid (HPLC grade), methanol (HPLC grade), 18-crown-6 (99%), methylene chloride ($CH_2Cl_2$, HPLC grade), trifluoroacetic acid (TFA, 99%), PEG that has a molecular weight of 400 g mol$^{-1}$, and organic light-emitting diode (OLED) materials were purchased from Sigma–Aldrich (St. Louis, MO, USA) and used without further purification. Silver nanowire (AgNWs) aqueous solution having a diameter and length of $35 \pm 5$ nm and $25 \pm 5$ μm, respectively, was purchased from Nanopyxis Co. Ltd. (Jeonju, Korea).

**Synthesis of SUPERBIO and BPA-SEP**. ISB (3.00 g, 20.5 mmol) [or BPA (4.68 g, 20.5 mmol)], DFPS (5.21 g, 20.5 mmol), and $K_2CO_3$ (3.55 g, 25.7 mmol) were added into a dried glass flask equipped with a mechanical stirrer and a Dean–Stark apparatus. Then, 0.05 molar equivalent of 18-crown-6 (0.271 g, 1.02 mmol) against diols, and DMSO (22.4 ml, 37 wt/v% to the monomer content) were added to a flask via a gas-tight syringe under a dry nitrogen atmosphere. 18-Crown-6 is a well-known additive used to reduce the side reaction of condensation polymerization[66]. The reaction mixture was heated for 24 h (4 h for BPA-SEP) at 155 °C with a mild nitrogen flow. After polymerization, the reaction mixture was diluted with DMSO (20 ml), cooled to room temperature, and precipitated into a water/methanol mixture (1 L, 50/50 vol%) containing acetic acid (10 ml). To remove residual additives, the solid was filtered and re-precipitated after dissolving in DMAc. The precipitated polymer was filtered off and washed with DI water and methanol. The polymer was dried under a vacuum at 80 °C overnight. SUPERBIO final polymer product (7.09 g, 96%), $M_w$: 113,900 g mol$^{-1}$, PDI: 2.04, $^1$H NMR (DMSO-$d_6$, 300 MHz, ppm): δ 7.90–7.86, 7.28–7.25, 7.04–6.93, 7.28–7.25, 1.71. BPA-SEP final polymer product (8.71 g, 96%), $M_w$: 151,300 g mol$^{-1}$, PDI: 1.80, $^1$H NMR (CDCl$_3$, 300 MHz, ppm): δ 7.87–7.78, 7.14–7.11, 4.99, 4.52, 3.93–3.87. A series of different molecular weight SUPERBIO were synthesized by decreasing reaction time.

**Synthesis of an ISB-based ketone-type SUPERBIO-K**. The same synthetic procedures of SUPERBIO were conducted, except that 4,4′-difluorobenzophenone (4.57 g, 20.5 mmol) was used instead of DFPS. The final polymer product (6.46 g, 97%), $M_w$: 93,600 g mol$^{-1}$, PDI: 1.98, $^1$H NMR (CDCl$_3$, 300 MHz, ppm): δ 7.82–7.78, 7.06–7.00, 5.09–5.06, 4.96–4.89, 4.72–4.70. 4.27–4.09.

**Synthesis of an ISB-based PAE #1**. The same synthetic procedure of SUPERBIO was conducted except as follows: (1) the absence of 18-crown-6; (2) toluene (5.0 ml) was added to the flask before the polymerization; and (3) after charging the chemicals, the reaction mixture was heated to 120 °C, and the water was removed azeotropically with toluene through a Dean–Stark trap for 2 h. The final polymer product (7.07 g, 96%), $M_w$: 71,900 g mol$^{-1}$, PDI: 1.75.

**Synthesis of an ISB-based PAE #2**. The same synthetic procedure of SUPERBIO was conducted except for the absence of 18-crown-6. The final polymer product (7.02 g, 95%), $M_w$: 11,800 g mol$^{-1}$, PDI: 1.88.

**Structure and molecular weight analysis**. $^1$H NMR spectra were obtained with a Bruker AVANCE 300-MHz spectrophotometer (Billerica, MA, USA). Samples were dissolved in CDCl$_3$ for BPA-SEP and SUPERBIO-K, and DMSO-$d_6$ for SUPERBIO, respectively. Tetramethylsilane (TMS) was used as an internal standard and as a reference for chemical shift. Inherent viscosity was measured using an Ubbelohde viscometer with an eluent of a co-solvent [$CH_2Cl_2$/TFA; 9:1, v/v] at $25 \pm 0.1$ °C. Number-average molecular weight ($M_n$), weight-average molecular

weight ($M_w$), and the polydispersity index (PDI) were determined by gel permeation chromatography (GPC) equipped with an ACQUITY refractive index detector using chloroform ($N,N'$-dimethylformamide, DMF) for BPA-SEP and SUPERBIO-K (or SUPERBIO) as a mobile phase flowing with a velocity of 0.6 mL min$^{-1}$. ACQUITY APC XT columns (Mixed bed, maximum pore size 450 Å, Waters Corp., Milford, MA, USA) were kept at 40 °C during the measurements. Universal calibration was based on polystyrene standards.

**Solution-casted film preparation**. Polymer solutions were prepared by dissolving polymers in DMAc to be 10 wt%. Each of the solutions was poured into a glass dish, and dried at 90 °C in a convection oven for 2 day. Transmittance experiments of the films were performed on a UV-2600 (Shimadzu Corp., Kyoto, Japan) UV/vis spectrometer at a resolution of 0.1 cm$^{-1}$. The contact angle was measured using a contact angle analyser (Phoenix 300, Surface Electro Optics, Gyeonggi-do, Korea). The volume of the sessile water drop was controlled at 0.2 μL using a microsyringe. The contact angle results were the average values calculated for five drops at different places on the samples.

**Thermal properties**. A differential scanning calorimeter (DSC) (Q2000, TA Instruments, New Castle, DE, USA) was operated with a heating and cooling rate of 10 °C min$^{-1}$ from 30 °C to 250 °C in an $N_2$ atmosphere. $T_g$ was determined at the second heating cycle. Thermal degradation was evaluated using a thermogravimetric analyser (PerkinElmer, Waltham, MA, USA) under a nitrogen purge flow of 50 mL min$^{-1}$. Samples were scanned from room temperature to 800 °C with a heating rate of 10 °C min$^{-1}$. CTE was measured using a thermomechanical analysis (TMA) instrument (TA Instruments) with a probe force of 20 mN and a heating rate of 10 °C min$^{-1}$ in a temperature range from 30 °C to 250 °C under an $N_2$ flow. The film specimens for the TMA testing had a length, width, and thickness of 15 mm, 5 mm, and 70 μm, respectively.

**Mechanical properties**. Tensile properties were measured using a universal testing machine (UTM) made by Instron (High Wycombe, UK) with a drawing rate of 10 mm/min, according to ASTM D638 (American Society for Testing and Materials). The polymer films for tensile properties were prepared on a glass petri dish by the solvent casting method. To reduce the roughness of the fabricated films, the films were hot-pressed at 200 °C under 100 bar for 5 min. The test specimens were cut into a dog-bone shape, which has a length, width, and thickness of 63.50 mm, 3.18 mm, and 100–115 μm, respectively, using a jockey type-cutting machine (Supplementary Fig. 23). Each tensile property values represents the mean and standard error. The tear tests were conducted by two methods: (1) a standard tear strength measurement (Fig. 2c) and (2) a customized tear resistance comparison under applied load weights (Supplementary Fig. 6 and Supplementary Movie 2). The tear strength measurement test according to KS M ISO 34-1:2014 was performed using an Instron UTM with a drawing rate of 100 mm/min. Angle type specimens (non-nicked, 90 °C), which have a length, width, and thickness of 100 mm, 19 mm, and 100–115 μm, respectively, were prepared for the tear test. The tear resistance test under applied load weights was performed as follows. Polymer films were cut into a rectangular shape having dimensions of 60 mm × 30 mm × 155–168 μm. A 10-mm-long notch was formed at the middle point on the side of 60 mm. One side of the film was fixed with a grab of a standing clip and the other side was gravitationally pulled down by loading 10-g-weights one-by-one until the film was completely torn. An impact strength test was performed as follows. SUPERBIO (or BPA-SEP) (4 g) and PEG (0.4 g) was dissolved in DMAc (40 ml) and dried at 100 °C in a convection oven for 2 day. The impact strength specimens (bar type) were prepared by injecting grinded powder into a Haake™ Minijet (Thermo Scientific, Waltham, MA, USA). The sample was melted at 270 °C for 8 min, and then injection-moulded into a rectangular bar. The cylinder temperature, injection pressure, filling time, and mould temperature were 270 °C, 500 bar, 20 s, and 200 °C, respectively. The impact strength test was measured with a pendulum impact testing machine (HIT-2492, Jinjian Testing Instrument Co., Ltd., Chengde, China) in accordance with the KS M ISO 180:2012. All impact test samples were V-shape notched. The test specimen was supported as a vertical cantilever beam and broken by a single swing of a pendulum. The velocity of the hammer was 3.5 m s$^{-1}$. The standard specimen for ISO is a Type 1 A multipurpose specimen with a size of 80 mm × 10 mm × 4 mm. For each case, a total of three samples were tested at 25 °C. Each impact strength value represents the mean and standard error of triplicate samples.

**AgNW-coated SUPERBIO/BPA-SEP electrodes**. A SUPERBIO (or BPA-SEP) film was fixed on a Si wafer with Kapton® tape. The film was surface-treated with UV-ozone for 30 min. The AgNW solution with a concentration of 0.5 wt% was spin-coated on the film at 500 rpm for 30 s and dried at room temperature for 12 h. The AgNW-coated film was pre-annealed to 120 °C for 1 h under an argon atmosphere. At the same atmosphere, the film was gradually heated to the three temperature stages of 250, 300, and 350 °C, and each stage was halted for 1 h. Then, the surface electrical resistance of the film was measured at the different temperature stages. The morphologies of the heated or non-heated AgNW-coated films were measured using an AFM, MultiMode V Veeco microscope (Plainview, NY, USA) with tapping mode, and a FE-SEM (Tescan MIRA3, Brno, Czech Republic).

To characterize the electrical resistance changes under mechanical bending, AgNW-coated films are held onto the microscope slide glasses and compressed by the uniaxial stretching stage. The bending radius and strain are characterized in geometrical aspects based on measured dimensions with callipers (Supplementary Fig. 11).

**OLED device fabrication**. Firstly, the OLED device fabrication started with the preparation of AgNW-embedded SUPERBIO film substrate. In order to define the pixel area, a thin and rectangular-shaped PDMS film was attached onto the glass petri dish. Then, AgNW ink was spin-coated (1000 rpm, 40 s) onto the glass petri dish, followed by thermal baking at 120 °C for 3 min. Polymer solution (10 wt% in DMAc) was poured into the as-prepared glass petri dish and the solvent was dried at 90 °C in a convection oven. After the film was totally casted, it was detached from the glass petri dish and cut into a 5 × 5 cm square shape, which was used as an AgNW-embedded SUPERBIO film substrate (Supplementary Figs 15–17). Afterwards, a thermal evaporator with a temperature of 250 °C was used to form organic layers as follows. 20 nm 1,4,5,8,9,11-hexaazatriphenylenehexacarbonitrile (HATCN) as a hole-injection layer, 50 nm $N,N'$-di(1-naphthyl)-$N,N'$-diphenyl-(1,1'-biphenyl)-4,4'-diamine (NPB) as a hole transport layer, and 5 nm tris(4-carbazoyl-9-ylphenyl)amine (TCTA) as an electron blocking layer were deposited in sequence. Then, for a green-coloured phosphorescence light-emitting-layer, 15 nm TCTA/2,2',2''-(1,3,5-benzinetriyl)-tris(1-phenyl-1-H-benzimidazole) (TPBi) as a host and tris[2-phenylpyridinato-C$^2$,N]iridium(III) (Ir(ppy)$_3$) as a dopant with a concentration of 12 % was deposited, followed by the deposition of 40 nm TPBi as an electron transport layer. Lastly, 1.5 nm 8-quinolinolato lithium (Liq) as an electron injection layer and 100 nm aluminium as a cathode were deposited with a 'KRICT-shaped' shadow mask. The fabricated OLED devices were operated with an applied voltage of 8 V, using an electrical source meter (Keithley 2400, Cleveland, OH, USA).

**Quantum chemical simulation**. The quantum chemical simulation method is described in the Supplementary Discussion chapter.

**In vitro cytotoxicity test**. The in vitro cytotoxicity test was performed based on international standard ISO 10993-5.[78] The cytotoxicity test started with liquid extracts of plastic materials (SUPERBIO or BPA-SEP). Each plastic film was immersed in a cell growth media with the plastic at a ratio of 1 cm$^3$ sample to 1 ml media at 36 ± 1 °C for 72 h. The culture media extract was filtered by a syringe filter. A fibroblast cell line L-929 was seeded in 96-well plates with $10^4$ cells per well and cultured at 36 ± 1 °C for 24 h in 5% $CO_2$ atmosphere in Dulbecco's modified Eagle's medium (DMEM) supplemented with 10% FBS, 100 U ml$^{-1}$ penicillin G, 100 μg ml$^{-1}$ streptomycin, and 0.025 μg ml$^{-1}$ amphotericin B. The culture media was replaced with the neat (negative), the 20% (v/v) extract-containing (experiment), and 5% DMSO-containing (positive) complete growth media. Next, they were incubated for an additional 24 h to expose the cell to the extract. To evaluate the viability, the media was replaced by 100 μl of 10% (Cell Counting Kit-8, CK04, Dojindo, Inc., Rockville, MD, USA) (CCK-8) solution which can measure cellular respiration activity. Afterwards, L929 cells were incubated for 2 h at 36 ± 1 °C. The incubated media were transferred to fresh 96-well plates for colorimetric assessment using a microplate reader at 450 nm. The absorbance intensity below 80% cell viability compared to negative control is considered a cytotoxic effect (ISO 10993-5:2009(E)). The data of quintuplicate samples are expressed as mean ± the standard deviation.

**In vitro protein adsorption test**. Empty 24-well culture plates were filled with 1 × 1 cm SUPERBIO (or BPA-SEP) films and incubated with 4.5 g L$^{-1}$ bovine serum albumin (BSA) solution at 36 ± 1 °C for 4 h. Next, BSA solution was removed and non-specific BSA bound to the specimen was excluded by washing with phosphate-buffered saline (PBS) several times. The tightly bound BSA was desorbed through sonication for 20 min using 0.025% sodium dodecyl sulfate (SDS) in PBS. The amount of adsorbed protein to specimen was quantified by Bradford assay based on colorimetric absorbance measurement at 590 nm. The data of quintuplicate samples are expressed as mean ± the standard deviation.

**In vivo biocompatibility test**. All surgical procedures were performed by a (public) contract clinical research organization, Daegu Gyeongbuk medical innovation foundation (DGMIF) (http://www.dgmif.re.kr/eng/index.do) with the approval of the national institutional review board (IRB). The samples were implanted in male Sprague–Dawley rats (8-weeks-old, 250–300 g) ($n$ = 5). The rats were allowed free access to food and water in a temperature- and humidity-controlled room (22 °C, 50%) with a 12/12 h day/night cycle (8 am/8 pm). Each rat was anesthetized with an intramuscular injection of 50 mg ml$^{-1}$ Zoletil 50 (tiletamine and zolazepam; Virbac, Carros, France) and 23 mg ml$^{-1}$ Rompun (xylazine; Bayer, Leverkusen, Germany), and the scalp was incised carefully. One experimental sample and one negative control (HDPE) films (10 mm diameter circle) were implanted in two different regions (15 mm incision) of subcutaneous tissues of a rate. The incised skins were closed with 4/0 Dafil sutures (Ethicon, Somerville, NJ) and disinfected with a povidone after the procedures. After the surgery, the rats

were bred in their cages for 12 weeks. Then, the rats were sacrificed for histological analyses.

The tissues samples were routinely dehydrated, paraffin embedded, cut, and stained with haematoxylin and eosin (H&E). Then, the cross-sections of the tissues were examined and semi-quantitatively evaluated according to International Standard (ISO 10993-6, Annex A) criteria for biological evaluation of the local effects of medical devices after implantation by a pathologist. The local effects were evaluated by comparison of the tissue response caused by the experimental samples and the negative control. The scoring system is the histological evaluation of the extent of the area affected. The presence, number, and distribution of polymorphonuclear cells, lymphocytes, plasma cells, macrophages, giant cells, and necrosis were evaluated. The tissue changes by neovascularization, fatty infiltration, and fibrosis were evaluated.

**Reporting summary**. Further information on research design is available in the Nature Research Reporting Summary linked to this article.

## Data availability

The source data that support the findings of this study are available (https://doi.org/10.6084/m9.figshare.8121314). We provide the source data underlying Fig. 2b–d, 2f, 3c, 4a–d, and 5a, and Supplementary Figs 1–5, 7, 9, 10, 12, and 21.

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

## Acknowledgements

D.X.O. and J.P. acknowledge funding from Korea Research Institute of Chemical Technology through Core Program (SI1941-20, KK1941-10, KK1941-30). S.Y.H. acknowledges funding from the Ministry of Trade, Industry and Energy (MOTIE, Korea) through the Technology Innovation Program (10070150). J.J. acknowledges funding from the Ministry of Trade, Industry and Energy (MOTIE, Korea), the Korea Institute for Advancement of Technology (KIAT) through the System Industrial Base Institution Support Program (P0001939). We are thankful to Prof. Sang Youl Kim and Prof. Myungeun Seo at KAIST, and Prof. In Hwan Jung at Kookmin University for a fruitful discussion. H.K. is grateful to Prof. Paul Zimmerman at University of Michigan for providing computation resources.

## Author contributions

S.A.P. and H.J. synthesized and characterized the polymer. H.K. performed the quantum chemical simulation. S.H.S. and J.M.K. performed the electronic device fabrication. S.C. and D.S.H. performed biocompatibility tests. J.J. and S.Y.H. analysed the data. J.P. and D. X.O. wrote the manuscript. S.Y.H., J.P., and D.X.O. supervised the whole project and revised manuscript. All authors have given approval to the final version of the manuscript. S.A.P., H.J., and H.K. equally contributed to this work.

## Additional information

**Competing interests:** The authors declare no competing interests.

