## [Transparent Peer Review File · Nature Communications]

Reviewers' comments:

Reviewer #1 (Remarks to the Author):

The paper is focused on synthesis and characterization of isosorbide-based polymers with high MW. Interestingly, preliminary results are also offered in terms of several possible applications. I evaluate this contribution as worth to be published after accounting for the comments below.

Major comments

- Lines 24: from the very beginning, the synthesized polymer is defined "sustainable". However, a sulfur-containing co-monomer is involved: how much is the sulfur content affecting the polymer sustainability?

- Lines 52-55: the term Super Engineering Plastic (SEP) is widely used in the manuscript as well as in the title: a clear and comprehensive definition of SEP should be provided to univocally identify the class of materials (refs. 2 and 3 are not enough).

- Line 70: the statement "mainly because of the low reactivity of these monomers" is quite generic and should be better specified. In the case of polycondensation, diffusion-limitations are actually hindering the chain growth, thus limiting the molecular weight. On the other hand, when talking about ROP, the intrinsic reactivity of endgroups matters, even though the choice of a suitable catalyst can affect the reaction rate at large extent. Please, rephrase and better specify.

- Line 121: the mechanical properties of the polymer are indeed quite remarkable. However, the elongation at break of about 8% (the same value for PET is >300%) should be mentioned and commented.

- Lines 124-125: a polymer called BPA-SEP was prepared as control material. Why higher MW was considered? (more similar MW of the control and the actual material would make the comparison more meaningful) Which was the conversion of the two monomers?

- Lines 152-153: the different MW of the two polymers is again affecting the reliability of this comparison. Once more, the preparation of polymers with much more similar MW would be very desirable.

- Line 165: which plasticizer was actually used? Please, specify and motivate its choice.

- Lines 166-168: the less coloring without antioxidant is a good result. But what about the MW after the melting? MW values before and after should be provided to support the good behavior during melting.

- Line 297: is more than 80% viability of the negative control good enough to claim bio-compatibility? Values for the same cell line and different polymers should be provided (from the literature or own experiments).

- Line 328: the polymer is claimed to be superior when compared to "unrecyclable thermosets and pseudo-thermoplastics". But what about its recyclability? This aspect deserves deeper investigation and at least comments about possible recycling (by chemical and/or mechanical methods) should be discussed. This is indeed a crucial aspect for its successful large scale production.

Minor comments

- Lines 62-65: the contributions of Gandini (e.g. <https://onlinelibrary.wiley.com/doi/pdf/10.1002/pola.24812>, FDCA-isosorbide polyesters) and De

Jong (e.g. <https://pubs.acs.org/doi/10.1021/bk-2012-1105.ch001>, PEF) should be also mentioned.

- Line 69: low-molecular-weight should become low molecular weight.
- Line 268: SEPERBIO  SUPERBIO (please check everywhere in the text)
- Figure 5a: please correct "pristine" on the horizontal axis.
- Line 319: ...this SEP, was achieved...
- References: all author names should be provided.

Reviewer #2 (Remarks to the Author):

The manuscript entitled "Sustainable and Recyclable Super Engineering Thermoplastic from Biorenewable Monomer" is an interesting report of a thermoplastic derived from isosorbide. The authors details the synthesis, characterization, application, and simulations of the material in this report. While the material and properties are of interest and impressive, there are several revisions this reviewer suggests.

- 1) The polymer contains an isosorbide monomer but is not really green or sustainable in the synthesis, in particular because it is copolymerized with bisphenol S (a majority of the mass fraction). After degradation, the bisphenol derivative will still be released into the environment and will cause the same severe toxic effects that are well known and studied. Thus, this is not a sustainable polymer and that aspect of the research should significantly be downplayed.
- 2) Only one molecular weight is studied of this system. It would be useful to study several molecular weights to improve overall understanding of the structure-activity relationships.
- 3) The English grammar should be carefully proofread and corrected throughout the manuscript.
- 4) Information about the ethical treatment of animals and all approved animal protocol information should be included in this manuscript.
- 5) There are many general references the authors include for sustainable polymers but more emphasis should be placed in the referencing on the large body of research in isosorbide polymers.

Reviewer #3 (Remarks to the Author):

This manuscript describes a novel synthetic route to achieve high molecular weight polymers from the biorenewable chemical isosorbide. The resulting polymers have very advantageous thermal and mechanical properties, equivalent to "super engineering thermoplastics", for which there are few examples in the biorenewable polymers literature. However, I have some concerns about the claims made in the paper, as well as the quality of the discussion and data presentation in the manuscript. These concerns must be addressed before considering this manuscript for publication.

1. The claims in the introduction and abstract regarding current state of the biorenewable polymers literature are wildly overstated, and this takes credibility away from this study. The authors needs to carefully consider the present literature and then discuss the novelty of their study in this context. For example:
 - a. In the abstract on page 2 line 26: "the research efforts produced only thermosets or low molecular weight thermoplastics..."
 - b. On page 3 line 47, "these researches have mostly produced commodity-grade thermoplastics with a glass transition temperature < 100 C..."However, a quick google search has yielded the following study on an isosorbide-based polymethacrylate, which has Tg around 130C (Gallagher, Hillmyer and Reineke, ACS Sustainable Chemistry and Engineering 2015, 3, 662). Additionally, the lignin-based polymer in ref 31 (Holmberg, A.L., Reno, K.H., Nguyen, N.A., Wool, R.P. & Epps III, T.H. Syringyl Methacrylate, a Hardwood Lignin-Based Monomer for High-Tg Polymeric Materials. ACS Macro Lett. 5, 574-578

(2016)) has a reported Tg of 205 C. These two examples suggest that a more thorough literature search could yield other biorenewable polymers, and even isosorbide-derived polymers, with high Tg's, minimizing the impact of this work. It is therefore essential to place this work in the context of the relevant literature, which likely goes beyond the two articles mentioned here.

2. If plastics used in electronics are going to be the main motivation for this work, then a larger discussion in the introduction is needed around this issue. For example, what percentage of plastic waste is from electronics? Is the BPA-SEP polymer used extensively in these applications and therefore the best comparison for the new bio polymers? The introduction does not provide enough context and motivation for this aspect.

3. Also in the introduction, can you provide more specific motivation for the use of isosorbide over other biorenewable sources? (lignin, etc) What about availability of the source?

4. I did not see water sorption mentioned in this article. This is a negative aspect that typically plagues isosorbide-based polymers. How does the water sorption of SUPERBIO compare to BPA-SEP?

5. Page 4 line 76: please provide literature context for the "unprecedented" molecular weight of your polymer

6. Page 6 line 117: please provide literature context for the claim that the scale is " 10^2 - 10^4 " times greater than other biobased high Tg thermoplastics

7. Regarding tensile testing: a) Why only test 3 specimens? Typically in the range of 10 test bars yields more reliable results. b) Please add error bars to the parameters on page 9 lines 153-154. c) On page 23 line 454: these are not the correct bar dimensions for ASTM 638-03. Therefore you cannot be following this ASTM standard, as the bar dimensions are a critical aspect of the ASTM standard. The tensile properties will be very different for a film vs bulk specimen.

8. Page 9 line 169: How do these results confirm the thermal stability? This is just a comparison to BPA-SEP. A better approach would be to report properties after each thermal cycle and see if they change.

9. Page 10 line 170: You refer to the quantum chemical simulations here without any details. This is awkward for the reader. I suggest bringing the quantum chemical simulations discussion right after the thermal property discussion, and not mentioning it here on this line. (Therefore, bring the quantum chemical simulations discussion to before the electronic device discussion).

10. Page 12 line 220: It seems that BPA-SEP is not a particularly suitable polymer for this application. Therefore, it is very confusing why this is the polymer chosen for comparison purposes. If you want to show that SUPERBIO is a suitable polymer in this application, why not compare it to a polymer that is currently used for this purpose?

11. Similarly, is the BPA-SEP actually used in the orthodontic devices discussed on page 16? It seems these are PSUs and PCs, based on your description of current state of the art. Why is BPA-SEP used for comparison purposes in this application?

12. In general the application discussions (electronic device and medical implant) seem somewhat out of place in this article. Perhaps if better motivation were provided in the introduction for one application (such as the electronic devices), then this would be a good way to end the article. However, putting both in with little motivation provided makes the article very difficult to follow.

13. Page 14 line 250: please provide derived equations and more info on the calculations.

14. There are grammatical and other writing style issues in this manuscript that need to be addressed. For example”:

- a. Page 2 line 26: “However, the research efforts” should be “However, previous research efforts” or similar
- b. Listing all of the properties of other polymers does not seem appropriate for the abstract
- c. Page 3 line 47, “these researches have” should be “this research has”
- d. Page 3 line 53, “with a high Tg of >150 C than engineering plastics...” – are you trying to say that the Tg’s of SPEs are 150 C greater than that of EPS?
- e. Page 4 line 80: listing specific figure and table numbers in the introduction is not appropriate
- f. Sentence at the end of the introduction seems out of place
- g. Page 9 line 167: “became coloured relatively as less as..”

**Answer to #1 reviewer**

*The paper is focused on synthesis and characterization of isosorbide-based polymers with high MW.*
*Interestingly, preliminary results are also offered in terms of several possible applications. I evaluate this*
*contribution as worth to be published after accounting for the comments below.*

**Answer:** We greatly appreciate the constructive, very detailed comments and suggestions of this reviewer.

*Major comments*

*1. Lines 24: from the very beginning, the synthesized polymer is defined "sustainable". However, a sulfur-*
*containing co-monomer is involved: how much is the sulfur content affecting the polymer sustainability?*

**Answer:** We agree with the reviewer's opinion. It is difficult to entirely ignore the environmental effects of
DFPS (the sulfur-containing co-monomer, 4,4'-difluorodiphenyl sulfone). Herein, we demonstrate the
objective of this study including the preparation of non-sulfur polymers, organization of the partially bio-
based polymers reported, and recycling of prepared polymers. In addition, we discuss the possible limitation
of this material in the Conclusion Section.

Firstly, to address the environmental issue of the sulfur-containing co-monomer, **an ISB-based, ketone-type**
**poly(arylene ether) (PAE) was synthesized using a sulfur-free monomer** (4,4'-difluorobenzophenone)
instead of DFPS. Furthermore, we showed that the mechanical properties of the ISB-based ketone-type PAE
are as high as those of **SUPERBIO** that is made of DFPS.

We have added the following content.

**(Page 8 Line 147)** Instead of DFPS, a sulphur-free co-monomer is applicable to this polymer system. An
ISB-based ketone-type PAE with a similar M_w (93.6 kg mol^{-1}) called **SUPERBIO-K** was synthesized with a
monomer combination of ISB and 4,4'-difluorobenzophenone by a method identical to DFPS synthesis (see
Method Section & Supplementary Fig. 3).

**(Page 10 Line 190) SUPERBIO-K** achieved a Young's modulus of 3.8 GPa, a UTS of 76 MPa, a tensile
toughness of 8.7 MJ m^{-3} , and an elongation at break of 13% (Supplementary Fig. 3). The tensile
performances of **SUPERBIO-K** are as high as those of **SUPERBIO**.

**(Method section) Synthesis of an ISB-based ketone-type SUPERBIO-K.**

The same synthetic procedures of **SUPERBIO** were conducted, except that 4,4'-difluorobenzophenone (4.57
51 g, 20.5 mmol) was used instead of DFPS. The final polymer product (6.46 g, 97%), M_w : $93,600 \text{ g mol}^{-1}$, PDI:
1.98, $^1\text{H NMR}$ (CDCl_3 , 300 MHz, ppm): δ 7.82–7.78, 7.06–7.00, 5.09–5.06, 4.96–4.89, 4.72–4.70, 4.27–
4.09.

**(Supplementary Information, Page 10)**

**Supplementary Figure 3.** (a) Synthetic route of an ISB-based ketone-type PAE called **SUPERBIO-K** (M_n
= 44.4 kg mol^{-1} , M_w = 93.6 kg mol^{-1} , PDI = 1.98). (b) Tensile stress-strain curves, (c) data statistics ($n = 3$),
and (d) $^1\text{H-NMR}$ spectrum of **SUPERBIO-K**.

Secondly, we would like to demonstrate **the relative superiority of our polymers in terms of**
 **“sustainability” over the previously reported high T_g biopolymers.** As shown in the Supplementary
 Table 1 and demonstrated in a review paper about high T_g biopolymers (*J. Mater. Chem. A* **6**, 9298-9331
 (2018); DOI: 10.1039/C8TA00377G), most high T_g biopolymers still comprise majorly petrochemicals.
 Nevertheless, most of them have poor or unknown mechanical properties, making them unsuitable for
 practical applications; moreover, their processabilities are mostly unknown. In the present study, the ISB-
 incorporated PAE, i.e., **SUPERBIO** achieved not only high T_g , but also great mechanical properties and
 processability.

We have dragged the following contents from the supplementary information of the original manuscript.

Supplementary Table 1. Characteristics of **SUPERBIO**, **BPA-SEP**, reported bio-based high T_g (>150 °C) thermoplastics, commercial petrochemical SEP, and commercial bioplastics (Continue).

Polymer type ^a	Biomass origin ^a	Polymerization ^a	Crystallinity ^a	M_n^b (kg mol ⁻¹) ^a	η_{inh}^b (dL g ⁻¹) ^a	T_g^b (°C) ^a	T_g/T_{m0}^b (°C) ^a	Biomass content ^b (wt%) ^a	E (GPa) ^a	UTS (MPa) ^a	Melt processing/ recycling ^a	Scale ^b (g) ^a	Reference ^a			
SUPERBIO (Poly(arylene ether)) ^a	Sugar (isohexide) ^a	Condensation ^a	Amorphous ^a	114 ^b	0.83 ^a	212 ^a	411/422 ^a	40-44 ^a	3.4 ^a	77 ^a	O ^a	~1,000 ^a	This study ^a			
BPA-SEP (Poly(arylene ether sulfone)) ^a	Petrochemical (bisphenol-A) ^a			151 ^b	1.61 ^a	196 ^a	497/502 ^a	0 ^a	3.0 ^a	51 ^a		~100 ^a				
Poly(arylene ether sulfone) ^a	Petrochemical (commercial PSU) ^a			35 ^b	0.37-0.39 ^a	190 ^a	~	0 ^a	~	~	~	~		~	Sigma Aldrich ^a	
				50-60 ^b	~	187 ^a		0 ^a							BASF ^a	
Poly(arylene ether ketone) ^a	Sugar (isohexide) ^a			<6 ^b	0.23-0.65 ^a	213-253 ^a	~	<40 ^a	~	~	~	~		~	~7 ^a	55,S1,S2 ^a
	Sugar (FDCA) ^a			<7 ^b	0.29-0.31 ^a	165-170 ^a		44 ^a							~2 ^a	56 ^a
Poly(carbonate-ester) ^a	Sugar (isohexide) ^a		~	0.21 ^a	159 ^a	424/~	32 ^a	~	~	~	~	~	~1 ^a	57 ^a		
	Sugar (isohexide) ^a		>7 ^b	0.31-0.59 ^a	154-161 ^a	~>458-470 ^a	32-42 ^a						~1 ^a	58 ^a		
Polyester ^a	Sugar (isohexide) ^a		32-43 ^b	0.47-0.56 ^a	167-193 ^a	>350/~	61-81 ^a	~	~	~	~	X ^a	~35 ^a	21 ^a		
	Sugar (isohexide/FDCA) ^a		>6 ^b	0.13-0.67 ^a	158-209 ^a	>360/~	82 ^a						~6 ^a	S3,S4,S5 ^a		
	Sugar (isohexide/LA) ^a		>9 ^b	0.11-0.38 ^a	173-196 ^a	>320/~	>99 ^a						~9 ^a	S6 ^a		
	Sugar (isohexide/SA) ^a		65-180 ^b	~	150-181 ^a	385/397 ^a	52-66 ^a						~15 ^a	S7,S8 ^a		
	Sugar (glucitol) ^a	Amorphous ^a	57 ^b	~	156 ^a	385/398 ^a	80 ^a	~	~	~	~	~15 ^a	22 ^a			
		Amorphous ^a	13 ^b	0.32 ^a	154 ^a	~377 ^a	82 ^a	~	~	~	X ^a	~	30 ^a			

Supplementary Table 1. Characteristics of **SUPERBIO**, **BPA-SEP**, reported bio-based high T_g (>150 °C) thermoplastics, commercial petrochemical SEP, and commercial bioplastics (Continue).

Polymer type ^a	Biomass origin ^a	Polymerization ^a	Crystallinity ^a	M_n^b (kg mol ⁻¹) ^a	η_{inh}^b (dL g ⁻¹) ^a	T_g^b (°C) ^a	T_g/T_{m0}^b (°C) ^a	Biomass content ^b (wt%) ^a	E (GPa) ^a	UTS (MPa) ^a	Melt processing/ recycling ^a	Scale ^b (g) ^a	Reference ^a
Polyester ^a	Lignin ^a	Condensation ^a	Semicrystalline ^a	48 ^b	~	157 ^a	~>305 ^a	>99 ^a	11 ^a	63 ^a	O ^a	~8 ^a	39 ^a
			Amorphous (not melting) ^a	~	0.83 ^a	168 ^a	~431 ^a	>99 ^a				~2 ^a	S9 ^a
	Terpene ^a	Ring-opening ^a	2-14 ^b	~	155-243 ^a	~<330 ^a	<51 ^a	~0.5 ^a				33 ^a	
Polycarbonate ^a	Sugar (isohexide) ^a	Condensation ^a	~	<15 ^b	~	<184 ^a	~	<66 ^a	~	~	~	~0.4 ^a	34 ^a
	Sugar (mannose) ^a	Ring-opening ^a	>14 ^b	0.37-0.80 ^a	156-175 ^a	~	31-84 ^a	~1 ^a				23,S10 ^a	
Polyacetal ^a	Lignin ^a	Condensation ^a	~	16 ^b	~	152 ^a	~220-240 ^a	73 ^a	~	~	~	~0.3 ^a	S11 ^a
Polyvinylacetal ^a		Post-modification ^a	36-39 ^b	~	152-159 ^a	265-307 ^a	67 ^a	~1 ^a				36,S12 ^a	
Polyurethane ^a	Sugar (isohexide) ^a	Condensation ^a	Amorphous ^a	53 ^b	~	157 ^a	224 ^a	46 ^a	~	~	~	~2 ^a	38 ^a
Polyketone ^a	Terpene ^a	Radical addition ^a	8-19 ^b	0.23-0.35 ^a	152-191 ^a	>271 ^a	~40 ^a	~5 ^a				24,S13,S14, S15 ^a	
Polydiene ^a	Lignin ^a	Metathesis ^a	~	~100 ^b	~	162 ^a	~327 ^a	>90 ^a	~	~	~	~1 ^a	35 ^a
Polyacrylate ^a		Radical addition ^a	49 ^b	~	156 ^a	380 ^a	82 ^a	~0.1 ^a				40 ^a	
Poly(N-aromatic imide) ^a	Itoic acid ^a	Radical addition ^a	~	18-50 ^b	~	154-205 ^a	302 ^a	~68 ^a	~	~	~	~	41,S16 ^a
Poly(hydroxylacrylamide) ^a	Lactone ^a	~	>19 ^b	~	153-238 ^a	>230 ^a	28-51 ^a	~1 ^a				42,S17,S18, S19 ^a	
Poly(methylene-lactide) ^a	~	~	~	40-100 ^b	~	157-178 ^a	~	50-54 ^a	~	~	~	~2 ^a	43 ^a
Poly(methylene-ester) ^a	Tulipalin A ^a	~	~	76-358 ^b	~	213-254 ^a	>305 ^a	>99 ^a				~0.3 ^a	S20 ^a
		~	~	~600 ^b	~	163-189 ^a	>320 ^a	63-72 ^a	~	~	~	~0.6 ^a	S21 ^a

Thirdly, **SUPERBIO** can be recycled through melting or dissolution (Fig. 2 e,f & Fig. 4g-i). It represents
the sustainability value of this polymer. The recycling minimizes the exposure of the monomer to the
environment.

We have added the following content (Fig. 2e,f & Fig. 4g-i).

**Fig. 2. e**, Recycling of **SUPERBIO** products (scale bar: 1 cm). **f**, DMF-GPC profiles before/after thermal
processing. M_w s of pristine and injection-moulded **SUPERBIO** were 92.2, and 89.7 kg mol⁻¹, respectively.

**Fig. 4. The recycling of the SUPERBIO electrode. g**, Photograph (inset) and FE-SEM image of AgNW-
coated **SUPERBIO** film. **h**, Electrode dissolved in DMAc was filtered using a Nylon syringe filter. Then,
the filtered solution was casted into a transparent freestanding film. **i**, Photograph (inset) and FE-SEM image
of the resultant recycled **SUPERBIO** film.

**(Page 18 Line 334)** For recycling the electrode, the **SUPERBIO** electrode (1 g) was dissolved in DMAc (9
90 g). The solution was then filtered by a Nylon syringe-filter with a pore size of 0.45 μm to separate AgNWs
and successfully solvent-casted into a transparent free-standing film (Fig. 4g-i).⁶

We have dragged the related contents from the original manuscript **(Page 11 Line 204)**.

The **SUPERBIO** or **BPA-SEP** films (4 g) was chopped and each melted with polyethylene glycol (PEG) of
0.4 g as a plasticiser at 270 °C for 8 min, and then injection-moulded into a rectangular bar (see Methods for
details) (Fig. 2d).

Lastly, we believe that plastics composed of polymers that consist partly of bio-derived monomers, can be
called “sustainable plastics.” For example, Bio-PET is considered a sustainable plastic, even though only a
glycol monomer is replaced by a bio-derived one, and the toxic petrochemical terephthalate remains (*Nature*
**540**, 354–362 (2016)). In addition, we think that it can be termed sustainable plastics from the viewpoint of
eco-toxicology. Particularly, in the field of R&D, researchers are encouraged to replace conventional petro-
based plastics with their more eco-friendly counter parts by various approaches using bio-based materials;
this allows the reduction of considerably environmentally hazardous substances, such as BPA. In the
industry, the biomass content in bio-PET is gradually increasing from 30% to 95%, until 2022
(<https://www.thesourcemagazine.org/new-alliance-develop-100-percent-bio-based-bottles>). Moreover, we
hope that as a step forward, an entirely bio-based SEP will be developed in the near future.

We added the following content in Conclusion **(Page 21 Line 404)**

**As a future research scope and in order to avoid the environmental effects of the petrochemical part of**
**SUPERBIO, an SEP completely derived from biological resources can be developed.**

We provide an image only for revision.

Title: Coca-Cola Plantbottle™ : The image from the presentation by Dr. Robert Kriegel, SPE Bioplastics
Conference, Seattle, WA. May 17, 2012 (<https://polymerinnovationblog.com/the-coca-cola-plant-bottle-a-step-in-the-right-direction/>).

2. Lines 52-55: the term Super Engineering Plastic (SEP) is widely used in the manuscript as well as in the
title: a clear and comprehensive definition of SEP should be provided to univocally identify the class of
materials (refs. 2 and 3 are not enough).

**Answer:** As suggested by the reviewer, we have strengthened the introduction part as follows, and added the
following references.

**(Page 3 Line 51)** According to superiority of thermal and mechanical performances, thermoplastics are
generally classified in the following order: commodity plastics < engineering plastics (EPs) < super
engineering plastics (SEPs). There is no appropriate quantitative standard for the precise classification
because most physical properties of thermoplastics exist across all the above-mentioned three classes.^{7,8} In
polymer science, glass transition temperature (T_g) is a general indicator to represent thermomechanical
characteristics of polymers. In the same order, the three classes of thermoplastics typically have the T_g
ranges of <100, 100–150, and >150 °C.^{1,7–10} SEPs, also known as high-performance or specialty
thermoplastics, are gradually replacing thermosets and pseudo-thermoplastics as thermally and mechanically
robust materials for aircrafts, automobiles, electronics, dental devices and in household/children's products
because of their recyclability.^{2,11}

3. Williams, P. T. Valorization of printed circuit boards from waste electrical and electronic equipment by
pyrolysis. *Waste Biomass Valorization* **1**, 107–120 (2010).

4. Hall, W. J. & Williams, P. T. Separation and recovery of materials from scrap printed circuit
boards. *Resour. Conserv. Recy.* **51**, 691–709 (2007).

5. Kim, J. W., Lee, A. S., Yu, S. & Han, J. W. En masse pyrolysis of flexible printed circuit board wastes
quantitatively yielding environmental resources. *J. Hazard. Mater.* **342**, 51–57 (2018).

6. Zou, Z., Zhu, C., Li, Y., Lei, X., Zhang, W. & Xiao, J. Rehealable, fully recyclable, and malleable
electronic skin enabled by dynamic covalent thermoset nanocomposite. *Sci. Adv.* **4**, eaaq0508 (2018).

7. Platt, D. K. *Engineering and high performance plastics market report: a Rapra market report.*
(iSmithers Rapra Publishing, 2003).

8. Gauthier, M. M. & Handbook Committee ASM International, *Engineered materials handbook, desk*

- *edition* (ASM International, 1995).
- 9. ULLMANN'S editorial team, *Ullmann's polymers and plastics: products and processes, 4 volume set*
(Wiley-VCH, 2016).
- 10. Tremblay, J.-F. A new engineering plastic from china. *Chem. Eng. News* **88**, 28–30 (2010).
- 11. Maekawa, M., Kanno, Z., Wada, T., Hongo, T., Doi, H., Hanawa, T., Ono, T. & Uo., M. Mechanical
properties of orthodontic wires made of super engineering plastic. *Dent. Mater. J.* **34**, 114–119 (2015).

3. Line 70: the statement "mainly because of the low reactivity of these monomers" is quite generic and
should be better specified. In the case of polycondensation, diffusion-limitations are actually hindering the
chain growth, thus limiting the molecular weight. On the other hand, when talking about ROP, the intrinsic
reactivity of endgroups matters, even though the choice of a suitable catalyst can affect the reaction rate at
large extent. Please, rephrase and better specify.

**Answer:** As suggested by the reviewer, we have deleted the sentence: "mainly because of the low reactivity
of these monomers," and added the following sentence.

**(Page 5 Line 91)** Condensation polymer for **thermoplastics** from bio-derived cyclic compounds have a
relatively **low molecular weight** of $<50 \text{ kg mol}^{-1}$, even though they could achieve a T_g as high as SEPs due
to their rigid cyclic structure (Supplementary Table 1).⁵⁵⁻⁵⁸ **The high melt viscosity of the bio-derived cyclic-**
**compound-based polymers causes diffusion limitations, which actually hinder the chain growth.**⁴⁵ As a result
of their **low molecular weight**, most of these bio-based polymers **with a high T_g of $>150 \text{ }^\circ\text{C}$** have poor or
unknown mechanical properties, let alone practical applications. To the best of our knowledge, there are few
studies investigating their mechanical properties as well as melt processability.^{29,39}

4. Line 121: the mechanical properties of the polymer are indeed quite remarkable. However, the elongation
 at break of about 8% (the same value for PET is >300%) should be mentioned and commented.

**Answer:** As suggested by the reviewer, we have modified the following sentence.

(Page 10 Line 183) The **SUPERBIO** film ($M_w = 114 \text{ kg mol}^{-1}$) exhibited superior tensile, tear, and impact
 strengths compared to **BPA-SEP** (Fig. 2b–d and Supplementary Fig. 5). **SUPERBIO**'s tensile Young's
 modulus (3.7 GPa), ultimate tensile strength (UTS, 78 MPa), tensile toughness (5.6 MJ m^{-3}), **tensile**
 **elongation (7.9 %)**, and tear strength (160 kN m^{-1}) were 1.2, 1.5, 1.8, 0.9, and 1.2 times those of **BPA-SEP**,
 respectively.

In general, robustness and ductility are trade-off factors. SEP materials have a higher modulus, while the
 elongation is lower than those of other plastics, i.e., PET.

We provide an image only for revision.

(<http://iwww.plasticsportal.com/products/datasheet.html?type=iso¶m=Ultrason+S+3010>)

Product Information		Ultrason® S 3010 Polysulfone (PSU)		BASF We create chemistry	
Mar 2019				Email PDF Datasheet Print/Save Version	
Product Description Ultrason S 3010 is medium viscosity injection molding grade with improved toughness and chemical resistance (stress crack resistance).					
Applications Typical applications include laboratory accessories and household parts.					
PHYSICAL					
Density, g/cm ³	ISO Test Method	1183	Property Value	1.23	
Mold Shrinkage, parallel, %	294-4		0.7		
Mold Shrinkage, normal, %	294-4		0.74		
Moisture, % (50% RH)	62		0.3		
(Saturation)			0.8		
RHEOLOGICAL					
Melt Volume Rate (MVR) (230°C, 10 min, 10 mm)	1113		40		
MECHANICAL					
Tensile Modulus, MPa	527		2,550		
Tensile stress at yield, MPa 23°C	527		75		
Tensile strain at yield, % 23°C	527		6		
Ball Indentation, MPa	2039-1		135		
IMPACT					
Izod Notched Impact, kJ/m ² -30°C	180		6		
23°C			5.5		
Charpy Notched, kJ/m ² -30°C	179		6		
23°C			5.5		
Charpy Unnotched, kJ/m ² -30°C	179		N		
23°C			N		
THERMAL					
HDT A, °C	ISO Test Method	75	Property Value	177	
Coef. of Linear Thermal Expansion, Parallel, mm/mm °C			0.53 X10 ⁻⁴		
ELECTRICAL					
Comparative Tracking Index	IEC 60112		120		
Volume Resistivity (Ohm-cm)	IEC 60093		>1E13		
Surface Resistivity (Ohm)	IEC 60093		>1E15		
Dielectric Constant (100 Hz)	IEC 60250		3.1		
Dissipation Factor (1 MHz)	IEC 60250		1.1		

5. Lines 124-125: a polymer called BPA-SEP was prepared as control material. Why higher MW was
 considered? (more similar MW of the control and the actual material would make the comparison more
 meaningful) Which was the conversion of the two monomers?
 6. Lines 152-153: the different MW of the two polymers is again affecting the reliability of this comparison.
 Once more, the preparation of polymers with much more similar MW would be very desirable.

**Answer to both Q5 & Q6:** As suggested by the reviewer, we have synthesized **BPA-SEP** with a similar
 molecular weight of $110 \text{ kg} \cdot \text{mol}^{-1}$. Its tensile mechanical properties have been investigated. As expected, the
 mechanical performances of **BPA-SEP** ($110 \text{ kg} \cdot \text{mol}^{-1}$) are lower than those of **SUPERBIO** as well as those
 of **BPA-SEP** of $151 \text{ kg} \cdot \text{mol}^{-1}$.

It would be preferable not to change the comparison group because excessive experiments (*in vivo* tests,
 electronics assembly, etc.) for **BPA-SEP** of $151 \text{ kg} \cdot \text{mol}^{-1}$ have already been performed. We find the
 comparison between the original experimental and control groups logical. The relatively low molecular
 weight of **SUPERBIO** ($114 \text{ kg} \cdot \text{mol}^{-1}$) afforded mechanical properties higher than those of **BPA-SEP** (151
 $\text{kg} \cdot \text{mol}^{-1}$). Remarkably, the higher mechanical property of **SUPERBIO** is not due to the molecular weight,
 but to the molecular structure.

We provide the tensile stress-strain curves ($n = 5$) and data statistics for **BPA-SEP** of $110 \text{ kg} \cdot \text{mol}^{-1}$. However,
 this is only for revisions; we did not update this data in the manuscript.

b

No.	Young's Modulus (GPa)	UTS (MPa)	Elongation at break (%)	Toughness (MJ/m ³)
1	2.5	55.8	2.90	1.55
2	2.2	47.4	2.91	1.31
3	2.3	51.4	2.16	1.06
4	2.8	59.2	2.51	1.41
5	2.4	56.9	3.93	2.09
Mean	2.5	53.1	2.88	1.48
Standard error	0.13	2.12	0.30	0.17

We have dragged the tensile mechanical properties of **SUPERBIO** ($114 \text{ kg} \cdot \text{mol}^{-1}$) and **BPA-SEP** (151 kg
 210 mol^{-1}) from the Supplementary Information.

SUPERBIO vs. BPA-SEP

**Supplementary Figure 5.** Tensile properties of **SUPERBIO** (n = 10) and **BPA-SEP** (n = 8).

7. Line 165: which plasticizer was actually used? Please, specify and motivate its choice.

**Answer:** PEG, with a molecular weight of $400 \text{ g} \cdot \text{mol}^{-1}$, was used as a plasticizer. PEG is generally
considered biologically inert and safe (Veronese, F. M. *Biomaterials*, 2001, 22, 405-417.). Bio-derived PEG
has recently become mass producible (see [http://www.bio-sourced.com/biobased-polyethylene-glycol-peg-](http://www.bio-sourced.com/biobased-polyethylene-glycol-peg-400)
[400](http://www.bio-sourced.com/biobased-polyethylene-glycol-peg-400) and <https://www.acme-hardesty.com/product/polyethylene-glycol-peg-200/>). The addition of PEG
barely damages the “sustainability” and “renewability” character of **SUPERBIO**.

**In the results and discussion section (Page 11 Line 204)**

The **SUPERBIO** or **BPA-SEP** films (4 g) was chopped and each melted with **polyethylene glycol (PEG)** of
**0.4 g** as a plasticiser at **270 °C** for **8 min**, and then injection-moulded into a rectangular bar (see Methods for
details) (Fig. 2d).

**In the method section (Page 23 Line 445)**

**PEG that has a molecular weight of $400 \text{ g} \cdot \text{mol}^{-1}$ (St. Louis, MO, USA) and.....**

8. Lines 166-168: the less coloring without antioxidant is a good result. But what about the MW after the
melting? MW values before and after should be provided to support the good behavior during melting.

**Answer:** As suggested by the reviewer, we have measured the molecular weight after the injection-molding.
As shown in Fig. 2e and f, the molecular weight was not changed. In addition, we have monitored the M_w
change of **SUPERBIO** during 5 programmed cycles of heat treatments (Supplementary Fig. 8). Each cycle
consists of heating (30 to 270 °C) and cooling (270 to 30 °C). M_w hardly changed until the second heat
treatment. The M_w of **SUPERBIO** decreased by only 9% after the fifth heat treatment. The results confirmed
its thermal stability at the melt state as well as the recyclability.

We have updated Fig. 2 as follows.

**Fig. 2. Damage-tolerant bio-based super engineering plastic.** a, Photographs of the solution-casted
pristine, origami-folded (top), and unfolded (bottom) films of **SUPERBIO** (scale bar: 1 cm, Supplementary
Movie 1). b, Tensile stress-strain curves of **SUPERBIO** (n = 10) and **BPA-SEP** (n = 8) films. c, (Left)
Original and (right) differential tear load-distance curves of **SUPERBIO** and **BPA-SEP** films. Inset is
photograph of the specimen for tear test (KS M ISO 34-1:2014, scale bar: 1 cm). d, Impact strength of the

injection-moulded **SUPERBIO** and **BPA-SEP**. Inset is photograph of the rectangular bar-shaped specimens
for the impact test (scale bar: 1 cm). Each impact strength value represents the mean and standard error of
triplicate samples. **e**, Recycling of **SUPERBIO** products (scale bar: 1 cm), and **f**, DMF-GPC profiles
before/after thermal processing. M_w s of pristine and injection-moulded **SUPERBIO** were 92.2, and 89.7 kg
255 mol^{-1} , respectively.

**(Page 11 Line 211)** To evaluate the thermal stability of **SUPERBIO** in detail during melt processing, we
have monitored the M_w change of **SUPERBIO** during five programmed cycles of heat treatments
(Supplementary Fig. 8). Each cycle consists of heating (30 to 270 °C) and cooling (270 to 30 °C) with a
ramp rate of 10 °C min^{-1} under a nitrogen atmosphere. The M_w hardly changed until the second heat
treatment. The M_w of **SUPERBIO** decreased to only 9% after the fifth heat treatment. This suggests that
**SUPERBIO** can be recycled through a series of melting and moulding.⁶³

We have updated the Supplementary Figure (Page 16).

**Supplementary Figure 8.** The M_w changes of **SUPERBIO** within programmed 5 cycles of heat treatments.
(a) One cycle of heat treatments consists of heating (30 to 270 °C) and cooling (270 to 30 °C) with a rate of
10 °C min^{-1} under nitrogen atmosphere. (b) The M_w of **SUPERBIO** depending on the number of heat
treatments.

9. Line 297: is more than 80% viability of the negative control good enough to claim bio-compatibility?
Values for the same cell line and different polymers should be provided (from the literature or own
experiments).

**Answer:** As suggested by the reviewer, we have updated the References to be more convincing to readers.

As suggested by the reviewer, the literature A (#67) utilized the same cell line (L929) and different polymers.

280 A. *Materials Science and Engineering C* 2015, 46, 202–206. doi: 10.1016/j.msec.2014.08.038

1. Introduction

Due to the character of biodegradability, magnesium alloys have attracted interest as biodegradable coronary stents or bone implants for recent years [1–12]. Coronary stents and bone implants belong to Class III Medical Devices, of which good cytocompatibility and good hemocompatibility are essential requirements. According to ISO 10993-5 [13], ISO 10993-4 [14] and ASTM F 756-00 [15], the cytotoxicity should be Grade 0 or Grade 1 (cell viability >80%), and the mean hemolysis from the replicate test samples should be less than 5%. However, among the novel developed magnesium alloys, many of them cause more death of the cells [16–18] and severe hemolysis which are much higher than 30% [8,19,20] in *in-vitro* tests.

Many factors such as bacterial toxins, pH and metabolic changes induced by temperature can compromise the erythrocyte membrane and kill the cells. As to the magnesium alloys, the main possible factors for

CCK-8 assay of murine fibroblast cell line (L929) were used to test the cell viability in different Mg^{2+} concentration and pH values of DMEM system following the ISO 10993-5 standard [13]. An extract of the alloys was prepared according to ISO 10993-12 [22], in which magnesium samples were immersed in the culture medium with serum at $37\text{ }^{\circ}\text{C} \pm 1\text{ }^{\circ}\text{C}$ for 24 h in humidified 5% CO_2 atmosphere in an incubator. The ratio of the surface area of the magnesium sample to the volume of the medium was $1.25\text{ cm}^2/\text{mL}$. Cells were seeded in 96-well culture plates at a density of 5000 cells/well, and incubated at $37\text{ }^{\circ}\text{C}$ in

B. *J. Funct. Biomater.* 2014, 5, 43-57. doi:10.3390/jfb5020043

2. Results and Discussion

2.1. Results

The extent of cytotoxicity from every single concentration of antibacterial agent was quantified as a percentage of cell viability including the absorbance values obtained to each system (Figure 3A–F). Pursuant to ISO 10993-5, percentages of cell viability above 80% are considered as non-cytotoxicity; within 80%–60% weak; 60%–40% moderate and below 40% strong cytotoxicity respectively [32]. It

We have updated the references in the manuscript as follows.

67. Zhen, Z., Liu, X., Huang, T., Xi, T. & Zheng, Y. Hemolysis and cytotoxicity mechanisms of
biodegradable magnesium and its alloys. *Mat. Sci. Eng. C* **46**, 202–206 (2015).

68. López-García, J., Lehocký, M., Humpolíček, P. & Sába, P. HaCaT keratinocytes response on
antimicrobial atelocollagen substrates: extent of cytotoxicity, cell viability and proliferation. *J. Funct.*
*Biomater.* **5**, 43–57 (2014).

10. Line 328: the polymer is claimed to be superior when compared to "unrecyclable thermosets and
pseudo-thermoplastics". But what about its recyclability? This aspect deserves deeper investigation and at
least comments about possible recycling (by chemical and/or mechanical methods) should be discussed.
This is indeed a crucial aspect for its successful large scale production.

**Answer:** As suggested by the reviewer, we have conducted two types of recycling approaches. First, as
shown in Fig. 2e,f, the films and molded products were melted and injected into new products. Second, as
shown in Fig. 4g–i, the **SUPERBIO**-based electrode was dissolved in a solvent. The solution was passed
through a filter, and then casted into a transparent freestanding film through solvent evaporation. Thus,
**SUPERBIO** is physically recyclable. These physical recycling processes are more beneficial than the
chemical-based processes are. For example, among plastics, thermosets (e.g., epoxy) and pseudo-
thermoplastics (e.g., polyimide) generally have higher thermal stability and are, thus, preferred over
thermoplastics as electronic parts, e.g., printed circuit boards (PCB). After curing, thermosets and pseudo-
thermoplastics are not meltable and dissolvable; the separation of metal from PCB requires harsh chemical
degradation or pyrolysis of the plastics, and the recycling of the plastics is difficult. If thermosets are
replaced by thermally durable thermoplastics such as **SUPERBIO**, for electronic parts, the recycling of both
plastics and metals from PCB can be effective.

We have strengthened the Introduction Section **(Page 3 Line 42)**.

Since plastics have become indispensable in our life, their consumption has exponentially increased.¹ The
colossal demand for plastics has led to a large amount of wastes. For example, abandoned electronics
notably create printed circuit board (PCB) waste. The typical content of metals, plastics, and ceramics in
PCBs is approximately 40, 30, and 30 wt%, respectively.^{2,3} Among plastics, thermosets, e.g. epoxy, and
pseudo-thermoplastics, e.g. polyimide, generally have higher thermal stability; thus, they are more preferred
to thermoplastics as materials for PCB.^{4,5} After curing, thermosets and pseudo-thermoplastics do not melt
and dissolve; the separation of metal from PCB requires harsh chemical degradation or pyrolysis of plastics,
and the recycling of plastics is difficult.^{4,5} If electronic parts are made of thermally durable thermoplastics,
both plastics and metals from PCB can be effectively recycled by melting or dissolution.⁶ Likewise,
substituting thermosets with thermoplastics for many other applications increases the recycling rate of
plastic wastes.

**(Page 18 Line 334)** For recycling the electrode, the **SUPERBIO** electrode (1 g) was dissolved in DMAc (9

325 g). The solution was then filtered by a Nylon syringe-filter with a pore size of 0.45 μm to separate AgNWs
and successfully solvent-casted into a transparent free-standing film (Fig. 4g-i).⁶

We have added the following content (Fig. 2e,f & Fig. 4g-i).

**Fig. 2. e**, Recycling of SUPERBIO products (scale bar: 1 cm). **f**, DMF-GPC profiles before/after thermal
processing. M_w s of pristine and injection-moulded SUPERBIO were 92.2, and 89.7 kg mol^{-1} , respectively.

**Fig. 4. The recycling of the SUPERBIO electrode. g**, Photograph (inset) and FE-SEM image of AgNW-
coated SUPERBIO film. **h**, Electrode dissolved in DMAc was filtered using a Nylon syringe filter. Then,
the filtered solution was casted into a transparent freestanding film. **i**, Photograph (inset) and FE-SEM image
of the resultant recycled SUPERBIO film.

We have dragged the related contents from the original manuscript (Page 11 Line 204).

The SUPERBIO or BPA-SEP films (4 g) was chopped and each melted with polyethylene glycol (PEG) of
0.4 g as a plasticiser at 270 $^{\circ}\text{C}$ for 8 min, and then injection-moulded into a rectangular bar (see Methods for
details) (Fig. 2d).

*Minor comments*

*11. Lines 62-65: the contributions of Gandini (e.g.*

<https://onlinelibrary.wiley.com/doi/pdf/10.1002/pola.24812>, FDCA-isosorbide polyesters) and De Jong (e.g.

<https://pubs.acs.org/doi/10.1021/bk-2012-1105.ch001>, PEF) should be also mentioned.

**Answer:** As suggested by the reviewer, we have updated both references.

**References**

25. Gomes, M., Gandini, A., Silvestre, A. J. D. & Reis, B. Synthesis and characterization of poly(2,5-furan
dicarboxylate)s based on a variety of diols. *J. Polym. Sci. Part A Polym. Chem.* **49**, 3759–3768 (2011).

26. De Jong, E., Dam, M. A., Sipos, L. & Gruter, G.-J. M. Furandicarboxylic acid (FDCA), a versatile
building block for a very interesting class of polyesters. *ACS Symp. Ser.* **1105**, 1–13 (2012). (ed. Smith,
P.)

*12. Line 69: low-molecular-weight should become low molecular weight.*

As suggested by the reviewer, we have corrected the following: “low molecular weight,” throughout the
manuscript.

**(Page 2, Line 26)** However, previously reported bio-derived thermosets or thermoplastics rarely offer
thermal/mechanical properties, scalability, or recycling that match those of petrochemical SEPs.

**(Page 5 Line 91)** Condensation polymer for thermoplastics from bio-derived cyclic compounds have a
relatively low molecular weight of $<50 \text{ kg mol}^{-1}$, even though they could achieve a T_g as high as SEPs due
to their rigid cyclic structure (Supplementary Table 1).^{55–58} The high melt viscosity of the bio-derived cyclic-
compound-based polymers causes diffusion limitations, which actually hinder the chain growth.⁴⁵ As a result
of their low molecular weight, most of these bio-based polymers with a high T_g of $>150 \text{ }^\circ\text{C}$ have poor or
unknown mechanical properties, let alone practical applications.

13. Line 268: SEPERBIO  SUPERBIO (please check everywhere in the text)

**Answer:** As suggested by the reviewer, we have corrected “SUPERBIO.”

**(Page 15 Line 280-287)** The elongation of the repeating unit in **SUPERBIO** has a k value 1.57 fold higher
than that of **BPA-SEP**. However, the k data cannot totally reflect the bulk mechanical properties. The
Young’s modulus and UTS are affected to a high degree by the molecular slipping and noncovalent failures,
as well as macroscopically defective morphologies that occur during processing. Nevertheless, it is manifest
that the structure restraint of **SUPERBIO**-single molecule by ISB playing an important role in the high
mechanical properties.

14. Figure 5a: please correct "pristine" on the horizontal axis.

**Answer:** As suggested by the reviewer, we have corrected “pristine.”

We have revised the letter in Fig. 5a.

15. Line 319: ...this SEP, was achieved...

**Answer:** As suggested by the reviewer, we have corrected the sentence.

**(Page 21 Line 393)** The high-molecular-weight of this SEP was achieved with the aid of 18-crown-6 to
activate S_NAr polymerisation.

16. References: all author names should be provided.

**Answer:** As suggested by the reviewer, we have corrected the references as follows.

**In references**

16. Jedeon, K., De la Dure-Molla, M., Brookes, S. J., Loidice, S., Marciano, C., Kirkham, J., Canivenc-
Lavier, M.-C., Boudalia, S., Bergès, R., Harada., H., Berdal, A. & Babajko, S. Enamel defects reflect
perinatal exposure to bisphenol A. *Am. J. Pathol.* **183**, 108–118 (2013).

20. Vilela, C. Sousa, A. F., Fonseca, A. C., Serra, A. C., Coelho, J. F., Freire, C. S., & Silvestre, A. J. The
quest for sustainable polyesters–insights into the future. *Polym. Chem.* **5**, 3119–3141 (2014).

21. Feng, L., Zhu, W., Zhou, W., Li, C., Zhang, D., Xiao, Y., & Zheng, L. A designed synthetic strategy
toward poly(isosorbide terephthalate) copolymers: a combination of temporary modification,
transesterification, cyclization and polycondensation. *Polym. Chem.* **6**, 7470–7479 (2015).

30. Japu, C., de Ilarduya, A. M., Alla, A., García-Martín, M. G., Galbis, J. A., & Muñoz-Guerra, S. Bio-
based PBT copolyesters derived from D-glucose: influence of composition on properties. *Polym. Chem.*
**5**, 3190–3202 (2014).

45. Park, S.-A., Choi, J., Ju, S., Jegal, J., Lee, K. M., Hwang, S. Y., Oh, D. X., & Park, J. Copolycarbonates
of bio-based rigid isosorbide and flexible 1,4-cyclohexanedimethanol: Merits over bisphenol-A based
polycarbonates. *Polymer* **116**, 153–159 (2017).

73. Kim, S.-M., Jeon, H., Shin, S.-H., Park, S.-A., Jegal, J., Hwang, S. Y., Oh, D. X., & Park, J. Superior
toughness and fast self-healing at room temperature engineered by transparent elastomers. *Adv. Mater.*
**30**, 1705145 (2018).

75. Nam, S., Song, M., Kim, D. H., Cho, B., Lee, H. M., Kwon, J. D., Park, S.-G., Nam, K.-S., Jeong, Y.,
Kwon, S.-H., Park, Y. C., Jin, S.-H., Kang, J.-W., Jo, S., Kim, C. S. Ultrasooth, extremely deformable
and shape recoverable Ag nanowire embedded transparent electrode. *Sci. Rep.* **4**, 4788 (2014).

76. Jeon, W. S. Park, T. J., Kim, S. Y., Pode, R., Jang, J., & Kwon, J. H. Ideal host and guest system in
phosphorescent OLEDs. *Org. Electron.* **10**, 240–246 (2009).

*Reviewer #2 (Remarks to the Author):*

*The manuscript entitled "Sustainable and Recyclable Super Engineering Thermoplastic from Biorenewable*
*Monomer" is an interesting report of a thermoplastic derived from isosorbide. The authors details the*
*synthesis, characterization, application, and simulations of the material in this report. While the material*
*and properties are of interest and impressive, there are several revisions this reviewer suggests.*

**Answer:** We greatly appreciate the constructive, very detailed comments and suggestions of this reviewer.

*1) The polymer contains an isosorbide monomer but is not really green or sustainable in the synthesis, in*
*particular because it is copolymerized with bisphenol S (a majority of the mass fraction). After degradation,*
*the bisphenol derivative will still be released into the environment and will cause the same severe toxic*
*effects that are well known and studied. Thus, this is not a sustainable polymer and that aspect of the*
*research should significantly be downplayed.*

**Answer:** We agree with the reviewer's opinion. It is difficult to entirely ignore the environmental effects of
DFPS (the sulfur-containing co-monomer: 4,4'-difluorodiphenyl sulfone). Herein, we demonstrate our
claims objectively, including the preparation of non-sulfur polymers, organization of partially bio-based
polymers reported, and recycling of the prepared polymers. In addition, we discussed the limitation of this
material in the Conclusion Section.

Firstly, to address the environmental issue of this co-monomer, **an ISB-based ketone-type poly(arylene**
**ether) (PAE) was synthesized using a sulfur-free monomer** (4,4'-difluorobenzophenone) instead of DFPS.
Furthermore, we showed that the mechanical properties of the ISB-based ketone-type PAE are as high as
those of **SUPERBIO** that is made of DFPS.

We have added the following content:

**(Page 8 Line 147)** Instead of DFPS, a sulphur-free co-monomer is applicable to this polymer system. An
ISB-based ketone-type PAE with a similar M_w (93.6 kg mol^{-1}) called **SUPERBIO-K** was synthesized with a

monomer combination of ISB and 4,4'-difluorobenzophenone by a method identical to DFPS synthesis (see
Method Section & Supplementary Fig. 3).

**(Page 10 Line 190) SUPERBIO-K** achieved a Young's modulus of 3.8 GPa, a UTS of 76 MPa, a tensile
toughness of 8.7 MJ m^{-3} , and an elongation at break of 13% (Supplementary Fig. 3). The tensile
performances of **SUPERBIO-K** are as high as those of **SUPERBIO**.

**(Method section)Synthesis of an ISB-based ketone-type SUPERBIO.**

The same synthetic procedures of **SUPERBIO** were conducted, except that 4,4'-difluorobenzophenone (4.57
460 g, 20.5 mmol) was used instead of DFPS. The final polymer product (6.46 g, 97%), M_w : $93,600 \text{ g mol}^{-1}$, PDI:
1.98, $^1\text{H NMR}$ (CDCl_3 , 300 MHz, ppm): δ 7.82–7.78, 7.06–7.00, 5.09–5.06, 4.96–4.89, 4.72–4.70. 4.27–
4.09.

**(Supplementary Information, Page 10)**

**Supplementary Figure 3.** (a) Synthetic route of an ISB-based ketone-type PAE called **SUPERBIO-K** (M_n
= 44.4 kg mol^{-1} , M_w = 93.6 kg mol^{-1} , PDI = 1.98). (b) Tensile stress-strain curves, (c) data statistics (n = 3),
and (d) $^1\text{H-NMR}$ spectrum of **SUPERBIO-K**.

Secondly, we would like to demonstrate **the relative superiority of our polymers in terms of**
 **“sustainability” over previously reported high T_g biopolymers.** As the reviewer can see in the
 Supplementary Table 1 and in a review paper about high T_g biopolymers (J. Mater. Chem. A, 2018, 6, 9298-
 9331; DOI: 10.1039/C8TA00377G), most high T_g biopolymers comprise majorly petrochemicals.
 Nevertheless, most of them have poor or unknown mechanical properties (making them unsuitable for
 practical applications), and their processabilities are mostly unknown. In this study, the ISB-incorporated
 PAE, i.e., **SUPERBIO** achieved not only a high T_g , but also great mechanical properties and processability.
 We have dragged the following contents from the supplementary information of the original manuscript.

Supplementary Table 1. Characteristics of SUPERBIO, BPA-SEP, reported bio-based high T_g (>150 °C) thermoplastics, commercial petrochemical SEP, and commercial bioplastics (Continue).

Polymer type ^a	Biomass origin ^a	Polymerization ^a	Crystallinity ^a	M_n^b (kg mol ⁻¹)	η_{inh}^b (dl g ⁻¹)	T_g^b (°C)	T_g/T_{d10}^b (°C)	Biomass content ^b (wt%)	E (GPa)	UTS (MPa)	Melt processing/ recycling ^c	Scale ^k (g)	Reference ^d
Polyester ^e	Lignin ^e	Condensation ^e	Semicrostalline ^e	48 ^b	~	157 ^e	>305 ^e	>99 ^e	11 ^e	63 ^e	O ^e	~8 ^e	39 ^e
			Amorphous- (not melting) ^e	~	0.83 ^e	168 ^e	~431 ^e	>99 ^e				~2 ^e	S9 ^e
	Terpene ^e	Ring-opening ^e	2-14 ^b	~	155-243 ^e	~330 ^e	<51 ^e	~0.5 ^e				33 ^e	
Polycarbonate ^e	Sugar (isohexide) ^e	Condensation ^e	Amorphous ^e	<15 ^b	~	<184 ^e	~	<66 ^e	~	~	X ^e	~0.4 ^e	34 ^e
	Sugar ^e (mannose)	Ring-opening ^e		>14 ^b	0.37-0.80 ^e	156-175 ^e	~	31-84 ^e				~1 ^e	23,S10 ^e
Polylactide ^e	Lignin ^e	Condensation ^e	Amorphous ^e	16 ^b	~	152 ^e	~220-240 ^e	73 ^e	~	~	X ^e	~0.3 ^e	S11 ^e
Polyvinylacetate ^e		Post- modification ^e		36-39 ^b	~	152-159 ^e	265-307 ^e	67 ^e				~1 ^e	36,S12 ^e
Polyurethane ^e	Sugar (isohexide) ^e	Condensation ^e	Amorphous ^e	53 ^b	~	157 ^e	224 ^e	46 ^e	~	~	X ^e	~2 ^e	38 ^e
Polyketone ^e	Terpene ^e	Radical addition ^e		8-19 ^b	0.23-0.35 ^e	152-191 ^e	>271 ^e	40 ^e				~5 ^e	24,S13,S14 ^e S15 ^e
Polydiene ^e	Lignin ^e	Metathesis ^e	Amorphous ^e	~100 ^b	~	162 ^e	~327 ^e	>90 ^e	~	~	X ^e	~1 ^e	35 ^e
Polyacrylate ^e		Radical addition ^e		49 ^b	~	156 ^e	380 ^e	82 ^e				~0.1 ^e	40 ^e
Poly(N-aromatic imide) ^e	Itaconic acid ^e	Radical addition ^e	Amorphous ^e	18-50 ^b	~	154-205 ^e	302 ^e	~68 ^e	~	~	X ^e	~	41,S16 ^e
Poly(hydroxylacrylamide) ^e	Lactone ^e			>19 ^b	~	153-238 ^e	>230 ^e	28-51 ^e				~1 ^e	42,S17,S18 ^e S19 ^e
Poly(methylene-lactide) ^e	Tulipalin A ^e	Radical addition ^e	Amorphous ^e	40-100 ^b	~	157-178 ^e	~	50-54 ^e	~	~	X ^e	~2 ^e	43 ^e
Poly(methylene-ester) ^e				~	76-358 ^b	~	213-254 ^e	>305 ^e				>99 ^e	~0.3 ^e
Poly(methylene-ester) ^e	~	~	~	~600 ^b	~	163-189 ^e	>320 ^e	63-72 ^e	~	~	X ^e	~0.6 ^e	S21 ^e

Supplementary Table 1. Characteristics of SUPERBIO, BPA-SEP, reported bio-based high T_g (>150 °C) thermoplastics, commercial petrochemical SEP, and commercial bioplastics (Continue).

Polymer type ^a	Biomass origin ^a	Polymerization ^a	Crystallinity ^a	M_n^b (kg mol ⁻¹)	η_{inh}^b (dl g ⁻¹)	T_g^b (°C)	T_g/T_{d10}^b (°C)	Biomass content ^b (wt%)	E (GPa)	UTS (MPa)	Melt processing/ recycling ^c	Scale ^k (g)	Reference ^d		
SUPERBIO ^e (Poly(arylene ether)) ^e	Sugar (isohexide) ^e	Condensation ^e	Amorphous ^e	114 ^b	0.83 ^e	212 ^e	411/422 ^e	40-44 ^e	3.4 ^e	77 ^e	O ^e	~1,000 ^e	This study ^e		
BPA-SEP ^e (Poly(arylene ether sulfone)) ^e	Petrochemical (bisphenol-A) ^e			151 ^b	1.61 ^e	196 ^e	497/502 ^e	0 ^e	3.0 ^e	51 ^e		~100 ^e			
Poly(arylene ether sulfone) ^e	Petrochemical (commercial PSU) ^e			35 ^b	0.37-0.39 ^e	190 ^e	~	0 ^e	~	~	~	~	~	Sigma Aldrich ^e	
				50-60 ^b	~	187 ^e	~	0 ^e						BASF ^e	
Poly(arylene ether ketone) ^e	Sugar (isohexide) ^e			<6 ^b	0.23-0.65 ^e	213-253 ^e	~	<40 ^e	~	~	~	~	~	~7 ^e	55,S1,S2 ^e
	Sugar ^e (FDCA)			<7 ^b	0.29-0.31 ^e	165-170 ^e	~	44 ^e						~2 ^e	56 ^e
Poly(carbonate-ester) ^e	Sugar (isohexide) ^e			Semicrostalline ^e	Amorphous ^e	~	0.21 ^e	159 ^e	424 ^e	32 ^e	~	~	X ^e	~2 ^e	57 ^e
				>7 ^b		0.31-0.59 ^e	154-161 ^e	>458-470 ^e	32-42 ^e	~1 ^e				58 ^e	
Polyester ^e	Sugar (isohexide) ^e			32-43 ^b	0.47-0.56 ^e	167-193 ^e	>350 ^e	61-84 ^e	~	~	~	X ^e	~35 ^e	21 ^e	
				>6 ^b	0.13-0.67 ^e	158-209 ^e	>360 ^e	52 ^e					~6 ^e	S3,S4,S5 ^e	
	Sugar (isohexide/FDCA) ^e	>9 ^b	0.11-0.38 ^e	173-196 ^e	>320 ^e	>99 ^e	~9 ^e	S6 ^e							
	Sugar (isohexide/LA) ^e	65-180 ^b	~	150-181 ^e	385/397 ^e	52-66 ^e	~15 ^e	S7,S8 ^e							
	Sugar (glucitol) ^e	57 ^b	~	156 ^e	385/398 ^e	60 ^e	~15 ^e	22 ^e							
Polyester ^e	~	Amorphous ^e	13 ^b	0.32 ^e	154 ^e	~377 ^e	52 ^e	~	~	X ^e	~	30 ^e			

Thirdly, **SUPERBIO** can be recycled through melting and dissolution (Fig. 2 e,f & Fig. 4h-i). It shows
the sustainability value of this polymer. The recycling minimizes the exposure of the monomer to the
environment.

We have added the following contents (Fig. 2e,f & Fig. 4g-i).

**Fig. 2. e**, Recycling of **SUPERBIO** products (scale bar: 1 cm). **f**, DMF-GPC profiles before/after thermal
processing. M_w s of pristine and injection-moulded **SUPERBIO** were 92.2, and 89.7 kg mol⁻¹, respectively.

**Fig. 4. The recycling of the SUPERBIO electrode. g**, Photograph (inset) and FE-SEM image of AgNW-
coated **SUPERBIO** film. **h**, Electrode dissolved in DMAc was filtered using a Nylon syringe filter. Then,
the filtered solution was casted into a transparent freestanding film. **i**, Photograph (inset) and FE-SEM image
of the resultant recycled **SUPERBIO** film.

**(Page 18 Line 334)** For recycling the electrode, the **SUPERBIO** electrode (1 g) was dissolved in DMAc (9
496 g). The solution was then filtered by a Nylon syringe-filter with a pore size of 0.45 μm to separate AgNWs
and successfully solvent-casted into a transparent free-standing film (Fig. 4g-i).⁶

We have dragged the related contents from the original manuscript **(Page 11 Line 204)**.

The **SUPERBIO** or **BPA-SEP** films (4 g) was chopped and each melted with polyethylene glycol (PEG) of
0.4 g as a plasticiser at 270 °C for 8 min, and then injection-moulded into a rectangular bar (see Methods for
details) (Fig. 2d).

Lastly, we believe that plastics composed of polymers that consist partly of bio-derived monomers, can be
called “sustainable plastics.” For example, Bio-PET is considered a sustainable plastic, even though only a
glycol monomer is replaced by a bio-derived one, and the toxic petro-chemical terephthalate remains
(*Nature* **540**, 354–362 (2016)). In addition, we think that it can termed sustainable plastics from the
viewpoint of eco-toxicology. Particularly, in the field of R&D, researchers are encouraged to replace
conventional petro-based plastics with more eco-friendly variants by various approaches using bio-based
materials; this allows the reduction of considerably environmentally hazardous substances, such as BPA. In
industry, the biomass content of bio-PET is gradually increasing from 30% to 95%, until 2022
(<https://www.thesourcemagazine.org/new-alliance-develop-100-percent-bio-based-bottles>). Moreover, we
hope that as a step forward, an entirely bio-based SEP will be developed in the near future.

We added the following content in Conclusion **(Page 21 Line 404)**

**As a future research scope and in order to avoid the environmental effects of the petrochemical part of**
**SUPERBIO, an SEP completely derived from biological resources can be developed.**

We provide an image only for revision.

Title: Coca-Cola Plantbottle™ : The image from the presentation by Dr. Robert Kriegel, SPE Bioplastics
Conference, Seattle, WA. May 17, 2012 (<https://polymerinnovationblog.com/the-coca-cola-plant-bottle-a-step-in-the-right-direction/>).

2) Only one molecular weight is studied of this system. It would be useful to study several molecular weights
to improve overall understanding of the structure-activity relationships.

**Answer:** As suggested by the reviewer, along with the sample having its actual M_w of $114 \text{ kg} \cdot \text{mol}^{-1}$,
**SUPERBIO** samples with three different M_w values of 30, 63, and $85 \text{ kg} \cdot \text{mol}^{-1}$ were synthesized by
controlling the reaction time, and the tensile properties of the four different samples were compared. The
Young modulus and UTS gradually increased with M_w to the aforementioned values, achieved by the
**SUPERBIO** sample with $M_w = 114 \text{ kg} \cdot \text{mol}^{-1}$, because of increasing chain entanglements. The tensile
toughness and elongation at break were the highest at M_w of $85 \text{ kg} \cdot \text{mol}^{-1}$. The increasing strength with M_w
negatively impacts the ductility of **SUPERBIO**.

We have updated the content: **(Page 10 Line 193)**.

We have investigated the effects of M_w on the tensile properties of **SUPERBIO** (Supplementary Fig. 7).
Along with the sample having its actual M_w of $114 \text{ kg} \cdot \text{mol}^{-1}$, **SUPERBIO** samples with three different M_w
values of 30, 63, and $85 \text{ kg} \cdot \text{mol}^{-1}$ were synthesized by controlling the reaction time, and the tensile
properties of the four different samples were compared. The Young's modulus and UTS gradually increased
with M_w to the aforementioned values achieved by the **SUPERBIO** sample with $M_w = 114 \text{ kg} \cdot \text{mol}^{-1}$ because
of increasing chain entanglements. The tensile toughness and elongation at break were the highest at the M_w
of $85 \text{ kg} \cdot \text{mol}^{-1}$. The increasing strength with M_w negatively impacts the ductility of **SUPERBIO**.

**Supplementary Figure 7.** (a) Representative tensile stress-strain curves and (b) data statistics of
**SUPERBIO** with four different M_w values of 30, 63, 85, and 114 kg mol⁻¹.

*3) The English grammar should be carefully proofread and corrected throughout the manuscript.*

**Answer:** As suggested by the reviewer, we have corrected the English grammar again throughout the
manuscript. We appreciate for your re-proofreading.

*4) Information about the ethical treatment of animals and all approved animal protocol information should*
*be included in this manuscript.*

**Answer:** As suggested by the reviewer, we have updated the information about the ethical treatment of
animals.

We have modified the sentences as follows **(Page 20 Line 374).**

The *in vivo* biocompatibility test of **SUPERBIO** was conducted by a contract clinical research organization
[Daegu Gyeongbuk medical innovation foundation (DGMIF)] using a rat subcutaneous model, following the
ISO 10993-6 “the local effects of medical devices after implantation” Annex A standard (Fig. 5b).⁶⁹ The
ethical issue was approved by institutional animal care and use committee (IACUC) (Korea), and the
approval code is DGMIF-18012301-00.

*5) There are many general references the authors include for sustainable polymers but more emphasis*
*should be placed in the referencing on the large body of research in isosorbide polymers.*

**Answer:** As suggested by the reviewer, we have strengthened the introduction part and updated the
references as follows.

**(Page 4, Line 72)** The growing environmental and health concerns have prompted efforts to substitute
toxic petro-based aromatic monomers for plastics^{17–20} by bio-derived cyclic compounds, such as isosorbide
(1,4:3,6-dianhydro-D-glucitol, ISB),^{21–24} 2,5-furandicarboxylic acid,^{25–29} sugar,³⁰ terpene,^{31–35} lignin

derivatives,^{36–41} and others.^{42, 43} ISB, a bicyclic sugar derivative, is an attractive alternative of BPA.^{23,44–46}
The ISB moiety enhances the mechanical, thermal, and optical properties of the host polymer due to its
unique molecular structure.^{47–56} Moreover, the safety of ISB has been demonstrated by its use in
pharmaceuticals and cosmetics. ISB production technology has become mature onto a commercial scale.^{53,54}
A French agricultural company recently have achieved the world's largest high purity isosorbide production
capacity of annual 20,000 tons. Many global chemical companies are interested in the commercialization of
isosorbide-based polymers.

21. Feng, L., Zhu, W., Zhou, W., Li, C., Zhang, D., Xiao, Y., & Zheng, L. A designed synthetic strategy
toward poly(isosorbide terephthalate) copolymers: a combination of temporary modification,
transesterification, cyclization and polycondensation. *Polym. Chem.* **6**, 7470–7479 (2015).

22. Saber, C., M. Weidner, S. M., Fildier, A. & Kricheldorf., H. R. Copolyesters of isosorbide, succinic acid,
and isophthalic acid: biodegradable, high Tg engineering plastics. *J. Polym. Sci. Part A Polym. Chem.*
**51**, 2464–2471 (2013).

23. Chatti, S., Schwarz, G. & Kricheldorf, H. R. Cyclic and noncyclic polycarbonates of isosorbide
(1,4:3,6-dianhydro-D-glucitol). *Macromolecules* **39**, 9064–9070 (2006).

24. Lee, C.-H., Takagi, H., Okamoto, H., Kato, M., & Usuki, A. Synthesis, characterization, and properties
of polyurethanes containing 1,4:3,6-dianhydro-D-sorbitol. *J. Polym. Sci. Part A Polym. Chem.* **47**,
6025–6031 (2009).

47. Koo, J. M., Hwang, S. Y., Yoon, W. J., Lee, Y. G., Kim, S. H., & Im, S. S. Structural and thermal
properties of poly (1,4-cyclohexane dimethylene terephthalate) containing isosorbide. *Polym. Chem.* **6**,
6973–6986 (2015).

48. Liu, D. D. & Chen, E. Y. X. Organocatalysis in biorefining for biomass conversion and upgrading.
*Green Chem.* **16**, 964–981 (2014).

49. Juais, D., Naves, A. F., Li, C., Gross, R. A. & Catalani, L. H. Isosorbide polyesters from enzymatic
catalysis. *Macromolecules* **43**, 10315–10319 (2010).

50. Zenner, M. D., Xia, Y., Chen, J. S. & Kessler, M. R. Polyurethanes from isosorbide-based
diisocyanates. *ChemSusChem* **6**, 1182–1185 (2013).

51. Chrysanthos, M., Galy, J. & Pascault, J.-P. Preparation and properties of bio-based epoxy networks
derived from isosorbide diglycidyl ether. *Polymer* **52**, 3611–3620 (2011).

52. Lee, C.-H., Takagi, H., Okamoto, H. & Kato, M. Improving the mechanical properties of isosorbide
copolycarbonates by varying the ratio of comonomers. *J. Appl. Polym. Sci.* **127**, 530–534 (2012).

- 53. Gallagher, J. J., Hillmyer, M. A. & Reineke, T. M. Acrylic triblock copolymers incorporating isosorbide
for pressure sensitive adhesives. *ACS Sustainable Chem. Eng.* **4**, 3379–3387 (2016).
- 54. Chatti, S., Hani, M. A., Bornhorst, K. & Kricheldorf, H. R. Poly(ether sulfone) of isosorbide,
isomannide and isoidide. *High Perform. Polym.* **21**, 105–118 (2009).
- 55. Abderrazak, H. B., Fildier, A., Romdhane, H. B., Chatti, S. & Kricheldorf, H. R. Synthesis of new poly
(ether ketone) s derived from biobased diols. *Macromol. Chem. Phys.* **214**, 1423–1433 (2013).
- 56. Gallagher, J. J., Hillmyer, M. A. & Reineke, T. M. Isosorbide-based polymethacrylates. *ACS*
*Sustainable Chem. Eng.* **3**, 662–667 (2015).

*Reviewer #3 (Remarks to the Author):*

*This manuscript describes a novel synthetic route to achieve high molecular weight polymers from the*
*biorenewable chemical isosorbide. The resulting polymers have very advantageous thermal and mechanical*
*properties, equivalent to “super engineering thermoplastics”, for which there are few examples in the*
*biorenewable polymers literature. However, I have some concerns about the claims made in the paper, as*
*well as the quality of the discussion and data presentation in the manuscript. These concerns must be*
*addressed before considering this manuscript for publication.*

**Answer:** We greatly appreciate the constructive, very detailed comments and suggestions of this reviewer.

*1. The claims in the introduction and abstract regarding current state of the biorenewable polymers*
*literature are wildly overstated, and this takes credibility away from this study. The authors needs to*
*carefully consider the present literature and then discuss the novelty of their study in this context. For*
*example: a. In the abstract on page 2 line 26: “the research efforts produced only thermosets or low*
*molecular weight thermoplastics”*
*b. On page 3 line 47, “these researches have mostly produced commodity-grade thermoplastics with a glass*
*transition temperature < 100 C....”*
*However, a quick google search has yielded the following study on an isosorbide-based polymethacrylate,*
*which has Tg around 130C (Gallagher, Hillmyer and Reineke, ACS Sustainable Chemistry and Engineering*
*2015, 3, 662). Additionally, the lignin-based polymer in ref 31 (Holmberg, A.L., Reno, K.H., Nguyen, N.A.,*
*Wool, R.P. & Epps III, T.H. Syringyl Methacrylate, a Hardwood Lignin-Based Monomer for High-Tg*
*Polymeric Materials. ACS Macro Lett. 5, 574-578 (2016)) has a reported Tg of 205 C. These two examples*
*suggest that a more thorough literature search could yield other biorenewable polymers, and even*
*isosorbide-derived polymers, with high Tg’s, minimizing the impact of this work. It is therefore essential to*
*place this work in the context of the relevant literature, which likely goes beyond the two articles mentioned*
*here.*

**Answer:**

**First, we have accepted the reviewer’s suggestion and have balanced our arguments by modifying the**
**sentences and the introduction part as follows.**

**(Abstract; Page 2 Line 26)** We have changed “did not offer” to “rarely offer.”

However, previously reported bio-derived thermosets or thermoplastics rarely offer thermal/mechanical
properties, scalability, or recycling that match those of petrochemical SEPs.

**(Page 4 Line 84)** We have changed “not” to “less reported.”

Bio-based high T_g thermoplastics are defined as polymers that i) are entirely or partially derived from bio-
derived feedstocks, ii) have T_g of >150 °C, and iii) are melt processible.¹ However, bio-based high-
performance thermoplastics, i.e. with a high T_g of >150 °C, have been relatively less reported than bio-based
thermosets/pseudo-thermoplastics and have the following limitations.

**Second, we earnestly request that the values of this polymer be reviewed again.** We have already cited
the following **40 references** of bio-based high T_g (>150 °C) thermoplastics in the main text and in the
Supporting Information, including “*ACS Macro Lett.* **5**, 574-578 (2016)” as follows (see reference #41).

We are well aware that bio-based high T_g polymers have been intensively reported. However, as we claimed,
most of these bio-based polymers have poor or unknown mechanical properties, and their molecular weights
were not enough to measure the mechanical properties. There are few studies investigating their mechanical
properties as well as melt/solvent processability. Only a few studies have showed the mechanical properties
and thermal processabilities of bio-based high T_g polymers. This is due majorly to their low molecular
weights or low production scale.

In *ACS Macro Lett.* **5**, 574-578 (2016) and *ACS Macro Lett.* **6**, 802-807 (2017) (ref. #41, #S16), the authors
did not explore the mechanical properties of their polymers. In addition, the molecular weights of the
polymers ranged from 18 to 50 kg·mol⁻¹.

Thus, the novelty of this study is that **SUPERBIO** not only has a T_g of >150 °C but also a high molecular
weight of 100 kg·mol⁻¹. Consequently, **SUPERBIO** exhibits considerably high mechanical performances
that are unprecedented for bio-based thermoplastics, petrochemical SEPs, or thermosets. In addition, it is
quite rare that a bio-based high-performance polymer is recyclable through melting or dissolution.

Please find the references and the Supplementary Table 1.

1. Supplementary Tables, Figures, and Movies

Supplementary Table 1. Characteristics of SUPERBIO, BPA-SEP, reported bio-based high T_g (>150 °C) thermoplastics, commercial petrochemical SEP, and commercial bioplastics (Continue).

Polymer type ^a	Biomass origin ^a	Polymerization ^a	Crystallinity ^a	M_n^b (kg mol ⁻¹) ^a	η_{inh}^b (dL g ⁻¹) ^a	T_g^b (°C) ^a	T_{5}/T_{410}^b (°C) ^a	Biomass content ^d (wt%) ^a	E (GPa) ^a	UTS (MPa) ^a	Melt processing ^c /recycling ^c	Scale ^k (g) ^a	Reference ^a		
SUPERBIO ^a (Poly(arylene ether)) ^a	Sugar (isohexide) ^a	Condensation ^a	Amorphous ^a	114 ^b	0.83 ^a	212 ^a	411/422 ^a	40-44 ^a	3.4 ^a	77 ^a	O ^a	~1,000 ^a	This study ^a		
BPA-SEP ^a (Poly(arylene ether sulfone)) ^a	Petrochemical (bisphenol-A) ^a			151 ^b	1.61 ^a	196 ^a	497/502 ^a	β ^a	3.0 ^a	51 ^a	~100 ^a				
Poly(arylene ether sulfone) ^a	Petrochemical (commercial PSU) ^a			35 ^b	0.37-0.39 ^a	190 ^a	-- ^a	0 ^a	-- ^a	-- ^a	-- ^a	-- ^a	↓ ^a	Sigma Aldrich ^a	
				50-60 ^b	-- ^a	187 ^a		0 ^a					BASF ^a		
Poly(arylene ether ketone) ^a	Sugar (isohexide) ^a			<6 ^b	0.23-0.65 ^a	213-253 ^a	-- ^a	<40 ^a	-- ^a	-- ^a	-- ^a	-- ^a	-- ^a	~7 ^a	55,S1,S2 ^a
	Sugar (FDCA) ^a			<7 ^b	0.29-0.31 ^a	165-170 ^a		44 ^a						~2 ^a	56 ^a
Poly(carbonate-ester) ^a	Sugar (isohexide) ^a		Semicrystalline ^a	Amorphous ^a	>7 ^b	0.21 ^a	159 ^a	424/- ^a	32 ^a	↓ ^a	↓ ^a	X ^a	~1 ^a	58 ^a	
					32-43 ^b	0.47-0.56 ^a	167-193 ^a	>350/- ^a	61-84 ^a	~35 ^a	21 ^a				
Polyester ^a	Sugar (isohexide/LA) ^a		Amorphous ^a	Amorphous ^a	>6 ^b	0.13-0.67 ^a	158-209 ^a	>360/- ^a	52 ^a	↓ ^a	↓ ^a	X ^a	~6 ^a	S3,S4,S5 ^a	
					>9 ^b	0.11-0.38 ^a	173-196 ^a	>320/- ^a	>99 ^a				~9 ^a	S6 ^a	
					65-180 ^b	-- ^a	150-181 ^a	385/397 ^a	52-66 ^a				~15 ^a	S7,S8 ^a	
					57 ^b	-- ^a	156 ^a	385/398 ^a	60 ^a				~15 ^a	22 ^a	
Polyester ^a	Sugar (glucitol) ^a	Amorphous ^a	Amorphous ^a	13 ^b	0.32 ^a	154 ^a	-/377 ^a	52 ^a	↓ ^a	↓ ^a	X ^a	↓ ^a	30 ^a		

Supplementary Table 1. Characteristics of SUPERBIO, BPA-SEP, reported bio-based high T_g (>150 °C) thermoplastics, commercial petrochemical SEP, and commercial bioplastics (Continue).

Polymer type ^a	Biomass origin ^a	Polymerization ^a	Crystallinity ^a	M_n^b (kg mol ⁻¹) ^a	η_{inh}^b (dL g ⁻¹) ^a	T_g^b (°C) ^a	T_{5}/T_{410}^b (°C) ^a	Biomass content ^d (wt%) ^a	E (GPa) ^a	UTS (MPa) ^a	Melt processing ^c /recycling ^c	Scale ^k (g) ^a	Reference ^a
Polyester ^a	Lignin ^a	Condensation ^a	Semicrystalline ^a	48 ^b	-- ^a	157 ^a	-/>>305 ^a	>99 ^a	11 ^a	63 ^a	O ^a	~8 ^a	39 ^a
			Amorphous (not melting) ^a	↓ ^a	0.83 ^a	168 ^a	-/431 ^a	>99 ^a				~2 ^a	S9 ^a
Polycarbonate ^a	Sugar (isohexide) ^a	Ring-opening ^a	Amorphous ^a	2-14 ^b	-- ^a	155-243 ^a	-/><330 ^a	<51 ^a	↓ ^a	↓ ^a	X ^a	~0.5 ^a	33 ^a
	Sugar (mannose) ^a	Ring-opening ^a		<15 ^b	-- ^a	<184 ^a	-/-- ^a	<66 ^a				~0.4 ^a	34 ^a
Polyacetal ^a	Lignin ^a	Condensation ^a		>14 ^b	0.37-0.80 ^a	156-175 ^a	-- ^a	31-84 ^a				~1 ^a	23,S10 ^a
		Post-modification ^a		16 ^b	-- ^a	152 ^a	-/220-240 ^a	73 ^a				~0.3 ^a	S11 ^a
Polyurethane ^a	Sugar (isohexide) ^a	Condensation ^a		36-39 ^b	-- ^a	152-159 ^a	265-307/- ^a	67 ^a				~1 ^a	36,S12 ^a
Polyketone ^a	Terpene ^a	Radical addition ^a		53 ^b	-- ^a	157 ^a	224/- ^a	46 ^a				~2 ^a	38 ^a
Polyketone ^a	Terpene ^a	Radical addition ^a		8-19 ^b	0.23-0.35 ^a	152-191 ^a	>271/- ^a	~40 ^a				~5 ^a	24,S13,S14, S15 ^a
Polyketone ^a	Terpene ^a	Radical addition ^a		~100 ^b	-- ^a	162 ^a	-/327 ^a	>90 ^a				~1 ^a	35 ^a
Polydiene ^a	Lignin ^a	Metathesis ^a		49 ^b	-- ^a	156 ^a	380/- ^a	82 ^a				~0.1 ^a	40 ^a
Polyacrylate ^a	Lignin ^a	Metathesis ^a		18-50 ^b	-- ^a	154-205 ^a	302/- ^a	~68 ^a				↓ ^a	41,S16 ^a
Poly(N-aromatic imide) ^a	Itaconic acid ^a	Radical addition ^a		>19 ^b	-- ^a	153-238 ^a	>230/- ^a	28-51 ^a				~1 ^a	42,S17,S18, S19 ^a
Poly(hydroxyacrylamide) ^a	Lactone ^a			40-100 ^b	-- ^a	157-178 ^a	-- ^a	50-54 ^a				~2 ^a	43 ^a
Poly(methylene-lactide) ^a	Lactone ^a		76-358 ^b	-- ^a	213-254 ^a	>305/- ^a	>99 ^a	~0.3 ^a	S20 ^a				
Poly(methylene-ester) ^a	Tulipalin A ^a		~600 ^b	-- ^a	163-189 ^a	>320/- ^a	63-72 ^a	~0.6 ^a	S21 ^a				

Available in the main text and Supplementary Information

21. Feng, L., Zhu, W., Zhou, W., Li, C., Zhang, D., Xiao, Y., & Zheng, L. A designed synthetic strategy
toward poly(isosorbide terephthalate) copolymers: a combination of temporary modification,
transesterification, cyclization and polycondensation. *Polym. Chem.* **6**, 7470–7479 (2015).

22. Saber, C., M. Weidner, S. M., Fildier, A. & Kricheldorf., H. R. Copolyesters of isosorbide, succinic acid,
and isophthalic acid: biodegradable, high Tg engineering plastics. *J. Polym. Sci. Part A Polym. Chem.*
**51**, 2464–2471 (2013).

- 23. Chatti, S., Schwarz, G. & Kricheldorf, H. R. Cyclic and noncyclic polycarbonates of isosorbide
(1,4:3,6-dianhydro-D-glucitol). *Macromolecules* **39**, 9064–9070 (2006).
- 24. Lee, C.-H., Takagi, H., Okamoto, H., Kato, M., & Usuki, A. Synthesis, characterization, and properties
of polyurethanes containing 1,4:3,6-dianhydro-D-sorbitol. *J. Polym. Sci. Part A Polym. Chem.* **47**,
6025–6031 (2009).
- 30. Japu, C., de Ilarduya, A. M., Alla, A., García-Martín, M. G., Galbis, J. A., & Muñoz-Guerra, S. Bio-
based PBT copolyesters derived from D-glucose: influence of composition on properties. *Polym. Chem.*
**5**, 3190–3202 (2014).
- 33. Peña Carrodegua, L., Martín, C. & Kleij, A. W. Semiaromatic polyesters derived from renewable
terpene oxides with high glass transitions. *Macromolecules* **50**, 5337–5345 (2017).
- 34. Sanford, M. J., Pena Carrodegua, L., Van Zee, N. J., Kleij, A. W. & Coates, G. W. Alternating
copolymerization of propylene oxide and cyclohexene oxide with tricyclic anhydrides: access to
partially renewable aliphatic polyesters with high glass transition temperatures. *Macromolecules* **49**,
6394–6400 (2016).
- 35. Miyaji, H., Satoh, K. & Kamigaito, M. Bio-Based polyketones by selective ring-opening radical
polymerization of α -pinene-derived pinocarvone. *Angew. Chem. Int. Ed.* **55**, 1372–1376 (2016).
- 36. Pemba, A. G., Rostagno, M., Lee, T. A. & Miller, S. A. Cyclic and spirocyclic polyacetal ethers from
lignin-based aromatics. *Polym. Chem.* **5**, 3214–3221 (2014).
- 38. Rostagno, M., Shen, S., Ghiviriga, I. & Miller, S.A. Sustainable polyvinyl acetals from bioaromatic
aldehydes. *Polym. Chem.* **8**, 5049–5059 (2017).
- 39. Kaneko, T., Thi, T. H., Shi, D. J. & Akashi, M. Environmentally degradable, high-performance
thermoplastics from phenolic phytomonomers. *Nat. Mater.* **5**, 966–970 (2006).
- 40. Llevot, A., Grau, E., Carlotti, S., Grelier, S. & Cramail, H. ADMET polymerization of bio-based
biphenyl compounds. *Polym. Chem.* **6**, 7693–7700 (2015).
- **41. Holmberg, A. L., Reno, K. H., Nguyen, N. A., Wool, R. P. & Epps III, T. H. Syringyl methacrylate,**
**a hardwood lignin-based monomer for high-Tg polymeric materials. *ACS Macro Lett.* **5**, 574–578**
**(2016).**
- 42. Okada, S. & Matyjaszewski, K. Synthesis of bio-based poly (N-phenylitaconimide) by atom transfer
radical polymerization. *J. Polym. Sci. Part A Polym. Chem.* **53**, 822–827 (2015).
- 43. Britner, J. & Ritter, H. Self-activation of poly(methylenelactide) through neighboring-group effects: a
sophisticated type of reactive polymer. *Macromolecules* **48**, 3516–3522 (2015).
- 55. Chatti, S., Hani, M. A., Bornhorst, K. & Kricheldorf, H. R. Poly(ether sulfone) of isosorbide,
isomannide and isoidide. *High Perform. Polym.* **21**, 105–118 (2009).
- 56. Abderrazak, H. B., Fildier, A., Romdhane, H. B., Chatti, S. & Kricheldorf, H. R. Synthesis of new poly
(ether ketone) s derived from biobased diols. *Macromol. Chem. Phys.* **214**, 1423–1433 (2013).
- 57. Kanetaka, Y., Yamazaki, S. & Kimura, K. Preparation of poly (ether ketone) s derived from 2, 5-
furandicarboxylic acid via nucleophilic aromatic substitution polymerization. *J. Polym. Sci. Part A*
*Polym. Chem.* **54**, 3094–3101 (2016).
- 58. Kanetaka, Y., Yamazaki, S. & Kimura, K. Preparation of Poly (ether ketone)s derived from 2, 5-
furandicarboxylic acid by polymerization in ionic liquid. *Macromolecules* **49**, 1252–1258 (2016).

**Only available in Supplementary Information**

- S1. Belgacem, C., Medimagh, R., Kricheldorf, H., Ben Romdhane, H. & Chatti, S. Copolyethersulfones of
1, 4: 3, 6-dianhydrohexitols and bisphenol A. *Des. Monomers Polym.* **19**, 248–255 (2016).
- S2. Belgacem, C., Medimagh, R., Fildier, A., Bulete, A., Kricheldorf, H., Ben Romdhane, H. & Chatti, S.
Synthesis and characterization of isosorbide-based α,ω -dihydroxyethersulfone oligomers. *Des.*
*Monomers Polym.* **18**, 64–72 (2015).
- S3. Reinhard, S., Matthias, R. & Matthias, B. Synthesis and properties of high-molecular-weight polyesters
based on 1,4:3,6-dianhydrohexitols and terephthalic acid. *Die Makromolekulare Chemie* **194**, 53–64
(1993).
- S4. Kricheldorf, H. R., Behnken, G. & Sell, M. Influence of Isosorbide on Glass-Transition Temperature
and Crystallinity of Poly(butylene terephthalate). *J. Macromol. Sci. A* **44**, 679–684 (2007).
- S5. Thiem, J. & Lüders, H. Synthesis of polyterephthalates derived from dianhydrohexitols. *Polym. Bull.* **11**,
365–369 (1984).
- S6. Storbeck, R. & Ballauff, M. Synthesis and properties of polyesters based on 2,5-furandicarboxylic acid
and 1,4:3,6-dianhydrohexitols. *Polymer* **34**, 5003–5006 (1993).
- S7. Kricheldorf, H. R. & Weidner, S. M. High Tg copolyesters of lactide, isosorbide and isophthalic acid.
*Eur. Polym. J.* **49**, 2293–2302 (2013).
- S8. Kricheldorf, H. R. & Weidner, S. M. Copolyesters of Lactide, Isosorbide, and Terephthalic Acid-
Biobased, Biodegradable, High-Tg Engineering Plastics. *Macromol. Chem. Phys.* **214**, 726–733 (2013).
- S9. Nagata, M. Synthesis, characterization, and hydrolytic degradation of copolyesters of 3-(4-
hydroxyphenyl) propionic acid and p-hydroxybenzoic acid, vanilic acid, or syringic acid. *J. Appl. Polym.*
*Sci.* **78**, 2474–2481 (2000).
- S10. Chatti, S., Kricheldorf, H. R. & Schwarz, G. Copolycarbonates of isosorbide and various diols. *J.*
*Polym. Sci. Part A Polym. Chem.* **44**, 3616–3628 (2006).
- S11. Gregory, G. L., Jenisch, L. M., Charles, B., Kociok-Köhn, G. & Buchard, A. Polymers from Sugars and
CO₂: synthesis and polymerization of a d-mannose-based cyclic carbonate. *Macromolecules* **49**, 7165–
7169 (2016).
- S12. Rostagno, M., Price, E. J., Pemba, A. G., Ghiriviga, I, Abboud, K. A. & Miller, S. A. Sustainable
polyacetals from erythritol and bioaromatics. *J. Appl. Polym. Sci.* **133**, 44089 (2016).
- S13. Beldi, M., Medimagh, R., Chatti, S., Marque, S., Prim, D., Loupy, A. & Delolme, F.
Characterization of cyclic and non-cyclic poly-(ether-urethane)s bio-based sugar diols by a combination
of MALDI-TOF and NMR. *Eur. Polym. J.* **43**, 3415–3433 (2007).
- S14. Kricheldorf, H. R., Mix, R. & Weidner, S. M. Poly(ester urethane)s derived from lactide, isosorbide,
terephthalic acid, and various diisocyanates. *J. Polym. Sci. Part A Polym. Chem.* **52**, 867–875 (2014).
- S15. Marín, R., Alla, A., Martínez de Ilarduya, A. & Muñoz-Guerra, S. Carbohydrate-based
polyurethanes: A comparative study of polymers made from isosorbide and 1,4-butanediol. *J. Appl.*
*Polym. Sci.* **123**, 986–994 (2012).
- S16. Wang, S., Bassett, A. W., Wieber, G. V., Stanzione III, J. F. & Epps III, T. H. Effect of Methoxy
Substituent Position on Thermal Properties and Solvent Resistance of Lignin-Inspired
Poly(dimethoxyphenyl methacrylate)s. *ACS Macro Lett.* **6**, 802–807 (2017).

- S17. Vishal, A. & Veena, C. Studies on the copolymerization of methyl methacrylate with N-(o/m/p-
chlorophenyl) itaconimides. *J. Appl. Polym. Sci.* **82**, 2078–2086 (2001).
- S18. Vishal, A. & Veena, C. Copolymerization and thermal behavior of methyl methacrylate with N-
(phenyl/p-tolyl) itaconimides. *J. Appl. Polym. Sci.* **89**, 1195–1202 (2003).
- S19. Satoh, K. , Lee, D. , Nagai, K. & Kamigaito, M. Precision Synthesis of Bio-Based Acrylic
Thermoplastic Elastomer by RAFT Polymerization of Itaconic Acid Derivatives. *Macromol. Rapid*
*Commun.* **35**, 161–167 (2014).
- S20. Miyake, G. M., Zhang, Y. & Chen, E. Y. Polymerizability of Exo-methylene-lactide toward vinyl
addition and ring opening. *J. Polym. Sci. Part A Polym. Chem.* **53**, 1523–1532 (2015).
- S21. Agarwal, S. & Kumar, R. Synthesis of High-Molecular-Weight Tulipalin-A-Based Polymers by
Simple Mixing and Heating of Comonomers. *Macromol. Chem. Phys.* **212**, 603–612 (2011).

The reference (*ACS Sustainable Chem. Eng.* **3**, 662-667 (2015)) is omitted because the T_g it provides is
only around 130 °C (< 150 °C). As suggested by the reviewer, this reference is newly added in the
manuscript. Importantly, this paper also did not provide mechanical properties or thermal/solvent
processabilities.

We have added the following reference.

- 56. Gallagher, J. J., Hillmyer, M. A. & Reineke, T. M. Isosorbide-based polymethacrylates. *ACS Sustainable*
*Chem. Eng.* **3**, 662–667 (2015).

2.1. If plastics used in electronics are going to be the main motivation for this work, then a larger discussion
in the introduction is needed around this issue. For example, what percentage of plastic waste is
from electronics?

**Answer to 2.1.** As suggested by the reviewer, we have reinforced the motivation in the introduction as
follows. In printed circuit boards (PCB) made of abandoned electronic waste, the typical content of metals,
plastics, and ceramics are approximately 40, 30, and 30 wt%, respectively. Thermosets, e.g., epoxy; and
pseudo-thermoplastics, e.g., polyimide, generally have higher thermal stabilities; thus, they are preferred to
thermoplastics as materials for PCB. After curing, thermosets and pseudo-thermoplastics do not melt or
dissolve; the separation of metal from PCB requires harsh chemical degradation or pyrolysis of plastics, and
recycling of plastics is difficult. If the electronic parts are made of thermally durable thermoplastics such as
super engineering thermoplastic (SEP), both plastics and metals from PCB can be effectively recycled by
melting or dissolution. In this study, we suggest SUPERBIO as a potential sustainable high-performance
thermoplastic for such applications.

**In Page 3 Line 41,**

Since plastics have become indispensable in our life, their consumption has exponentially increased.¹ The
colossal demand for plastics has led to a large amount of wastes. For example, abandoned electronics
notably create printed circuit board (PCB) waste. The typical content of metals, plastics, and ceramics in
PCBs is approximately 40, 30, and 30 wt%, respectively.^{2,3} Among plastics, thermosets, e.g. epoxy, and
pseudo-thermoplastics, e.g. polyimide, generally have higher thermal stability; thus, they are more preferred
to thermoplastics as materials for PCB.^{4,5} After curing, thermosets and pseudo-thermoplastics do not melt
and dissolve; the separation of metal from PCB requires harsh chemical degradation or pyrolysis of plastics,
and the recycling of plastics is difficult.^{4,5} If electronic parts are made of thermally durable thermoplastics,
both plastics and metals from PCB can be effectively recycled by melting or dissolution.⁶ Likewise,
substituting thermosets with thermoplastics for many other applications increases the recycling rate of
plastic wastes.

**In reference**

3. Williams, P. T. Valorization of printed circuit boards from waste electrical and electronic equipment by
pyrolysis. *Waste Biomass Valorization* **1**, 107–120 (2010).

4. Hall, W. J. & Williams, P. T. Separation and recovery of materials from scrap printed circuit

boards. Resour. Conserv. Recy. **51**, 691–709 (2007).

5. Kim, J. W., Lee, A. S., Yu, S. & Han, J. W. En masse pyrolysis of flexible printed circuit board wastes
quantitatively yielding environmental resources. J. Hazard. Mater. **342**, 51–57 (2018).

6. Zou, Z., Zhu, C., Li, Y., Lei, X., Zhang, W. & Xiao, J. Rehealable, fully recyclable, and malleable
electronic skin enabled by dynamic covalent thermoset nanocomposite. *Sci. Adv.* **4**, eaaq0508 (2018).

2.2. Is the BPA-SEP polymer used extensively in these applications and therefore the best comparison for the
new bio polymers? The introduction does not provide enough context and motivation for this aspect.

**Answer to 2.2.** We would like to emphasize **BPA-SEP = polysulfone (PSU)**, a representative super
engineering thermoplastic (SEP). Among the high-performance thermoplastics, PSU is widely used as
transparent and heat/stress-resistant parts in electronic and biomedical devices, such as circuit boards,
battery seals, heat shields, power circuits, and dental instruments. In this study, **SUPERBIO** exhibited
higher mechanical and thermal properties than BPA-SEP, i.e., PSU did. Thus, we believe that **SUPERBIO**
can replace PSU for these applications.

We dragged the related content (**Page 8 Line 146**).

As a control, a PSU with $M_w = 151 \text{ kg mol}^{-1}$ and $\eta_{inh} = 1.61 \text{ dL g}^{-1}$ was synthesised with BPA and DFPS in
the presence of the crown-ether, and coded as **BPA-SEP**.

(Only for revision) Please find the information about the applications of PSU at

http://www2.basf.us/businesses/plasticportal/pdfs/Ultrason_brochure_final.pdf

Ultrason® S
Polysulfone (PSU)

The material of choice for demanding high temperature and filtration needs.

www.ultrason.net

BASF
The Chemical Company

Electrical Engineering and Electronics:
Coil formers, plug-and-socket connectors, injection-molded printed circuit boards, parts for power circuit breakers; parts for power contactors and relays, transparent covers for signal lamps and switchboards, lamp holders and lampshades, heat shields, sensors, chip carriers, chip trays and battery seals.

We have updated the content in **Page 4 Line 69.**

Among the EPs and SEPs, PSU and polycarbonate (PC) are widely used as transparent and heat/stress-
resistant parts of electronic and biomedical devices such as circuit boards, battery seals, heat shields, power
circuits, and dental instruments. There is great public health concern about BPA in PSU and PC, because it
causes developmental and reproductive problems in humans.¹¹⁻¹⁶

3. Also in the introduction, can you provide more specific motivation for the use of isosorbide over other
biorenewable sources? (lignin, etc) What about availability of the source?

**Answer:** We have added the following sentences and reference. The commercial application of isosorbide
(ISB) production technology has been under development over the past few years. Recently, a company
achieved the world's highest annual high-purity ISB production of 20,000 tons. In contrast, the production
volume of other bio-derived cyclic compounds including 2,5-furandicarboxylic acid, terpene, and lignin
derivatives are relatively low in industry.

**(Page 4 Line 80)** The commercial application of ISB production technology has been developing over the
past few years.^{53,54} A French agricultural company recently has achieved the world's highest annual high-
purity ISB production of 20,000 tons. Many global chemical companies are interested in the
commercialization of ISB-based polymers.

53. Gallagher, J. J., Hillmyer, M. A. & Reineke, T. M. Acrylic triblock copolymers incorporating isosorbide
for pressure sensitive adhesives. *ACS Sustainable Chem. Eng.* **4**, 3379–3387 (2016).

4. I did not see water sorption mentioned in this article. This is a negative aspect that typically plagues
isosorbide-based polymers. How does the water sorption of SUPERBIO compare to BPA-SEP?

**Answer:** As suggested by the reviewer, we have conducted the following experiments. **SUPERBIO** and
**BPA-SEP** were incubated in deionized (DI) water at 25 or 90 °C, for 24 h; subsequently, the specimen
weight and M_w of **SUPERBIO** and **BPA-SEP** were measured. The experiments did not affect the specimen
weight and M_w of both types of samples (Supplementary Figure 19). This suggests that **SUPERBIO** neither
depolymerizes nor absorbs water.

a		b	
SUPERBIO		SUPERBIO	
Weight (mg) before/after immersion in DI water at 25 °C for 24 h		Weight (mg) before/after immersion in DI water at 90 °C for 24 h	
Before	7.9	Before	7.3
After	7.9	After	7.3
M_w (g mol ⁻¹)		M_w (g mol ⁻¹)	
Before	93,800	Before	89,700
After	94,200	After	90,500
BPA-SEP		BPA-SEP	
Weight (mg) before/after immersion in DI water at 25 °C for 24 h		Weight (mg) before/after immersion in DI water at 90 °C for 24 h	
Before	7.5	Before	7.4
After	7.5	After	7.4
M_w (g mol ⁻¹)		M_w (g mol ⁻¹)	
Before	137,500	Before	134,100
After	132,400	After	131,800

**Supplementary Figure 19.** Weight (mg) and molecular weight of (Top) **SUPERBIO** and (Bottom) **BPA-**
**SEP** before and after immersion in deionized (DI) water for 24 h at (a) 25 and (b) 90 °C.

(Page 19 Line 356) The bracket materials must provide hydration resistance because of the moist
physiological environment". **SUPERBIO** and **BPA-SEP** were incubated in deionized (DI) water at 25 or 90
882 °C for 24 h; the specimen weight and M_w of **SUPERBIO** and **BPA-SEP** were then measured. The
883 experimental conditions did not affect the specimen weight and M_w of both types of samples
(Supplementary Figure 19). Thus, it can be concluded that **SUPERBIO** neither depolymerises nor absorbs
water.

5. Page 4 line 76: please provide literature context for the “unprecedented” molecular weight of your
 polymer

6. Page 6 line 117: please provide literature context for the claim that the scale is “10²-10⁴” times
 greater than other biobased high T_g thermoplastics

**Answer to both Q5 and Q6.** As suggested by the reviewer, we have conducted a literature survey for the **40**
 references (Supplementary Table 1) as follows.

Please find the references and the Supplementary Table 1.

1. Supplementary Tables, Figures, and Movies

Supplementary Table 1. Characteristics of SUPERBIO, BPA-SEP, reported bio-based high T_g (>150 °C) thermoplastics, commercial petrochemical SEP, and commercial bioplastics (Continue).

Polymer type ^a	Biomass origin ^a	Polymerization ^a	Crystallinity ^c	M _n ^a (kg mol ⁻¹) ^b	η _{inh} ^a (dL g ⁻¹) ^b	T _g ^a (°C) ^b	T _g ^a /T _{AI0} ^d (°C) ^b	Biomass content ^d (wt%) ^b	E (GPa) ^b	UTS (MPa) ^b	Melt processing ^e /recycling ^e	Scale ^b (g) ^b	Reference ^e	
SUPERBIO- (Poly(arylene ether)) ^a	Sugar (isohexide) ^a	Condensation ^a	Amorphous ^c	114 ^b	0.83 ^c	212 ^c	411/422 ^c	40-44 ^c	3.4 ^c	77 ^c	O ^c	~1,000 ^c	This study ^e	
BPA-SEP ^a (Poly(arylene ether sulfone)) ^a	Petrochemical (bisphenol-A) ^a			151 ^b	1.61 ^c	196 ^c	497/502 ^c	β ^c	3.0 ^c	51 ^c	~100 ^c			
Poly(arylene ether sulfone) ^a	Petrochemical (commercial PSU) ^a		Amorphous ^c	35 ^a	0.37-0.39 ^c	190 ^c	-	0 ^c	-	-	-	-	↓ ^c	Sigma Aldrich ^e
				50-60 ^b	-	187 ^c		0 ^c					BASF ^e	
Poly(arylene ether ketone) ^a	Sugar (isohexide) ^a		Amorphous ^c	<6 ^a	0.23-0.65 ^c	213-253 ^c	-	<40 ^c	-	-	-	-	~7 ^c	55, S1, S2 ^e
				<7 ^a	0.29-0.31 ^c	165-170 ^c		44 ^c					~2 ^c	56 ^e
				↓ ^c	0.21 ^c	159 ^c		424/- ^c					32 ^c	~2 ^c
Poly(carbonate-ester) ^a	Sugar (isohexide) ^a		Semicrystalline ^c	>7 ^a	0.31-0.59 ^c	154-161 ^c	-/458-470 ^c	32-42 ^c	↓ ^c	↓ ^c	↓ ^c	X ^c	~1 ^c	58 ^e
				32-43 ^b	0.47-0.56 ^c	167-193 ^c	>350/- ^c	61-84 ^c	~35 ^c	21 ^e				
Polyester ^a	Sugar (isohexide/FDCA) ^a		Amorphous ^c	>6 ^a	0.13-0.67 ^c	158-209 ^c	>360/- ^c	52 ^c	-	-	-	-	~6 ^c	S3, S4, S5 ^e
		>9 ^a		0.11-0.38 ^c	173-196 ^c	>320/- ^c	>99 ^c	~9 ^c					S6 ^e	
		65-180 ^b		-	150-181 ^c	385/397 ^c	52-66 ^c	~15 ^c					S7, S8 ^e	
		57 ^b		-	156 ^c	385/398 ^c	60 ^c	~15 ^c					22 ^e	
	Sugar (glucitol) ^a	Amorphous ^c	13 ^b	0.32 ^c	154 ^c	-/377 ^c	52 ^c	↓ ^c	↓ ^c	X ^c	↓ ^c	30 ^e		

Supplementary Table 1. Characteristics of SUPERBIO, BPA-SEP, reported bio-based high T_g (>150 °C) thermoplastics, commercial petrochemical SEP, and commercial bioplastics (Continue).^a

Polymer type ^a	Biomass origin ^a	Polymerization ^a	Crystallinity ^a	M_w^b (kg mol ⁻¹) ^a	η_{inh}^b (dL g ⁻¹) ^a	T_g^b (°C) ^a	T_g/T_{10}^b (°C) ^a	Biomass content ^d (wt%) ^a	E (GPa) ^a	UTS (MPa) ^a	Melt processing/ recycling ^a	Scale ^k (g) ^a	Reference ^a
Polyester ^a	Lignin ^a	Condensation ^a	Semicrystalline ^a	48 ^b	- ^a	157 ^a	->305 ^a	>99 ^a	11 ^a	63 ^a	O ^a	-8 ^a	39 ^a
			Amorphous ^a (not melting) ^a	- ^a	0.83 ^a	168 ^a	-/431 ^a	>99 ^a				-2 ^a	S9 ^a
	Terpene ^a	Ring-opening ^a	2-14 ^b	- ^a	155-243 ^a	-/<330 ^a	<51 ^a	-0.5 ^a				33 ^a	
Polycarbonate ^a	Sugar (isohexide) ^a	Condensation ^a	Amorphous ^a	<15 ^b	- ^a	<184 ^a	-/ ^a	<66 ^a				-0.4 ^a	34 ^a
	Sugar (mannose) ^a	Ring-opening ^a		>14 ^b	0.37-0.80 ^a	156-175 ^a	- ^a	31-84 ^a				-1 ^a	23,S10 ^a
Polyacetal ^a	Lignin ^a	Condensation ^a		16 ^b	- ^a	152 ^a	-/220-240 ^a	73 ^a				-0.3 ^a	S11 ^a
Polyvinylacetal ^a	Lignin ^a	Post-modification ^a		36-39 ^b	- ^a	152-159 ^a	265-307/ ^a	67 ^a				-1 ^a	36,S12 ^a
Polyurethane ^a	Sugar (isohexide) ^a	Condensation ^a		53 ^a	- ^a	157 ^a	224/ ^a	46 ^a				-2 ^a	38 ^a
Polyketone ^a	Terpene ^a	Radical addition ^a		8-19 ^b	0.23-0.35 ^a	152-191 ^a	>271/ ^a	~40 ^a				-5 ^a	24,S13,S14, S15 ^a
Polydiene ^a	Lignin ^a	Metathesis ^a		~100 ^b	- ^a	162 ^a	-/327 ^a	>90 ^a				-1 ^a	35 ^a
Polyacrylate ^a	Lignin ^a	Metathesis ^a		49 ^b	- ^a	156 ^a	380/ ^a	82 ^a				-0.1 ^a	40 ^a
Poly(N-aromatic imide) ^a	Itaconic acid ^a	Radical addition ^a		18-50 ^b	- ^a	154-205 ^a	302/ ^a	~68 ^a				- ^a	41,S16 ^a
Poly(hydroxyacrylamide) ^a	Lactone ^a			>19 ^b	- ^a	153-238 ^a	>230/ ^a	28-51 ^a				-1 ^a	42,S17,S18, S19 ^a
Poly(methylene lactide) ^a	Lactone ^a			40-100 ^b	- ^a	157-178 ^a	- ^a	50-54 ^a				-2 ^a	43 ^a
Poly(methylene-ester) ^a	Tulipalin A ^a			76-358 ^b	- ^a	213-254 ^a	>305/ ^a	>99 ^a				-0.3 ^a	S20 ^a
Poly(methylene-ester) ^a	Tulipalin A ^a			-600 ^b	- ^a	163-189 ^a	>320/ ^a	63-72 ^a	-0.6 ^a	S21 ^a			

Available in the main text and Supplementary Information

- 21. Feng, L., Zhu, W., Zhou, W., Li, C., Zhang, D., Xiao, Y., & Zheng, L. A designed synthetic strategy
toward poly(isosorbide terephthalate) copolymers: a combination of temporary modification,
transesterification, cyclization and polycondensation. *Polym. Chem.* **6**, 7470–7479 (2015).
- 22. Saber, C., M. Weidner, S. M., Fildier, A. & Kricheldorf., H. R. Copolyesters of isosorbide, succinic acid,
and isophthalic acid: biodegradable, high Tg engineering plastics. *J. Polym. Sci. Part A Polym. Chem.*
**51**, 2464–2471 (2013).
- 23. Chatti, S., Schwarz, G. & Kricheldorf, H. R. Cyclic and noncyclic polycarbonates of isosorbide
(1,4:3,6-dianhydro-D-glucitol). *Macromolecules* **39**, 9064–9070 (2006).
- 24. Lee, C.-H., Takagi, H., Okamoto, H., Kato, M., & Usuki, A. Synthesis, characterization, and properties
of polyurethanes containing 1,4:3,6-dianhydro-D-sorbitol. *J. Polym. Sci. Part A Polym. Chem.* **47**,
6025–6031 (2009).
- 30. Japu, C., de Ilarduya, A. M., Alla, A., García-Martín, M. G., Galbis, J. A., & Muñoz-Guerra, S. Bio-
based PBT copolyesters derived from D-glucose: influence of composition on properties. *Polym. Chem.*
**5**, 3190–3202 (2014).
- 33. Peña Carrodegua, L., Martín, C. & Kleij, A. W. Semiaromatic polyesters derived from renewable
terpene oxides with high glass transitions. *Macromolecules* **50**, 5337–5345 (2017).
- 34. Sanford, M. J., Pena Carrodegua, L., Van Zee, N. J., Kleij, A. W. & Coates, G. W. Alternating
copolymerization of propylene oxide and cyclohexene oxide with tricyclic anhydrides: access to
partially renewable aliphatic polyesters with high glass transition temperatures. *Macromolecules* **49**,
6394–6400 (2016).
- 35. Miyaji, H., Satoh, K. & Kamigaito, M. Bio-Based polyketones by selective ring-opening radical
polymerization of α -pinene-derived pinocarvone. *Angew. Chem. Int. Ed.* **55**, 1372–1376 (2016).

- 36. Pemba, A. G., Rostagno, M., Lee, T. A. & Miller, S. A. Cyclic and spirocyclic polyacetal ethers from
lignin-based aromatics. *Polym. Chem.* **5**, 3214–3221 (2014).
- 38. Rostagno, M., Shen, S., Ghiviriga, I. & Miller, S.A. Sustainable polyvinyl acetals from bioaromatic
aldehydes. *Polym. Chem.* **8**, 5049–5059 (2017).
- 39. Kaneko, T., Thi, T. H., Shi, D. J. & Akashi, M. Environmentally degradable, high-performance
thermoplastics from phenolic phytomonomers. *Nat. Mater.* **5**, 966–970 (2006).
- 40. Llevot, A., Grau, E., Carlotti, S., Grelier, S. & Cramail, H. ADMET polymerization of bio-based
biphenyl compounds. *Polym. Chem.* **6**, 7693–7700 (2015).
- 41. Holmberg, A. L., Reno, K. H., Nguyen, N. A., Wool, R. P. & Epps III, T. H. Syringyl methacrylate, a
hardwood lignin-based monomer for high-Tg polymeric materials. *ACS Macro Lett.* **5**, 574–578 (2016).
- 42. Okada, S. & Matyjaszewski, K. Synthesis of bio-based poly (N-phenylitaconimide) by atom transfer
radical polymerization. *J. Polym. Sci. Part A Polym. Chem.* **53**, 822–827 (2015).
- 43. Britner, J. & Ritter, H. Self-activation of poly(methylenelactide) through neighboring-group effects: a
sophisticated type of reactive polymer. *Macromolecules* **48**, 3516–3522 (2015).
- 55. Chatti, S., Hani, M. A., Bornhorst, K. & Kricheldorf, H. R. Poly(ether sulfone) of isosorbide,
isomannide and isodide. *High Perform. Polym.* **21**, 105–118 (2009).
- 56. Abderrazak, H. B., Fildier, A., Romdhane, H. B., Chatti, S. & Kricheldorf, H. R. Synthesis of new poly
(ether ketone) s derived from biobased diols. *Macromol. Chem. Phys.* **214**, 1423–1433 (2013).
- 57. Kanetaka, Y., Yamazaki, S. & Kimura, K. Preparation of poly (ether ketone) s derived from 2, 5-
furandicarboxylic acid via nucleophilic aromatic substitution polymerization. *J. Polym. Sci. Part A*
*Polym. Chem.* **54**, 3094–3101 (2016).
- 58. Kanetaka, Y., Yamazaki, S. & Kimura, K. Preparation of Poly (ether ketone)s derived from 2, 5-
furandicarboxylic acid by polymerization in ionic liquid. *Macromolecules* **49**, 1252–1258 (2016).
- **Only available in Supplementary Information**
- S1. Belgacem, C., Medimagh, R., Kricheldorf, H., Ben Romdhane, H. & Chatti, S. Copolyethersulfones of
1, 4: 3, 6-dianhydrohexitols and bisphenol A. *Des. Monomers Polym.* **19**, 248–255 (2016).
- S2. Belgacem, C., Medimagh, R., Fildier, A., Bulete, A., Kricheldorf, H., Ben Romdhane, H. & Chatti, S.
Synthesis and characterization of isosorbide-based α,ω -dihydroxyethersulfone oligomers. *Des.*
*Monomers Polym.* **18**, 64–72 (2015).
- S3. Reinhard, S., Matthias, R. & Matthias, B. Synthesis and properties of high-molecular-weight polyesters
based on 1,4:3,6-dianhydrohexitols and terephthalic acid. *Die Makromolekulare Chemie* **194**, 53–64
(1993).
- S4. Kricheldorf, H. R., Behnken, G. & Sell, M. Influence of Isosorbide on Glass-Transition Temperature
and Crystallinity of Poly(butylene terephthalate). *J. Macromol. Sci. A* **44**, 679–684 (2007).
- S5. Thiem, J. & Lüders, H. Synthesis of polyterephthalates derived from dianhydrohexitols. *Polym. Bull.* **11**,
365–369 (1984).
- S6. Storbeck, R. & Ballauff, M. Synthesis and properties of polyesters based on 2,5-furandicarboxylic acid
and 1,4:3,6-dianhydrohexitols. *Polymer* **34**, 5003–5006 (1993).
- S7. Kricheldorf, H. R. & Weidner, S. M. High Tg copolyesters of lactide, isosorbide and isophthalic acid.
*Eur. Polym. J.* **49**, 2293–2302 (2013).

- S8. Kricheldorf, H. R. & Weidner, S. M. Copolyesters of Lactide, Isosorbide, and Terephthalic Acid-
Biobased, Biodegradable, High-Tg Engineering Plastics. *Macromol. Chem. Phys.* **214**, 726–733 (2013).
- S9. Nagata, M. Synthesis, characterization, and hydrolytic degradation of copolyesters of 3-(4-
hydroxyphenyl) propionic acid and p-hydroxybenzoic acid, vanilic acid, or syringic acid. *J. Appl. Polym.*
*Sci.* **78**, 2474–2481 (2000).
- S10. Chatti, S., Kricheldorf, H. R. & Schwarz, G. Copolycarbonates of isosorbide and various diols. *J.*
*Polym. Sci. Part A Polym. Chem.* **44**, 3616–3628 (2006).
- S11. Gregory, G. L., Jenisch, L. M., Charles, B., Kociok-Köhn, G. & Buchard, A. Polymers from Sugars and
CO₂: synthesis and polymerization of a d-mannose-based cyclic carbonate. *Macromolecules* **49**, 7165–
7169 (2016).
- S12. Rostagno, M., Price, E. J., Pemba, A. G., Ghiriviga, I., Abboud, K. A. & Miller, S. A. Sustainable
polyacetals from erythritol and bioaromatics. *J. Appl. Polym. Sci.* **133**, 44089 (2016).
- S13. Beldi, M., Medimagh, R., Chatti, S., Marque, S., Prim, D., Loupy, A. & Delolme, F.
Characterization of cyclic and non-cyclic poly-(ether-urethane)s bio-based sugar diols by a combination
of MALDI-TOF and NMR. *Eur. Polym. J.* **43**, 3415–3433 (2007).
- S14. Kricheldorf, H. R., Mix, R. & Weidner, S. M. Poly(ester urethane)s derived from lactide, isosorbide,
terephthalic acid, and various diisocyanates. *J. Polym. Sci. Part A Polym. Chem.* **52**, 867–875 (2014).
- S15. Marín, R., Alla, A., Martínez de Ilarduya, A. & Muñoz-Guerra, S. Carbohydrate-based
polyurethanes: A comparative study of polymers made from isosorbide and 1,4-butanediol. *J. Appl.*
*Polym. Sci.* **123**, 986–994 (2012).
- S16. Wang, S., Bassett, A. W., Wieber, G. V., Stanzione III, J. F. & Epps III, T. H. Effect of Methoxy
Substituent Position on Thermal Properties and Solvent Resistance of Lignin-Inspired
Poly(dimethoxyphenyl methacrylate)s. *ACS Macro Lett.* **6**, 802–807 (2017).
- S17. Vishal, A. & Veena, C. Studies on the copolymerization of methyl methacrylate with N-(o/m/p-
chlorophenyl) itaconimides. *J. Appl. Polym. Sci.* **82**, 2078–2086 (2001).
- S18. Vishal, A. & Veena, C. Copolymerization and thermal behavior of methyl methacrylate with N-
(phenyl/p-tolyl) itaconimides. *J. Appl. Polym. Sci.* **89**, 1195–1202 (2003).
- S19. Satoh, K., Lee, D., Nagai, K. & Kamigaito, M. Precision Synthesis of Bio-Based Acrylic
Thermoplastic Elastomer by RAFT Polymerization of Itaconic Acid Derivatives. *Macromol. Rapid*
*Commun.* **35**, 161–167 (2014).
- S20. Miyake, G. M., Zhang, Y. & Chen, E. Y. Polymerizability of Exo-methylene-lactide toward vinyl
addition and ring opening. *J. Polym. Sci. Part A Polym. Chem.* **53**, 1523–1532 (2015).
- S21. Agarwal, S. & Kumar, R. Synthesis of High-Molecular-Weight Tulipalin-A-Based Polymers by
Simple Mixing and Heating of Comonomers. *Macromol. Chem. Phys.* **212**, 603–612 (2011).

7. Regarding tensile testing: a) Why only test 3 specimens? Typically in the range of 10 test bars yields more reliable results. b) Please add error bars to the parameters on page 9 lines 153-154. c) On page 23 line 454: these are not the correct bar dimensions for ASTM 638-03. Therefore you cannot be following this ASTM standard, as the bar dimensions are a critical aspect of the ASTM standard. The tensile properties will be very different for a film vs bulk specimen.

.002

Answer 7a: As suggested by the reviewer, we have measured the tensile properties with 10 specimens for **SUPERBIO** and 8 specimens for **BPA-SEP**. All the tensile stress-strain curves and the data statistics are presented in Fig. 2b and Supplementary Fig. 5, respectively. The data of the duplicate tensile tests confirm the reproducibility.

.007

.008

Fig. 2. Damage-tolerant bio-based super engineering plastic. a, Photographs of the solution-casted pristine, origami-folded (top), and unfolded (bottom) films of **SUPERBIO** (scale bar: 1 cm, Supplementary Movie 1). **b,** Tensile stress-strain curves of **SUPERBIO** (n = 10) and **BPA-SEP** (n = 8) films. **c,** (Left) Original and (right) differential tear load-distance curves of **SUPERBIO** and **BPA-SEP** films. Inset is photograph of the specimen for tear test (KS M ISO 34-1:2014, scale bar: 1 cm). **d,** Impact strength of the

.014 injection-moulded **SUPERBIO** and **BPA-SEP**. Inset is photograph of the rectangular bar-shaped specimens
.015 for the impact test (scale bar: 1 cm). Each impact strength value represents the mean and standard error of
.016 triplicate samples. **e**, Recycling of **SUPERBIO** products (scale bar: 1 cm). **f**, DMF-GPC profiles before/after
.017 thermal processing. M_w s of pristine and injection-moulded **SUPERBIO** were 92.2, and 89.7 kg mol⁻¹,
.018 respectively.

.019

.020

.021 **Supplementary Figure 5.** Tensile properties of **SUPERBIO** (n = 10) and **BPA-SEP** (n = 8).

.022

.023 **Answer 7b:** As the reviewer suggested, we have added the standard error in Fig. 2.

.024

.025 **Answer 7c:**

.026 Firstly, ASTM D638 is less preferred for measuring thin films; however, it can be employed. ASTM D638
.027 covers the determination of the tensile properties of unreinforced and reinforced plastics in the form of
.028 standard dumbbell-shaped test specimens. This test method is applicable for materials of **any thickness**, up
.029 to 14 mm.

.030

Standard Test Method for Tensile Properties of Plastics¹

This standard is issued under the fixed designation D638; the number immediately following the designation indicates the year of original adoption or, in the case of revision, the year of last revision. A number in parentheses indicates the year of last reapproval. A superscript epsilon (ϵ) indicates an editorial change since the last revision or reapproval.

This standard has been approved for use by agencies of the U.S. Department of Defense.

1. Scope*

1.1 This test method covers the determination of the tensile properties of unreinforced and reinforced plastics in the form of standard dumbbell-shaped test specimens when tested under defined conditions of pretreatment, temperature, humidity, and testing machine speed.

1.2 This test method is applicable for testing materials of any thickness up to 14 mm (0.55 in.). However, for testing specimens in the form of thin sheeting, including film less than 1.0 mm (0.04 in.) in thickness, ASTM standard D882 is the preferred test method. Materials with a thickness greater than 14 mm (0.55 in.) shall be reduced by machining.

1.3 This test method includes the option of determining Poisson's ratio at room temperature.

NOTE 1—This standard and ISO 527-1 address the same subject matter, but differ in technical content.

NOTE 2—This test method is not intended to cover precise physical

consider the precautions and limitations of this method found in Note 2 and Section 4 before considering these data for engineering design.

1.5 The values stated in SI units are to be regarded as standard. The values given in parentheses are for information only.

1.6 This standard does not purport to address all of the safety concerns, if any, associated with its use. It is the responsibility of the user of this standard to establish appropriate safety and health practices and determine the applicability of regulatory limitations prior to use.

2. Referenced Documents

2.1 ASTM Standards:²

D229 Test Methods for Rigid Sheet and Plate Materials Used for Electrical Insulation

D412 Test Methods for Vulcanized Rubber and Thermoplastic Elastomers—Tension

.031

.032

.033

.034

.035

.036

.037

.038

.039

.040

.041

.042

.043

.044

.045

Secondly, the main aim of this study is to show that a newly synthesized bio-based plastic (**SUPERBIO**) has mechanical properties considerably higher than those of a representative petrochemical high-performance plastic, i.e., PSU (=BPA-SEP). **In the comparison, all mechanical properties were measured under identical conditions.** The data of the duplicated tests in the identical condition prove our claim.

Thirdly, we would like to emphasize that the shape of the specimens is highly uniform. As shown in Supplementary Fig. 23, all the specimens were prepared using a jockey type-cutter, and reliable data could be obtained. In contrast, many groups still measure the mechanical properties without the use of an official standard, e.g., ASTM. In reference #39 (*Nature Materials* **5**, 966–970 (2006)), the sample for tensile mechanical property measurement was prepared without using an official standard method. In contrast, our system can provide reproducible data of tensile properties because the regular shape of specimens are replicated.

1-4 Mechanical properties. Young's modulus, mechanical strength at break, and strain at break of poly(4HCA-co-DHCA)s were measured by three-points bending test of the rectangular samples (0.3 x 5.0 x >25 mm³). All tests were carried out at 25 °C using a fixed nominal displacement rate of 1mm·min⁻¹. The data of at least three samples were averaged.

.046

.047 (Only for revision) Method section for mechanical property measurement in *Nature Materials* 5, 966–970
.048 (2006)

.049

.050 We have updated the following Figure in Supplementary Information.

.051

.052 **Supplementary Figure 23.** A jockey type-cutter for dog-bone shape specimen following ASTM D638.

.053

.054

.055

.056

.057 8. Page 9 line 169: How do these results confirm the thermal stability? This is just a comparison to BPA-SEP.
.058 A better approach would be to report properties after each thermal cycle and see if they change.

.059

.060 **Answer:** As suggested by the reviewer, we have measured the molecular weight after the injection-molding.
.061 As shown in Fig. 2e,f, the molecular weight was not changed. In addition, we have monitored the M_w
.062 change of **SUPERBIO** during 5 programmed cycles of heat treatments (Supplementary Fig. 8). Each cycle
.063 consists of heating (30 °C to 270 °C) and cooling (270 °C to 30 °C). The M_w barely changed until the second
.064 heat treatment. The M_w of **SUPERBIO** decreased by only 9% after the fifth heat treatment. The results
.065 confirmed its thermal stability at the melt state as well as the recyclability.

.066

.067 We have updated Fig. 2 as follows.

.068

.069 **Fig. 2. Damage-tolerant bio-based super engineering plastic.** **a**, Photographs of the solution-casted
.070 pristine, origami-folded (top), and unfolded (bottom) films of **SUPERBIO** (scale bar: 1 cm, Supplementary
.071 Movie 1). **b**, Tensile stress-strain curves of **SUPERBIO** (n = 10) and **BPA-SEP** (n = 8) films. **c**, (Left)
.072 Original and (right) differential tear load-distance curves of **SUPERBIO** and **BPA-SEP** films. Inset is
.073 photograph of the specimen for tear test (KS M ISO 34-1:2014, scale bar: 1 cm). **d**, Impact strength of the

.074 injection-moulded **SUPERBIO** and **BPA-SEP**. Inset is photograph of the rectangular bar-shaped specimens
.075 for the impact test (scale bar: 1 cm). Each impact strength value represents the mean and standard error of
.076 triplicate samples. **e**, Recycling of **SUPERBIO** products (scale bar: 1 cm), and **f**, DMF-GPC profiles
.077 before/after thermal processing. M_w s of pristine and injection-moulded **SUPERBIO** were 92.2, and 89.7 kg
.078 mol^{-1} , respectively.

.079

.080 **(Page 11 Line 211)** To evaluate the thermal stability of **SUPERBIO** in detail during melt processing, we
.081 have monitored the M_w change of **SUPERBIO** during five programmed cycles of heat treatments
.082 (Supplementary Fig. 8). Each cycle consists of heating (30 to 270 °C) and cooling (270 to 30 °C) with a
.083 ramp rate of 10 °C min^{-1} under a nitrogen atmosphere. The M_w hardly changed until the second heat
.084 treatment. The M_w of **SUPERBIO** decreased to only 9% after the fifth heat treatment. This suggests that
.085 **SUPERBIO** can be recycled through a series of melting and moulding.⁶³

.086

.087 We have updated the Supplementary Figure.

.088

.089 **Supplementary Figure 8.** The M_w changes of **SUPERBIO** within programmed 5 cycles of heat treatments.
.090 (a) One cycle of heat treatments consists of heating (30 to 270 °C) and cooling (270 to 30 °C) with a rate of
.091 10 °C min^{-1} under nitrogen atmosphere. (b) The M_w of **SUPERBIO** depending on the number of heat
.092 treatments.

.093

.094

.095

.096

.097 9. Page 10 line 170: You refer to the quantum chemical simulations here without any details. This is
.098 awkward for the reader. I suggest bringing the quantum chemical simulations discussion right after the
.099 thermal property discussion, and not mentioning it here on this line. (Therefore, bring the quantum chemical
.100 simulations discussion to before the electronic device discussion).

.101

.102 **Answer:** As suggested by the reviewer, we have edited the manuscript. Now, the quantum chemical
.103 simulations discussion is located right after the thermal property. We provide the details of the quantum
.104 chemical simulations in the answer to question #13 of reviewer #3.

.105

.106

.107

.108

.109 10. Page 12 line 220: It seems that BPA-SEP is not a particularly suitable polymer for this application.
 .110 Therefore, it is very confusing why this is the polymer chosen for comparison purposes. If you want to show
 .111 that SUPERBIO is a suitable polymer in this application, why not compare it to a polymer that is currently
 .112 used for this purpose?
 .113 11. Similarly, is the BPA-SEP actually used in the orthodontic devices discussed on page 16? It seems these
 .114 are PSUs and PCs, based on your description of current state of the art. Why is BPA-SEP used for
 .115 comparison purposes in this application?
 .116

.117 **Answer to both Q10 and Q11:** We would like to emphasize **BPA-SEP = polysulfone (PSU) again**. Among
 .118 high performance thermoplastics, PSU is widely used as transparent and heat/stress-resistant parts in
 .119 electronic and biomedical devices, such as circuit boards, battery seals, heat shields, power circuits, and
 .120 dental instruments. In this study, **SUPERBIO** exhibited mechanical and thermal properties higher than those
 .121 of BPA-SEP, i.e., PSU. Thus, we believe that **SUPERBIO** can replace PSU for these applications. For better
 .122 understanding for the readers, we have modified Fig. 1a as follows.
 .123

.124 We have marked “BPA-SEP is the commercially available polysulphone (PSU)” in Fig. 1a.

.125

.126

.127 We dragged the related content (Page 8 Line 146).

.128 As a control, a PSU with $M_w = 151 \text{ kg mol}^{-1}$ and $\eta_{inh} = 1.61 \text{ dL g}^{-1}$ was synthesised with BPA and DFPS in
.129 the presence of the crown-ether, and coded as **BPA-SEP**.

.130

.131 (Only for revision) Please find the information about the applications of PSU at
.132 http://www2.basf.us/businesses/plasticportal/pdfs/Ultrason_brochure_final.pdf

.133

.134

.135 We have updated the content in Page 4 Line 69.

.136 Among the EPs and SEPs, PSU and polycarbonate (PC) are widely used as transparent and heat/stress-
.137 resistant parts of electronic and biomedical devices such as circuit boards, battery seals, heat shields, power
.138 circuits, and dental instruments. There is great public health concern about BPA in PSU and PC, because it
.139 causes developmental and reproductive problems in humans.¹¹⁻¹⁶

.140

.141

.142 *12. In general the application discussions (electronic device and medical implant) seem somewhat out of*
.143 *place in this article. Perhaps if better motivation were provided in the introduction for one application (such*
.144 *as the electronic devices), then this would be a good way to end the article. However, putting both in with*
.145 *little motivation provided makes the article very difficult to follow.*

.146

.147 **Answer:** As suggested by the reviewer, we have strengthened the introduction section of the manuscript as
.148 follows.

.149 **(Page 3 Line 41)**

.150 Since plastics have become indispensable in our life, their consumption has exponentially increased.¹ The
.151 colossal demand for plastics has led to a large amount of wastes. For example, abandoned electronics
.152 notably create printed circuit board (PCB) waste. The typical content of metals, plastics, and ceramics in
.153 PCBs is approximately 40, 30, and 30 wt%, respectively.^{2,3} Among plastics, thermosets, e.g. epoxy, and
.154 pseudo-thermoplastics, e.g. polyimide, generally have higher thermal stability; thus, they are more preferred
.155 to thermoplastics as materials for PCB.^{4,5} After curing, thermosets and pseudo-thermoplastics do not melt
.156 and dissolve; the separation of metal from PCB requires harsh chemical degradation or pyrolysis of plastics,
.157 and the recycling of plastics is difficult.^{4,5} If electronic parts are made of thermally durable thermoplastics,
.158 both plastics and metals from PCB can be effectively recycled by melting or dissolution.⁶ Likewise,
.159 substituting thermosets with thermoplastics for many other applications increases the recycling rate of
.160 plastic wastes.

.161 According to superiority of thermal and mechanical performances, thermoplastics are generally classified in
.162 the following order: commodity plastics < engineering plastics (EPs) < super engineering plastics (SEPs).
.163 There is no appropriate quantitative standard for the precise classification because most physical properties
.164 of thermoplastics exist across all the above-mentioned three classes.^{7,8} In polymer science, glass transition
.165 temperature (T_g) is a general indicator to represent thermomechanical characteristics of polymers. In the
.166 same order, the three classes of thermoplastics typically have the T_g ranges of <100, 100–150, and >150 °C.
.167 ^{1,7–10} SEPs, also known as high-performance or specialty thermoplastics, are gradually replacing thermosets
.168 and pseudo-thermoplastics as thermally and mechanically robust materials for aircrafts, automobiles,
.169 electronics, dental devices and in household/children's products because of their recyclability.^{2,11}
.170 Poly(arylene ether)s (PAEs) are a major group of SEPs, and they include polysulphone (PSU), polyether
.171 ether ketone, and polyphenylsulfone.^{12,13}

.172

.173 **(Page 4 Line 68)**

.174 Among the EPs and SEPs, PSU and polycarbonate (PC) are widely used as transparent and heat/stress-
.175 resistant parts of electronic and biomedical devices such as circuit boards, battery seals, heat shields, power
.176 circuits, and dental instruments.

.177 13. Page 14 line 250: please provide derived equations and more info on the calculations.

.178

.179 **Answer:** We have strengthened the section about quantum chemical simulations as follows. In addition, we
.180 have discussed quantum chemical simulations in the original Supplementary Information.

.181

.182 We have updated Fig. 3a as follows.

.183

.184

.185 **Fig. 3. Quantum chemical simulation. a,** Schematic illustration of different vibration energy levels in a
.186 potential energy well (x-axis: interatomic distance). **b,** Optimised ground state geometries of (top)
.187 **SUPERBIO** and (bottom) **BPA-SEP** repeating units. **c,** black; H, white; O, red; and S, yellow. **c,** Potential
.188 energy curve along with relative length of repeating unit.

.189

.190 We have updated Fig. 3a as follows (**Page 13,14**).

.191 The higher thermal and mechanical properties of **SUPERBIO** over **BPA-SEP** are quite surprising, because
.192 the aromatic BPA has been considered to be more suited for such purposes than the aliphatic ISB. In the
.193 glassy state ($T < T_g$), a polymer behaves like a typical elastic solid, the thermal and mechanical expansion is

.194 the sum result of different energy-dependent oscillatory bonds: strong covalent and weak van der Waals
.195 bonds.⁶⁵ By a quantum chemical simulation, we have studied the effects of ISB as well as BPA on the
.196 thermal/mechanical properties of a single molecule of **SUPERBIO** by using Q-Chem 4.3 (Q-Chem Inc.,
.197 Pleasanton, CA, USA), i.e. we have explored their contributions on the geometric restraint of the covalent
.198 linkages with exclusion of the physical interactions (i.e. inter-polymeric interactions). This approach marks a
.199 starting point to understand isosorbide's thermomechanical property at a fundamental level. **Supporting**
.200 **Information** includes the detailed description of quantum chemical simulations.

.201 A correlation between the vibrational energy gap and the relative **covalent bond** length was derived by
.202 calculating an anharmonic potential energy curve (PEC), $V(x) = \frac{1}{2}kx^2 - \lambda x^3$ where k is the spring constant
.203 of chemical bond and λ is the anharmonicity constant (Fig. 3a). The one-dimensional average position (green
.204 dot position) is expressed as $\langle x \rangle^{(1)} = \frac{3\lambda\hbar}{\sqrt{mk}}\left(\nu + \frac{1}{2}\right)$ where \hbar is the reduced Planck constant, m is the mass,
.205 and ν is the vibrational energy level. For example, on increasing the temperature, as the vibrational energy is
.206 excited from ground state toward $\nu = 4$, the green dot deviates from the original position, i.e. bond length
.207 increases. This indicates that the increasing system energy gives rise to elongation of covalent bonds. The
.208 shape of PEC is dependent on the geometric restraint of covalent linkages. At the given ν , i.e. temperature,
.209 the higher curvature (or steeper slope) of PEC results in the lower elongation of chemical bonds.

.210 In the theoretical model, the repeating unit for each polymer is chosen with an assumption that the relative
.211 length change of each polymer chain is not significantly different from that of the repeating unit. This
.212 assumption is reasonable because both are fully amorphous polymers with only short-range order. After the
.213 geometry of each systematically elongated repeating unit was optimised to consider the relaxation effect
.214 from angle changes according to the density functional theory, well-known as DFT, by using the B3LYP/6-
.215 31G* basis set (Fig. 3b), a PEC along with the relative bond length was calculated for a given vibrational
.216 level (Fig. 3c).

.217
.218 We have dragged the related contents from Supplementary Information (**Page 34**).

.219 **Quantum chemical simulation.**

.220 A cause of thermal expansion is well explained with an increase in the interatomic distances.^{S40} The bond
.221 length increase can be understood using an anharmonic potential energy curve (PEC) with explicit vibration
.222 energy levels. All molecules have vibration energy even at zero temperature, and every bond length
.223 oscillates with respect to its expectation value of position (green circles in Figure 3a). Upon heating, the
.224 molecules obtain thermal energy, which is converted to molecular vibration energy, and this leads to the

.225 increased population in the vibrationally excited states ($v \geq 2$). Due to the asymmetry of intermolecular
.226 potential energy, vibrational excited states have a larger expectation value of position. As marked in Figure
.227 3a, the expectation value of position gets away from the ground state geometry as the vibration excitation
.228 goes from $v = 1$ to $v = 4$. Therefore, the molecule in the vibrationally excited state has longer bond length
.229 compared to its ground state correspondent.

.230

.231 **Figure 3a (Main text).** Potential energy curve (PEC) along the arbitrary interatomic distance with vibration
.232 energy levels: low vs. high curvature PECs.

.233 This discussion indicates that the population in the vibrationally excited states is directly related to thermal
.234 expansion: The larger vibrational energy gap implies a lower population in the vibrationally excited states.
.235 Harmonic approximation provides a relatively easy way to estimate the vibrational energy gap analytically
.236 compared to the anharmonic case. Different molecules have different interatomic potential and this yields
.237 the different shape of the PEC, especially in terms of the steepness of the potential energy wall (i.e. the
.238 second derivative of the PEC). The molecule having stronger (weaker) interatomic interaction would have a
.239 relatively steep (flat) PEC. According to a simple harmonic oscillator model, the steepness of the PEC is
.240 proportional to the square of the vibration energy gap ($\Delta E_v = E_{v=n+1} - E_{v=n}$). The molecules exhibiting a steep
.241 (flat) PEC have larger ΔE_v (smaller ΔE_v), and this results in relatively less (more) occupation in the
.242 vibrationally excited states. The relationship between the population in the vibrationally excited states and
.243 thermal expansion is discussed (*vide supra*). This theoretical description implies that the steepness of the
.244 PEC, significantly, is directly related to the molecular thermal expansion: The steepness of the PEC has an
.245 inverse relationship with the expansion of a single molecule. Having established the theoretical description
.246 of thermal expansion, quantum chemical simulations were conducted to describe the PEC of **SUPERBIO**
.247 and **BPA-SEP** repeating unit, and analyse and understand the origin of the single molecular expansion of the
.248 **SUPERBIO** polymer than the **BPA-SEP** polymer based on the steepness of the PEC.

.249

.250 Theoretical models of ‘repeating units’ of each polymer are selected with an assumption that the CTE value
.251 of the polymer is not significantly different from the repeating unit’s value. This is a reasonable assumption
.252 since the CTE value measures the relative change. The ground state geometries are obtained with density
.253 functional theory (DFT) using the B3LYP functional with 6-31G* basis sets. Based on the ground state
.254 geometry, each bond is elongated by 0.1% up to 2.7% for the **SUPERBIO** repeating unit and 2.3% for the
.255 **BPA-SEP** repeating unit, except the bonds involving hydrogen atoms. Each geometry is re-optimized to
.256 consider the relaxation effect from angle changes. The length of the repeating unit is measured between two
.257 non-hydrogen terminal atoms (left-end oxygen atom and right-end carbon atom). The ground state
.258 geometries of the **SUPERBIO** and **BPA-SEP** repeating units are described in Figure 3b. All quantum
.259 chemical simulations were performed using Q-Chem 4.3.^{S41}

.260

.261 **Figure 3b (Main text).** Optimized ground state geometries of (Left) **SUPERBIO** and (Right) **BPA-SEP**
.262 repeating units. Colour scheme: carbon atom, black; hydrogen atom, white; oxygen atom, red; sulphur atom,
.263 yellow.

.264 Interestingly, a larger energy is required to elongate the **SUPERBIO** repeating unit compared with the **BPA-**
.265 **SEP** repeating unit to attain the same degree of geometric alternation (Figure 3c). The potential energy along
.266 the relative length change of each repeating unit can be well described using a harmonic approximation
.267 within the relatively small distortion region. The second order polynomial fitting gives the coefficient of the
.268 square term, which represents the steepness of the potential wall as 94132 and 59999 for the **SUPERBIO**
.269 and **BPA-SEP** repeating units, respectively. As discussed above, this result implies that the **SUPERBIO**
.270 repeating unit features a larger vibration energy gap than that of the **BPA-SEP** repeating unit, which induces
.271 a smaller single molecular expansion value. Considering that ISB consists of only single bonds, it was
.272 expected to require less energy to force bond lengthening compared with the BPA part, which contains
.273 benzene groups. However, the simulation outcome might indicate that the ring structure of ISB induces
.274 higher geometric restraint when it is forced to have elongated bonds compared with the planar benzene group
.275 in BPA. More interestingly, the close examination of the harmonic fitting curve reveals that the steepness of
.276 **SUPERBIO**’s PEC keeps increasing (the harmonic fitting curve tends to underestimate the last three data

.277 points). Meanwhile, the steepness of the **BPA-SEP** repeating unit remains relatively unchanged. The ratio of
.278 steepness between **SUPERBIO** and **BPA-SEP** repeating units is 1.41, 1.43, and 1.57, estimated with the
.279 first 5, first 10, and all data points.

.280
.281 **Figure 3c (Main text).** Potential energy curve versus the relative length of the **SUPERBIO** and **BPA-SEP**
.282 repeating units. (**SUPERBIO**: Blue circle, **BPA-SEP**: Orange square) The second order polynomial fitting is
.283 given as a solid line.

.284
.285
.286
.287
.288
.289
.290
.291
.292
.293
.294

.295 14. There are grammatical and other writing style issues in this manuscript that need to be addressed. For
.296 example”:

.297 a. Page 2 line 26: “However, the research efforts” should be “However, previous research efforts” or
.298 similar

.299

.300 **Answer:** We have modified the sentence as follows.

.301 **(Page 2 Line 26)**

.302 However, previously reported bio-derived thermosets or thermoplastics rarely offer thermal/mechanical
.303 properties, scalability, or recycling that match those of petrochemical SEPs.

.304

.305 b. Listing all of the properties of other polymers does not seem appropriate for the abstract

.306

.307 **Answer:** We have modified the Abstract.

.308 (Abstract, Page 2, Line 24) Environmental and health concerns force the search for sustainable super
.309 engineering plastics (SEPs) that utilise bio-derived cyclic monomers, e.g. isosorbide instead of restricted
.310 petrochemicals. However, previously reported bio-derived thermosets or thermoplastics rarely offer
.311 thermal/mechanical properties, scalability, or recycling that match those of petrochemical SEPs. Here, using
.312 a phase transfer catalyst, we synthesised an isosorbide-based polymer with a high molecular weight > 100
.313 kg mol⁻¹, which is reproducible at a 1-kg-scale production. It is transparent and solvent/melt-processible for
.314 recycling, with a glass transition temperature of 212 °C, a tensile strength of 78 MPa, and a thermal
.315 expansion coefficient of 23.8 ppm K⁻¹. Such a performance combination has not been reported before for
.316 bio-based thermoplastics, petrochemical SEPs, or thermosets. Interestingly, quantum chemical simulations
.317 show the alicyclic bicyclic ring structure of isosorbide imposes stronger geometric restraint to polymer chain
.318 than the aromatic group of bisphenol-A. This holds great potential as a recyclable load-bearing material for
.319 optoelectronics and bio-devices.

.320

.321 c. Page 3 line 47, “these researches have” should be “this research has”

.322

.323 **Answer:** We have deleted the sentence because we modified the entire Introduction Section. Please refer to

.324 question #12 of reviewer #3.

.325

.326 *d. Page 3 line 53, “with a high T_g of >150 C than engineering plastics...” – are you trying to say that the*
.327 *T_g 's of SPEs are 150 C greater than that of EPs?*

.328

.329 **Answer:** We have modified the sentences as follows.

.330 According to superiority of thermal and mechanical performances, thermoplastics are generally classified in
.331 the following order: commodity plastics < engineering plastics (EPs) < super engineering plastics (SEPs).

.332 There is no appropriate quantitative standard for the precise classification because most physical properties
.333 of thermoplastics exist across all the above-mentioned three classes.^{7,8} In polymer science, glass transition
.334 temperature (T_g) is a general indicator to represent thermomechanical characteristics of polymers. In the
.335 same order, the three classes of thermoplastics typically have the T_g ranges of <100, 100–150, and >150 °C.

.336 1,7–10

.337

.338 *e. Page 4 line 80: listing specific figure and table numbers in the introduction is not appropriate*

.339

.340 **Answer:** We appreciate the reviewer's comment. Fig. 1c,d and Supplementary Table 1–3, appearing in the
.341 Introduction Section, inevitably represent the outlook for the status of bio-based high T_g polymers. We have
.342 corrected the figure and table numbers throughout the manuscript.

.343

.344 *f. Sentence at the end of the introduction seems out of place*

.345

.346 **Answer:** As suggested by the reviewer, we have deleted the sentence.

.347 ~~Bio-based poly aromatic amide (aramid) and polyimide are pseudo-thermoplastics because they are not~~
.348 ~~melt/solvent processible (Supplementary Table 2).~~

.349

.350 *g. Page 9 line 167: “became coloured relatively as less as..”*

.351

.352 **Answer:** As suggested by the reviewer, we have modified the sentence.

.353 The **SUPERBIO** bar became brown relatively as less as the **BPA-SEP** one without an antioxidant.

.354

REVIEWERS' COMMENTS:

Reviewer #1 (Remarks to the Author):

Authors did a remarkable effort to account for all my major and minor comments. I then believe that the revised manuscript can be published as is.

Reviewer #2 (Remarks to the Author):

All my comments have been addressed.

Reviewer #3 (Remarks to the Author):

The revised manuscript has sufficiently addressed reviewer concerns and is suitable for publication.

REVIEWERS' COMMENTS:

Reviewer #1 (Remarks to the Author):

Authors did a remarkable effort to account for all my major and minor comments. I then believe that the revised manuscript can be published as is.

We greatly appreciate the constructive and very detailed comments and suggestions from this reviewer.

Reviewer #2 (Remarks to the Author):

All my comments have been addressed.

We greatly appreciate the constructive and very detailed comments and suggestions from this reviewer.

Reviewer #3 (Remarks to the Author):

The revised manuscript has sufficiently addressed reviewer concerns and is suitable for publication.

We greatly appreciate the constructive and very detailed comments and suggestions from this reviewer.